# A Minimum Variance Path Principle for Accurate and Stable Score-Based Density Ratio Estimation

**Wei Chen[1], Jiacheng Li[2], Shigui Li[1], Zhiqi Lin[3], Junmei Yang[2], John Paisley[4], Delu Zeng[2,5]***

[1]School of Mathematics, South China University of Technology
[2]School of Electronic and Information Engineering, South China University of Technology
[3]School of Computer Science and Engineering, South China University of Technology
[4]Department of Electrical Engineering, Columbia University
[5]Department of Electrical and Computer Engineering, University of Waterloo
{maweichen,eejiacheng_li,lishigui,202311089192}@mail.scut.edu.cn,
{yjunmei,dlzeng}@scut.edu.cn, jpaisley@columbia.edu

## Abstract

Score-based methods are powerful across machine learning, but they face a paradox: theoretically path-independent, yet practically path-dependent. We resolve this by proving that practical training objectives differ from the ideal, ground-truth objective by a crucial, overlooked term: the path variance of the score function. We propose the MVP (**M**imum **V**ariance **P**ath) Principle to minimize this path variance. Our key contribution is deriving a closed-form expression for the variance, making optimization tractable. By parameterizing the path with a flexible Kumaraswamy Mixture Model, our method learns data-adaptive, low-variance paths without heuristic manual selection. This principled optimization of the complete objective yields more accurate and stable estimators, establishing new state-of-the-art results on challenging benchmarks and providing a general framework for optimizing score-based interpolation. Our code can be found in code for MVP.

## 1 Introduction

Density Ratio Estimation (DRE) is a fundamental machine learning task, underpinning applications such as $f$-divergence estimation (Chen et al., 2025c), Riesz regression (Kato, 2025b;c; 2026), aligning large language models (Higuchi & Suzuki, 2025), causal inference (Wang et al., 2025), domain adaptation (Wang et al., 2023), and likelihood-free inference (Cobb et al., 2024). Classical methods often fail under the "density-chasm" problem where two densities exhibit low overlap or large discrepancy (Rhodes et al., 2020). A key advance is the rise of continuous, score-based methods (Choi et al., 2022; Yu et al., 2025; Chen et al., 2025c), which express the log-density ratio as a path integral of a time-dependent score function along a smooth interpolation between two densities.

Theoretically, these methods are path-invariant: integrating the score along any smooth path recovers the exact target (Choi et al., 2022). However, a practical paradox emerges in practice: with neural network approximation, performance becomes significantly path-dependent, as shown in our preliminary experiments (see Fig. 1a). This sensitivity mirrors challenges in broader probabilistic inference and generative modeling (Sahoo et al., 2024; Strasman et al., 2025).

We trace this paradox to a subtle but crucial gap between the ideal objective, which minimizes the true mean-squared error to the ground-truth score, and the tractable score-matching loss used in practice. The two objectives differ by a path-dependent term that is typically ignored as a constant when the path is fixed (Choi et al., 2022). Our analysis reveals that this term is, in fact, the dominant factor driving performance differences across paths, akin to the impact of noise schedules in generative modeling (Kingma et al., 2021; Guo & Schwing, 2025; Strasman et al., 2025).

Our main contribution is to close this gap. We prove that the missing term is precisely the ***path variance*** of the ground-truth score, a quantity that must be minimized and cannot be treated as a

---

*Corresponding author.

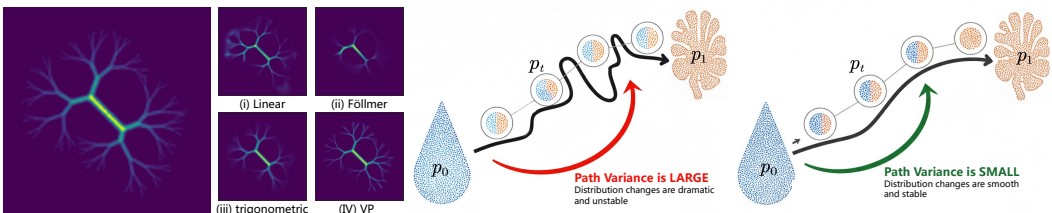

(a) Estimates vary across path settings.    (b) Path with large variance.    (c) Path with small variance.

Figure 1: (a) Preliminary experiments under various path settings. The left image denotes the ground truth. (b-c) Probability paths between $p_0$ and $p_1$ with large and small variance, respectively.

constant (illustrated in Figs. 1b and 1c and proved in Theorem 4.2). Building on this insight, we propose MVP (**M**imum **V**ariance **P**ath) Principle, the first framework that directly optimize the complete, ideal objective. Our contributions are threefold:

1. We formally identify path variance as the missing term bridging practical and ideal objectives, and derive closed-form, tractable expressions for it under two widely used interpolants.
2. We parameterize the path with a flexible Kumaraswamy Mixture Model (KMM) and learn an optimal, data-adaptive path by directly minimizing the analytical path variance.
3. Our approach resolves the empirical paradox described above and removes the need for heuristic path selection, achieving state-of-the-art performance on challenging benchmarks.

## 2    RELATED WORKS

**Score-Based Density Ratio Estimation.**    DRE is a fundamental statistical problem with a rich history (Sugiyama et al., 2012). Classical methods, such as those based on statistical divergences, like KLIEP (Sugiyama et al., 2008) or noise-contrastive principles (Gutmann & Hyvärinen, 2010; 2012), often falter when the supports of the two distributions have minimal overlap, which is a challenge known as the density-chasm problem (Liu et al., 2017). A recent paradigm shift addresses this via divide-and-conquer strategies that bridge $p_0$ and $p_1$. Starting with discrete methods like TRE (Rhodes et al., 2020), which decomposes the density ratio into a product of simpler intermediate ratios. It was later extended to the continuous setting in DRE-$\infty$ (Choi et al., 2022), linking DRE to score-based generative modeling through a time-dependent score along a smooth path. Later work improved this framework, for example, via the dequantified diffusion bridge interpolant (DDBI) for better stability and weaker assumptions (Chen et al., 2025c). Closely related is the work on conditional probability paths (Yu et al., 2025), providing theoretical guarantees for DRE by bounding the estimation error in terms of path smoothness, which is closely related to our Lemma 4.1. Other notable advances include improving robustness to outliers (Nagumo & Fujisawa, 2024), mitigating overfitting (Kato & Teshima, 2021), mapping distributions into simpler latent spaces (Choi et al., 2021; Wang et al., 2025), building direct DRE via low-variance secant alignment identity (Chen et al., 2025b) or Riesz regression (Kato, 2025a), and implicitly defining a probability path through endpoint densities (Kimura & Bondell, 2025; Yu et al., 2025). However, nearly all these score-based methods rely on fixed, hand-crafted interpolation paths. The crucial question of how to design an optimal, data-dependent path tailored to a specific DRE task remains largely unexplored.

**The Importance of Paths in Probabilistic Inference and Generative Modeling.**    A parallel evolution in the broader field of probabilistic inference (Paisley & Carin, 2009; Paisley et al., 2012a; Xu & Zeng, 2024) and generative modeling (Shen et al., 2025; Chen et al., 2025a; Xu et al., 2025a) highlights the path (a.k.a. noise) schedule as a critical design choice for capturing complex data structures (Li et al., 2023; 2025c; Lin et al., 2025). In conformal prediction, nested prediction sets are often built upon paths that guarantee calibrated uncertainty under distribution shift (Luo et al., 2026a;b; Bao et al., 2025a;b). In time-series forecasting, temporal paths and noise schedules directly govern how uncertainty propagates over multi-step horizons and how long-range dependencies are captured (Li et al., 2024a; 2025a;b). Furthermore, Gaussian processes inherently model functions as stochastic paths (Xu et al., 2025c; 2024; 2025b), with modern variants increasingly learning adaptive kernels and path densities to handle non-stationarity and high-dimensional manifolds.

Despite these advances, early score-based generative models like DDPM (Ho et al., 2020) employed fixed heuristics such as the variance-preserving (VP) schedule (Song et al., 2021b) or the cosine schedule (Nichol & Dhariwal, 2021). It quickly became evident that these choices were non-trivial, with theoretical analyses showing how schedules directly impact model stability and performance

(Song et al., 2021a; Sahoo et al., 2024; Strasman et al., 2025). This realization spurred a shift towards learnable, data-dependent paths, a trend prominent across the field, from variational diffusion models that learn schedules by maximizing a variational bound (Kingma et al., 2021; Guo & Schwing, 2025), to continuous normalizing flows that optimize the trajectory of a governing ODE (Zhang et al., 2024). A highly relevant advancement is neural diffusion models (Bartosh et al., 2024), which generalize the forward process by enabling learnable, non-linear, efficient (Li et al., 2024b), and time-dependent transformations of the data. While powerful for generative tasks, this line of work primarily optimizes the variational bound (Hoffman et al., 2013; Paisley et al., 2012b) for density estimation. Furthermore, the problem of finding an optimal path to reduce the estimation error of density ratios has been analyzed in the context of annealing, demonstrating that even basic paths offer provable benefits over no annealing (Chehab et al., 2023). A complementary perspective from optimal transport (OT) has been used to find geometrically optimal paths, as seen in rectified flow (Liu et al., 2023) and other OT-inspired models (Du et al., 2022; Shaul et al., 2023; Albergo et al., 2023). This body of work establishes a clear principle: for optimal performance in score-based models, the path should not be fixed but adapted to the data. While generative modeling has adopted this principle, DRE still has not.

**Bridging the Gap: Our Contribution.** We propose the first DRE framework that goes beyond fixed or heuristic paths by explicitly learning an optimal path through a theoretically grounded objective. Our analysis shows that this objective (the path variance) is the missing term in the ideal score-matching loss that prior DRE methods have overlooked. By parameterizing the path with a flexible KMM and optimizing it with its path variance, we convert the intractable functional search into a tractable, low-dimensional optimization. Our method, MVP, thus adapts the path schedule to the data distributions, achieving more accurate density ratio estimation.

## 3 BACKGROUND

Let $\mathbf{x}_0 \sim p_0$ and $\mathbf{x}_1 \sim p_1$ be random variables in $\mathbb{R}^d$ with $p_1 \ll p_0$. The goal of density ratio estimation (DRE) is to estimate $r(\boldsymbol{x}) = \frac{p_1(\boldsymbol{x})}{p_0(\boldsymbol{x})}$ for a given sample $\boldsymbol{x}$ using i.i.d. samples from $p_0$ and $p_1$. Conventional DRE methods often deteriorate when $p_0$ and $p_1$ are non-overlapping or significantly different (Gutmann & Hyvärinen, 2012; Liu et al., 2017).

**Interpolants.** To address this problem, Rhodes et al. (2020) proposed TRE, factorizing the direct ratio $r$ into $M$ ratios via intermediate variables $\mathbf{x}_{m/M} \sim p_{m/M}$ defined by

$$\mathbf{x}_{m/M} = \sqrt{1 - \beta(m/M)^2}\mathbf{x}_0 + \beta(m/M)\mathbf{x}_1, \quad m = 0, 1, \ldots, M, \tag{1}$$

where $\beta : [0, 1] \to [0, 1]$ is a monotone increasing schedule with $\beta(0) = 0, \beta(1) = 1$.

As $M \to \infty$, this discrete scheme converges to the deterministic interpolant (DI) (Choi et al., 2022)

$$\mathbf{x}_t = \alpha(t)\mathbf{x}_0 + \beta(t)\mathbf{x}_1, \quad \mathbf{x}_0 \sim p_0, \mathbf{x}_1 \sim p_1, \mathbf{x}_t \sim p_t, t \in [0, 1], \tag{2}$$

where $\{(\alpha(t), \beta(t))\}_{t \in [0,1]}$ is a path schedule satisfying the boundary conditions $\alpha(0) = \beta(1) = 1, \alpha(1) = \beta(0) = 0$. Common choices satisfy $\alpha(t) + \beta(t) = 1$ or $\alpha(t)^2 + \beta(t)^2 = 1$. Conditioning on $\mathbf{x}_1 = \boldsymbol{x}_1$ and assuming $\mathbf{x}_0$ follows a standard Gaussian, the transition kernel admits a closed form $p_t(\boldsymbol{x}_t \mid \boldsymbol{x}_1) = \mathcal{N}(\boldsymbol{x}_t; \beta(t)\boldsymbol{x}_1, \alpha(t)^2\boldsymbol{I}_d)$ (Vincent, 2011; Yu et al., 2025).

While the the Gaussian assumption on $\mathbf{x}_0 \sim p_0(\boldsymbol{x})$ is convenient, it may not hold for more general applications such as mutual information estimation. To relax this requirement and improve the stability of DRE-$\infty$, Chen et al. (2025c) extended DI to the dequantified diffusion bridge interpolant (DDBI) by adding a noise term to the interpolation, creating a bridge process of the form

$$\mathbf{x}_t = \alpha(t)\mathbf{x}_0 + \beta(t)\mathbf{x}_1 + \sqrt{t(1-t)\gamma^2 + (\alpha(t)^2 + \beta(t)^2)\varepsilon}\mathbf{z}, \tag{3}$$

where $\mathbf{z} \sim \mathcal{N}(\mathbf{0}, \boldsymbol{I}_d)$ is a Gaussian noise, $\gamma \geq 0$ is the noise factor, and $\varepsilon \geq 0$ is a small constant. In this case, the closed-form transition kernel conditioned on $\boldsymbol{x}_0$ and $\boldsymbol{x}_1$ becomes $p_t(\boldsymbol{x}_t \mid \boldsymbol{x}_0, \boldsymbol{x}_1) = \mathcal{N}(\boldsymbol{x}_t; \alpha(t)\boldsymbol{x}_0 + \beta(t)\boldsymbol{x}_1, (t(1-t)\gamma^2 + (\alpha(t)^2 + \beta(t)^2)\epsilon)\boldsymbol{I}_d)$ (Yu et al., 2025; Chen et al., 2025c).

**Density Ratio Estimation.** For the discrete path in Eq. (1), the log-density ratio for a given sample $\boldsymbol{x}$ is expressed as a telescoping summation of intermediate log-density ratios,

$$\log r(\boldsymbol{x}) = \log p_1(\boldsymbol{x}) - \log p_0(\boldsymbol{x}) = \sum_{m=0}^{M-1} \log \frac{p_{(m+1)/M}(\boldsymbol{x})}{p_{m/M}(\boldsymbol{x})}, \tag{4}$$

which is a Riemann sum approximation of the continuous representation (Choi et al., 2022),

$$\log r(\boldsymbol{x}) = \log p_1(\boldsymbol{x}) - \log p_0(\boldsymbol{x}) = \int_0^1 \partial_t \log p_t(\boldsymbol{x}) \mathrm{d}t \triangleq \int_0^1 s^{(t)}(\boldsymbol{x}, t) \mathrm{d}t. \qquad (5)$$

Here, $s^{(t)}(\boldsymbol{x}, t) \triangleq \partial_t \log p_t(\boldsymbol{x})$ is the *time* score, and $s^{(\boldsymbol{x})}(\boldsymbol{x}, t) \triangleq \partial_{\boldsymbol{x}} \log p_t(\boldsymbol{x})$ is the *data* score. Note that $(t)$ and $(\boldsymbol{x})$ are labels indicating "time" and "data", not derivatives.

To approximate $s^{(t)}$, we train a time score model $s_{\boldsymbol{\theta}}^{(t)}$ by minimizing the ideal but intractable time score matching objective, $\mathcal{L}_{\text{TSM}}(\boldsymbol{\theta}) = \mathbb{E}_{p(t)p_t(\boldsymbol{x})} \left| s^{(t)}(\boldsymbol{x}, t) - s_{\boldsymbol{\theta}}^{(t)}(\boldsymbol{x}, t) \right|^2$. Since $s^{(t)}$ is unknown, this loss is replaced by a tractable alternative derived via integration by parts, known as the sliced time score matching (STSM) [1] (Choi et al., 2022) objective,

$$\mathcal{L}_{\text{STSM}}(\boldsymbol{\theta}) = 2 \mathbb{E}_{p_0(\boldsymbol{x}_0)p_1(\boldsymbol{x}_1)} \left[ s_{\boldsymbol{\theta}}^{(t)}(\boldsymbol{x}_0, 0) - s_{\boldsymbol{\theta}}^{(t)}(\boldsymbol{x}_1, 1) \right] + \mathbb{E}_{p(t)p_t(\boldsymbol{x})} \left[ 2 \partial_t s_{\boldsymbol{\theta}}^{(t)}(\boldsymbol{x}, t) + s_{\boldsymbol{\theta}}^{(t)}(\boldsymbol{x}, t)^2 \right], \quad (6)$$

where the first term enforces boundary conditions, $\partial_t s_{\boldsymbol{\theta}}^{(t)}$ is the time derivative, and $p(t)$ is a timestep distribution over $(0, 1)$ (see Appendix Sec. B.2.5 for details). This objective is fully computable, depending only on $s_{\boldsymbol{\theta}}^{(t)}$ and its derivative, and differs from $\mathcal{L}_{\text{TSM}}$ by a path-dependent constant. The log-density ratio is estimated by $\log \hat{r}(\boldsymbol{x}) = \int_0^1 s_{\boldsymbol{\theta}}^{(t)}(\boldsymbol{x}, t) \mathrm{d}t$.

# 4 MINIMUM PATH VARIANCE PRINCIPLE

Our proposed method, MVP, is designed to resolve an important discrepancy between the theory and practice of score-based DRE. In this section, we detail the core components of our method: the derivation of the analytical path variance and our flexible KMM-based path parameterization.

## 4.1 THE PATH VARIANCE: AN OVERLOOKED TERM IN THE OBJECTIVE

Our analysis begins by relating the expected density ratio estimation error to the score-matching objective. We consider the estimation error,

$$\Delta(\boldsymbol{x}) = |\log r(\boldsymbol{x}) - \log \hat{r}(\boldsymbol{x})|^2, \quad \boldsymbol{x} \sim p_1, \qquad (7)$$

where the true and estimated log-density ratios are given by $\log r(\boldsymbol{x}) = \int_0^1 s^{(t)}(\boldsymbol{x}, t) \mathrm{d}t$ and $\log \hat{r}(\boldsymbol{x}) = \int_0^1 s_{\boldsymbol{\theta}}^{(t)}(\boldsymbol{x}, t) \mathrm{d}t$. The lemma below provides an explicit upper bound for this error.

**Lemma 4.1.** *Let $\{p_t\}_{t \in [0,1]}$ be a probability path connecting $p_0$ and $p_1$. Under Assumption A.1, the expected estimation error admits the following upper bound:*

$$\mathbb{E}_{p_1(\boldsymbol{x})} [\Delta(\boldsymbol{x})] \leq \left( \frac{e^{nL} - 1}{nL} \right)^{\frac{1}{n}} \left( \int_0^1 \left( \mathbb{E}_{p_t(\boldsymbol{x})} \left[ \epsilon(\boldsymbol{x}, t)^2 \right] \right)^m \mathrm{d}t \right)^{\frac{1}{m}}, \qquad (8)$$

*where $\epsilon(\boldsymbol{x}, t) = s^{(t)}(\boldsymbol{x}, t) - s_{\boldsymbol{\theta}}^{(t)}(\boldsymbol{x}, t)$ is the error of the score model, the exponents satisfy $\frac{1}{n} + \frac{1}{m} = 1$, and $L$ is the Lipschitz constant in Assumption A.1.*

**Relationship to Existing Error Bounds.** Our derivation in Lemma 4.1 is conceptually aligned with prior theoretical analyses of path-based density estimation. For example, Yu et al. (2025) also obtain an error bound by controlling the path integral via smoothness assumptions (e.g., Lipschitz constants), highlighting the central role of path regularity. However, important differences remain. Their bound is expressed in terms of statistical divergence between the true and estimated density ratios, whereas our Lemma 4.1 provides a direct upper bound on the expected squared error of the log-ratio estimate. Moreover, the goals diverge: Yu et al. (2025) leverage their bound to motivate the $\mathcal{L}_{\text{CTSM}}$ loss, while our analysis uses the general bound to decompose the total theoretical error and justify explicitly optimizing the previously unaddressed path variance term $\mathcal{V}$.

The proof is presented in Sec. A.2. The case $(n = \infty, m = 1)$ is particularly insightful, as it simplifies the bound to be proportional to the integrated squared error, $\int_0^1 \mathbb{E}_{p_t(\boldsymbol{x})} [\epsilon(\boldsymbol{x}, t)^2] \mathrm{d}t = \mathbb{E}_{p(t)p_t(\boldsymbol{x})} |\epsilon(\boldsymbol{x}, t)|^2$, which is exactly the score-matching loss, $\mathcal{L}_{\text{TSM}}(\boldsymbol{\theta})$. This reveals that minimizing the TSM loss, $\mathcal{L}_{\text{TSM}}(\boldsymbol{\theta})$, is essential for achieving minimum estimation error.

---

[1] We adopt the term "sliced" because the objective, derived through integration by parts, involves projecting onto random directions, a technique known as Sliced Score Matching (Song et al., 2020) in the generative modeling literature. Choi et al. (2022) simply call this objective Time Score Matching.

However, as established in Choi et al. (2022), one cannot directly optimize $\mathcal{L}_{\text{TSM}}$. Instead, a tractable objective like $\mathcal{L}_{\text{STSM}}$ is used. The connection between them is an exact algebraic identity:

$$\mathcal{L}_{\text{TSM}}(\boldsymbol{\theta}) = \mathcal{L}_{\text{STSM}}(\boldsymbol{\theta}) + \int_0^1 \mathbb{E}_{p_t(\boldsymbol{x})} \left| \partial_t \log p_t(\boldsymbol{x}) \right|^2 \mathrm{d}t. \tag{9}$$

This identity reveals a crucial gap: Minimizing the tractable loss $\mathcal{L}_{\text{STSM}}$ does not account for the second term, $\int_0^1 \mathbb{E}_{p_t(\boldsymbol{x})} |\partial_t \log p_t(\boldsymbol{x})|^2 \mathrm{d}t$. The main contribution of our work is to identify this overlooked term and propose a method to minimize it. We first present the following theorem.

**Theorem 4.2** (Minimum Path Variance Principle)**.** *In the loss decomposition (Eq. (9)), the second-moment term is exactly the path variance $\mathcal{V} \triangleq \int_0^1 \mathrm{Var}_{p_t(\boldsymbol{x})}(\partial_t \log p_t(\boldsymbol{x})) \mathrm{d}t$ of a probability path $\{p_t\}_{t \in [0,1]}$. Consequently, the expected estimation error of a model $s_{\boldsymbol{\theta}}^{(t)}$ is bounded by*

$$\mathbb{E}_{p_1(\boldsymbol{x})}[\Delta(\boldsymbol{x})] \leq e^L \left[ \mathcal{L}_{\text{STSM}}(\boldsymbol{\theta}) + \int_0^1 \mathrm{Var}_{p_t(\boldsymbol{x})}(\partial_t \log p_t(\boldsymbol{x})) \mathrm{d}t \right]. \tag{10}$$

*This establishes the **M**inimum **V**ariance **P**ath (**MVP**) principle: Minimizing the overall error bound requires jointly minimizing the tractable model loss $\mathcal{L}_{\text{STSM}}$ and the path variance $\mathcal{V}$.*

See Appendix Sec. A.3 for proof. This theorem reveals that path variance is not an auxiliary regularizer but the *overlooked component* of the ideal objective. Conventional approaches fix the path *a priori*, treating $\mathcal{V}$ as constant and ignoring its effect. Our key insight is that, by learning the path itself, we can explicitly minimize this term, not as a guarantee of a tighter bound, but as a heuristic that empirically improves estimation accuracy and stability.

**Remark on the Lipschitz Constant $L$ in Theorem 4.2.** Our principle of minimizing the path variance $\mathcal{V}$ also serves to indirectly control the Lipschitz constant $L$ that appears as a multiplicative factor in the error bound. We discuss the empirical nature of this relationship in Remark A.3.

**Remark on Stabilizing $\mathcal{L}_{\text{STSM}}$ in Theorem 4.2.** The score matching loss $\mathcal{L}_{\text{STSM}}$ is optimized with respect to $\boldsymbol{\theta}$ at every training step, but its empirical estimate depends on the current path $p_{t \in [0,1]}$ through samples $\boldsymbol{x}_t \sim p_t(\boldsymbol{x})$. To reduce the sensitivity of this estimate to path-induced score instability, we adopt an alternating scheme: after each update of the path parameters (see Sec. 4.2) by minimizing the path variance $\mathcal{V}$, we refresh the variance-based time sampler $p(t) \propto 1/(\mathrm{Var}_{p_t}(\partial_t \log p_t) + \varepsilon)$ (see Sec. B.2.5). This shifts sampling toward time steps with low score variance, reducing gradient noise and improving the reliability of the stochastic estimator of $\mathcal{L}_{\text{STSM}}$. Although this procedure does not minimize $\mathcal{L}_{\text{STSM}}$ with respect to the path, it stabilizes its optimization by controlling its variance dependence on $p_t$. Thus, minimizing $\mathcal{V}$ indirectly facilitates a tighter overall error bound by ensuring stable and effective optimization of the model loss.

**Closed-form Path Variance.** Directly minimizing $\mathcal{V}$ is nontrivial since it depends on the unknown time score $s^{(t)}(\boldsymbol{x}, t) \triangleq \partial_t \log p_t(\boldsymbol{x})$. We therefore derive analytical, computable forms of $\mathcal{V}$ that depend only on path coefficients and data distribution moments. Closed-form results are obtained for both the DI (Eq. (2)) and the DDBI (Eq. (3)) interpolants, summarized in the following proposition.

**Proposition 4.3.** *The path variance $\mathcal{V}$ for the DI and DDBI interpolants, which depends on the path schedules $(\alpha, \beta) = \{\alpha(t), \beta(t)\}_{t \in [0,1]}$, can be expressed in the following analytical forms:*

*1. **For the DI in Eq. (2)** with a Gaussian prior $p_0(\boldsymbol{x}) = \mathcal{N}(\boldsymbol{x}; \boldsymbol{0}, \boldsymbol{I}_d)$:*

$$\mathcal{V}_{DI}[\alpha, \beta] = \int_0^1 \left( \frac{2d\dot{\alpha}(t)^2}{\alpha(t)^2} + \frac{\dot{\beta}(t)^2}{\alpha(t)^2} \mathbb{E}_{p_1(\boldsymbol{x}_1)} \left[ \|\boldsymbol{x}_1\|^2 \right] \right) \mathrm{d}t. \tag{11}$$

*2. **For the DDBI in Eq. (3)** with noise schedule $\sigma_t^2 = t(1-t)\gamma^2 + (\alpha(t)^2 + \beta(t)^2)\varepsilon$:*

$$\mathcal{V}_{DDBI}[\alpha, \beta] = \int_0^1 \left( \frac{d}{2} \frac{(\dot{\sigma}_t^2)^2}{(\sigma_t^2)^2} + \frac{\mathbb{E}_{p_0(\boldsymbol{x}_0)p_1(\boldsymbol{x}_1)} \left\| \dot{\alpha}(t)\boldsymbol{x}_0 + \dot{\beta}(t)\boldsymbol{x}_1 \right\|^2}{\sigma_t^2} \right) \mathrm{d}t, \tag{12}$$

The proofs for these results are provided in Appendix Sec. A.4. These analytical forms are the key technical contribution that enables the practical implementation of our **Minimum Path Variance** principle, as they provide a direct, computable target for optimizing the path schedules.

## 4.2 MINIMIZING THE PATH VARIANCE VIA KMM PARAMETERIZATION

The analytical expressions in Proposition 4.3 transform the problem of minimizing the path variance into a functional optimization problem over the space of valid path schedules $(\alpha, \beta)$. To solve this tractably, we introduce a flexible, low-dimensional parameterization for the path that satisfies the necessary boundary conditions and monotonicity constraints by construction. This transforms an infinite-dimensional functional optimization into a low-dimensional parametric one, offering superior stability and tractability.

**A Function Class with Built-in Constraints.** A valid schedule $\{\alpha(t)\}_{t\in[0,1]}$ must satisfy the boundary conditions $\alpha(0) = 1, \alpha(1) = 0$ and be monotonically decreasing. Instead of enforcing these properties with external constraints, we select a function class that satisfies them inherently. We achieve this by defining $\alpha$ based on the cumulative distribution function (CDF) of a probability distribution defined on $[0, 1]$. A CDF, $F(t)$, is by definition a monotonically increasing function from 0 to 1. Therefore, the transformation $1 - F(t)$ yields a function that is monotonically decreasing from 1 to 0, matching our requirements for $\{\alpha(t)\}_{t\in[0,1]}$.

**The Kumaraswamy Mixture Model (KMM).** While a single distribution on $[0, 1]$ (e.g., Beta or Kumaraswamy) could define a valid path, its expressive power is limited, being unimodal. To construct a significantly more flexible function class, we employ a **Kumaraswamy Mixture Model (KMM)** (Kumaraswamy, 1980; Sindhu et al., 2015). The Kumaraswamy distribution is chosen over the more common Beta distribution for its simple and closed-form CDF, which allows for efficient and stable computation of derivatives. By taking a weighted sum of several Kumaraswamy components, we can create complex, multimodal probability density shapes, which in turn allows the path schedule to have varying rates of change to optimally minimize variance.

Formally, we define a $K$-component KMM, parameterized by $\phi = \{w_k, a_k, b_k\}_{k=1}^K$, via its probability density function (PDF) and CDF,

$$p_\phi(t) = \sum_{k=1}^K w_k \cdot \text{KS}(t; a_k, b_k), \quad F_\phi(t) = \sum_{k=1}^K w_k \left[ 1 - (1 - t^{a_k})^{b_k} \right], \tag{13}$$

where $\text{KS}(t; a, b) = abt^{a-1}(1 - t^a)^{b-1}$ is the Kumaraswamy PDF, $w_k$ are mixture weights satisfying $\sum w_k = 1$, and $a_k, b_k > 0$ are shape parameters.

**Final Construction and Guaranteed Properties.** We now define our learnable path schedule $\alpha_\phi(t)$ using the KMM's CDF. The derivative $\dot{\alpha}_\phi(t)$ is simply the negative of the KMM's PDF,

$$\alpha_\phi(t) = 1 - F_\phi(t), \quad \dot{\alpha}_\phi(t) = -p_\phi(t). \tag{14}$$

This construction guarantees the following desired properties without external enforcement:

- Boundary Conditions: $\alpha_\phi(0) = 1 - F_\phi(0) = 1$ and $\alpha_\phi(1) = 1 - F_\phi(1) = 0$.
- Monotonicity: $\alpha_\phi(t)$ is guaranteed to be monotonically decreasing since $p_\phi(t) \geq 0$.
- Smoothness: $\alpha_\phi(t)$ is infinitely differentiable on $(0, 1)$, ensuring a smooth probability flow.

The coupling schedule $\beta_\phi(t)$ and its derivative follow from the constraint selected below.

$$\text{Affine: } \alpha_\phi(t) + \beta_\phi(t) = 1 \quad \Rightarrow \quad \beta_\phi(t) = 1 - \alpha_\phi(t), \quad \dot{\beta}_\phi(t) = -\dot{\alpha}_\phi(t). \tag{15}$$

$$\text{Spherical: } \alpha_\phi(t)^2 + \beta_\phi(t)^2 = 1 \quad \Rightarrow \quad \beta_\phi(t) = \sqrt{1 - \alpha_\phi(t)^2}, \quad \dot{\beta}_\phi(t) = -\frac{\alpha_\phi(t)\dot{\alpha}_\phi(t)}{\beta_\phi(t)}. \tag{16}$$

Substituting into the path variance expressions in Proposition 4.3, we recast the functional optimization as a finite-dimensional problem over the KMM parameters $\phi$, namely minimizing $\mathcal{V}[\alpha_\phi, \beta_\phi]$.

**Parameterization for Unconstrained Optimization.** To perform gradient-based optimization for $\phi$, we must handle the constraints on the KMM parameters ($w_k > 0, \sum w_k = 1, a_k > 0, b_k > 0$). We use a standard reparameterization technique, optimizing a set of unconstrained latent variables $\hat{\phi} = \{\hat{w}_k, \hat{a}_k, \hat{b}_k\}$ which are mapped to valid KMM parameters via softmax and softplus transformations,

$$w_k = \frac{e^{\hat{w}_k}}{\sum_{j=1}^K e^{\hat{w}_k}}, \quad a_k = \text{softplus}(\hat{a}_k) = \log(1 + e^{\hat{a}_k}), \quad b_k = \text{softplus}(\hat{b}_k). \tag{17}$$

The path parameters $\phi$ and score model parameters $\boldsymbol{\theta}$ are jointly optimized via the total objective $\mathcal{L}_{\text{total}} = \lambda_1 \mathcal{L}_{\text{STSM}}(\boldsymbol{\theta}) + \lambda_2 \mathcal{V}[\alpha_{\boldsymbol{\phi}}, \beta_{\boldsymbol{\phi}}]$ (described in Appendix Sec. B.2). Details of path optimization are given in Algorithm 2, and a PyTorch implementation is provided in Algorithm 3 (see appendix). After training, MVP uses the same inference as conventional DRE methods shown in Algorithm 1.

---

**Algorithm 1** DRE with MVP: Inference.

---

**Require:** Sample $\boldsymbol{x}$, optimal parameters $\boldsymbol{\theta}^\star$ and $\boldsymbol{\phi}^\star$, integration steps $I$.
1: $t_i = i/I, i = \{0, 1, \ldots, I\}$.
2: $s_i^{(t)} \leftarrow s_{\boldsymbol{\theta}^\star}^{(t)}(\boldsymbol{x}, t_i), i = \{0, 1, \ldots, I\}$.
3: $\log \hat{r}(\boldsymbol{x}) \leftarrow \frac{1}{I} \sum_{i=0}^{I} s_i^{(t)}$.
**Ensure:** $\hat{r}(\boldsymbol{x}) = \exp(\log \hat{r}(\boldsymbol{x}))$.

---

**Algorithm 2** Unconstrained Optimization of the MVP Path.

---

**Require:** Num components $K$, initial latent params $\hat{\boldsymbol{\phi}} = \{\hat{w}_k, \hat{a}_k, \hat{b}_k\}_{k=0}^{K}$, learning rate $\eta$.
1: Sample a batch of $N$ time steps $\{t_i\}_{i=1}^{N} \sim p(t)$ (described in Appendix Sec. B.2.5).
2: Transform $\hat{\boldsymbol{\phi}}$ to constrained KMM params $\boldsymbol{\phi} = \{w_k, a_k, b_k\}_{k=0}^{K}$ using Eq. (17).
3: Compute path schedules $\{(\alpha_{\boldsymbol{\phi}}(t_i), \beta_{\boldsymbol{\phi}}(t_i))\}_{i=1}^{N}$ and their derivatives using Eqs. (14) to (16).
4: Approximate the path variance $\mathcal{V}[\alpha_{\boldsymbol{\phi}}, \beta_{\boldsymbol{\phi}}]$ via Monte Carlo using either Eq. (11) or Eq. (12).
5: Compute gradients $\nabla_{\hat{\boldsymbol{\phi}}} \mathcal{V}[\alpha_{\boldsymbol{\phi}}, \beta_{\boldsymbol{\phi}}]$ via automatic differentiation.
6: Update latent params: $\hat{\boldsymbol{\phi}} \leftarrow \texttt{OptimizerStep}(\hat{\boldsymbol{\phi}}, \nabla_{\hat{\boldsymbol{\phi}}} \mathcal{V}[\alpha_{\boldsymbol{\phi}}, \beta_{\boldsymbol{\phi}}], \eta)$.
**Ensure:** Optimized latent parameters $\hat{\boldsymbol{\phi}}^\star$ defining the path function $\alpha_{\boldsymbol{\phi}^\star}$ and $\beta_{\boldsymbol{\phi}^\star}$.

---

## 5 EXPERIMENTS

We conduct a comprehensive set of experiments to validate the effectiveness of our Minimum Path Variance principle and demonstrate the superior performance of our proposed method, MVP.

### 5.1 BASELINE SETTING AND EXPERIMENTAL SETUP

Our evaluation is designed to cover a wide range of tasks and distribution types. We follow the experimental protocol established by Choi et al. (2022); Chen et al. (2025c), employing two primary interpolant structures based on the task requirements. For standard density estimation tasks where $p_0$ can be assumed to be a simple distribution, we use the DI framework defined in Eq. (2). For more general tasks such as $f$-divergence and mutual information estimation, where this assumption is restrictive, we use the more robust DDBI framework from Eq. (3).

To evaluate our approach, we benchmark against a comprehensive suite of common, fixed path schedules representing the current standard in the DRE and generative modeling literatures. These baselines include the Linear (Albergo et al., 2023), VP (Ho et al., 2020; Song et al., 2021b), Cosine (Nichol & Dhariwal, 2021), Föllmer (Föllmer, 2005; Chen et al., 2024), and Trigonometric (Albergo et al., 2023) paths, with detailed formulations provided in Sec. B.1. In contrast to these pre-defined paths, our MVP path schedule utilizes a learnable path parameterized by a KMM, as discussed in Sec. 4.2 above. The path is optimized by directly minimizing the analytical path variance expressions from Proposition 4.3, allowing it to adapt to the specific data distributions of a given task.

### 5.2 ABLATION STUDIES.

**KMM Components.** We first conduct an ablation study on the sensitivity of MVP to the number of components $K$ in the KMM path parameterization. Experiments are done under the most challenging settings for both tasks: mutual information (MI) estimation at $d = 160$ and density estimation on BSDS300. As shown in Tab. 1, a single component ($K = 1$) is too restrictive and performs poorly, highlighting the need for path flexibility. Performance improves markedly at $K = 2$ and reaches its peak around $K = 5$. Increasing the number of components further ($K = 8$) brings no benefit and may slightly hurt performance due to overfitting the path variance objective. Thus, MVP empirically remains robust, with $K \in [2, 5]$. In practice, we set $K = 5$, which achieves the best trade-off.

Table 1: Ablation study on the number of KMM components ($K$) on the most challenging mutual information (MI) estimation ($d = 160$) and density estimation (BSDS300) tasks. We use mean squared error (MSE) negative log-likelihood (NLL). Lower is better. We recommend $K = 5$.

| | $d = 160$ (MI = 40) | | **BSDS300** |
| $K$ | **Est. MI** | **MSE↓** | **NLL↓** |
| --- | --- | --- | --- |
| 1 | $40.60 \pm 0.66$ | 1.32 | $-48.51 \pm 0.65$ |
| 2 | $40.55 \pm 0.60$ | 1.09 | $-145.44 \pm 0.31$ |
| 5 | $40.59 \pm 0.55$ | 1.02 | $-143.97 \pm 0.22$ |
| 8 | $41.16 \pm 0.63$ | 2.32 | $-143.60 \pm 0.45$ |

**Path Constraints.** We treat the choice between the ***affine*** constraint ($\alpha(t) + \beta(t) = 1$) and the ***spherical*** constraint ($\alpha(t)^2 + \beta(t)^2 = 1$) as a hyperparameter. The ablation results reveal a clear, task-dependent pattern that provides practical guidance: 1) For density estimation on tabular data (see Tab. 4), the affine constraint performs best on lower-dimensional cases (POWER, GAS), whereas the spherical constraint is superior on higher-dimensional and more complex cases (HEPMASS, MINIBOONE, BSDS300). This indicates that the optimal path geometry is closely linked to the intrinsic complexity of the data manifold. 2) For MI estimation, the result is different. On Gaussian tasks with high discrepancy but simple geometry (see Tab. 3), the affine constraint consistently delivers better accuracy and stability. By contrast, on more challenging MI tasks with pathological geometries, we found that the spherical constraint proves more robust, particularly for datasets with discontinuities (Additive Noise) or complex dependencies (Gamma–Exponential). (See Tab. 6 in appendix for these additional experiments.) These findings validate our treatment of the path constraint as a data-dependent hyperparameter rather than a fixed choice. In practice, the affine constraint is well-suited for distributions with high discrepancy but simple geometry, while the spherical constraint is preferable for complex, high-dimensional, or non-Gaussian settings.

## 5.3 $f$-DIVERGENCE ESTIMATION

**Geometrically Pathological Distributions.** We first test robustness on five mutual information (MI) estimation tasks that can be viewed as challenging $f$-divergence problems. (For a formal MI definition, see Appendix Sec. B.4.) Following Czyż et al. (2023), these distributions exhibit concentrated densities, sharp discontinuities, heavy tails, and extreme correlations, all of which break standard estimator assumptions. Results in Tab. 2 show that fixed-path baselines degrade severely on these geometries, while MVP consistently achieves low error (see also Tab. 6 in the appendix). On the Additive Noise task with discontinuities and the Gamma–Exponential task with complex non-linear dependence, our method outperforms all baselines by a wide margin, confirming that path-variance optimization enables MVP to adapt to pathological structures effectively.

Table 2: MSE results on the Additive Noise (sharp discontinuities) and Gamma–Exponential (non-linear dependency) datasets. Across all correlation levels (top row of each sub-table), MVP achieves notably lower MSE than fixed-path baselines. Full results on 5 datasets given in Tab. 6 in appendix.

| Path | Constraint | 0.1 | 0.2 | 0.3 | 0.4 | 0.5 | 0.6 | 0.7 | 0.8 | 0.9 |
|------|-----------|-----|-----|-----|-----|-----|-----|-----|-----|-----|
| Linear | affine | 0.3053 | 0.0541 | 0.0233 | 0.0465 | 0.0367 | 0.0714 | 0.0339 | 0.0307 | 0.0268 |
| Cosine | spherical | 0.4803 | 0.1086 | 0.0788 | 0.0969 | 0.0719 | 0.0867 | 0.1015 | 0.1204 | 0.1304 |
| Föllmer | spherical | 0.4731 | 0.1048 | 0.0510 | 0.0916 | 0.0864 | 0.1024 | 0.1226 | 0.0944 | 0.1115 |
| Trigonometric | spherical | 0.3615 | 0.0565 | 0.0508 | 0.0654 | 0.0596 | 0.0517 | 0.0502 | 0.0448 | 0.0603 |
| VP | spherical | 0.3455 | 0.0290 | 0.0303 | 0.0346 | 0.0210 | 0.0313 | 0.0426 | 0.0390 | 0.0123 |
| **MVP (ours)** | affine | 0.2360 | 0.0199 | 0.0081 | 0.0167 | 0.0161 | 0.0216 | 0.0304 | 0.0155 | **0.0010** |
| **MVP (ours)** | spherical | **0.1016** | **0.0134** | **0.0078** | **0.0009** | 0.0138 | 0.0117 | 0.0196 | **0.0026** | 0.0029 |
| **Path** | **Constraint** | **1.0** | **1.1** | **1.2** | **1.3** | **1.4** | **1.5** | **1.6** | **1.7** | **1.8** |
| Linear | affine | 2.8166 | 0.7064 | 0.1686 | 0.1368 | 0.0513 | 0.0420 | 0.0051 | 0.0147 | 0.0066 |
| Cosine | spherical | 7.2466 | 1.0785 | 0.4875 | 0.2582 | 0.2802 | 0.3063 | 0.2936 | 0.1864 | 0.3438 |
| Föllmer | spherical | 0.4006 | 0.4275 | 0.6304 | 0.3741 | 0.2822 | 0.4662 | 0.2473 | 0.3838 | 0.3105 |
| Trigonometric | spherical | 10.4420 | 1.3841 | 0.2705 | 0.3854 | 0.0184 | 0.0681 | 0.1109 | 0.0103 | 0.0668 |
| VP | spherical | 2.7080 | 1.3872 | 0.1116 | 0.1224 | 0.1320 | 0.1817 | 0.4407 | 0.0919 | 0.2507 |
| **MVP (ours)** | affine | 0.4785 | 1.5466 | 0.1137 | 0.1274 | 0.0915 | **0.0063** | **0.0032** | 0.0294 | 0.0069 |
| **MVP (ours)** | spherical | **0.2721** | **0.2304** | **0.0917** | **0.0434** | **0.0075** | 0.0187 | 0.0091 | **0.0030** | **0.0004** |

**High-dimensional & High-discrepancy Distributions.** We further evaluate MI estimation in a high-dimensional, high-discrepancy setting, designed to induce the "density-chasm" problem in DRE (Rhodes et al., 2020). Performance is measured by MI accuracy and MSE against the true value. As shown in Tab. 3, most fixed-path baselines collapse as dimension $d$ and MI increase. For example, Föllmer and Cosine paths perform reasonably at low $d$ but diverge at $d = 160$, MI= $40$. In contrast, MVP with an affine constraint maintains a low MSE of $1.02$, outperforming the next best baseline by a large margin. These results highlight that adaptive path learning under our Minmimum Path Variance principle reliably bridges the density chasm in settings where heuristic fixed paths fail.

## 5.4 DENSITY ESTIMATION

To assess our model's ability to capture the true data distribution $p_{\text{data}}$, we perform density estimation by expressing complex, possibly intractable data distribution $p_{\text{data}}$ in terms of a simple base distribution $p_0$ (e.g., $\mathcal{N}(\mathbf{0}, \mathbf{I}_d)$) using the density ratio $r(\boldsymbol{x}) = p_{\text{data}}(\boldsymbol{x})/p_0(\boldsymbol{x})$. This yields $\log p_{\text{data}}(\boldsymbol{x}) =$

Table 3: Mutual information estimation under high-discrepancy settings (MI $\in \{10, 20, 30, 40\}$ nats). We report the estimated mutual information (mean $\pm$ std) and MSE across different path settings and constraint types. Bolded MSE values indicate the best performance for each dimension. Our MVP path demonstrates superior performance in most high-discrepancy settings.

| Path | Constraint | $d = 40$ (MI = 10) | | $d = 80$ (MI = 20) | | $d = 120$ (MI = 30) | | $d = 160$ (MI = 40) | |
| | | Est. MI | MSE | Est. MI | MSE | Est. MI | MSE | Est. MI | MSE |
| --- | --- | --- | --- | --- | --- | --- | --- | --- | --- |
| Linear | Affine | $9.79_{\pm 0.17}$ | 0.13 | $20.44_{\pm 0.34}$ | 0.44 | $30.18_{\pm 0.50}$ | 0.63 | $37.07_{\pm 0.51}$ | 9.17 |
| Cosine | Spherical | $9.62_{\pm 0.28}$ | 0.34 | $19.21_{\pm 0.47}$ | 1.19 | $28.60_{\pm 0.71}$ | 2.97 | $33.95_{\pm 0.71}$ | 37.91 |
| Föllmer | Spherical | $9.41_{\pm 0.27}$ | 0.54 | $17.94_{\pm 0.46}$ | 4.75 | $25.31_{\pm 0.70}$ | 23.06 | $31.51_{\pm 0.60}$ | 72.98 |
| Trigonometric | Spherical | $10.12_{\pm 0.24}$ | 0.13 | $19.88_{\pm 0.47}$ | 0.47 | $33.37_{\pm 0.74}$ | 12.41 | $41.74_{\pm 0.70}$ | 4.18 |
| VP | Spherical | $10.02_{\pm 0.34}$ | 0.28 | $20.08_{\pm 0.47}$ | 0.66 | $26.14_{\pm 0.57}$ | 15.53 | $33.47_{\pm 0.61}$ | 43.60 |
| MVP (ours) | Spherical | $9.81_{\pm 0.24}$ | 0.16 | $20.43_{\pm 0.44}$ | 0.62 | $28.88_{\pm 0.69}$ | 2.35 | $37.22_{\pm 0.72}$ | 8.79 |
| MVP (ours) | Affine | $10.08_{\pm 0.20}$ | **0.10** | $19.80_{\pm 0.36}$ | **0.34** | $30.22_{\pm 0.48}$ | **0.58** | $40.59_{\pm 0.55}$ | **1.02** |

$\log r(\boldsymbol{x}) + \log p_0(\boldsymbol{x})$, so that density estimation reduces to approximating $r(\mathbf{x})$, and the log-likelihood is estimated via $\log p_{\text{data}}(\boldsymbol{x}) \approx \log \hat{r}(\boldsymbol{x}) + \log p_0(\boldsymbol{x})$.

**Structured and Multi-modal Datasets.** We further evaluate MVP on six challenging datasets: circles, rings, pinwheel, 2spirals, checkerboard, and tree (Grathwohl et al., 2018; Karras et al., 2024), which feature disconnected components, intricate spirals, and discontinuous densities. As shown in Fig. 2, MVP yields sharper and more accurate density estimates on checkerboard and tree, and matches the best baseline (VP path) on the remaining datasets.

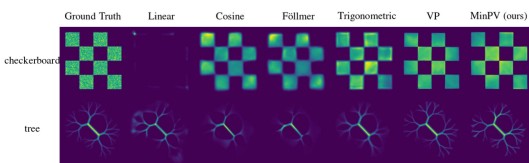

Figure 2: Density estimation on checkerboard and tree datasets. MVP successfully learns a data-adaptive path tailored to the specific manifold of each dataset. Full results in Fig. 4 of appendix.

Full results are provided in Fig. 4 in the appendix. These results demonstrate the significant adaptability of MVP and help confirm that minimizing path variance enables MVP to adapt effectively to diverse data geometries without manual path selection.

**Real-world Tabular Datasets.** Finally, we evaluate MVP on five widely-used tabular benchmarks (Grathwohl et al., 2018), reporting test negative log-likelihood (NLL), where lower is better. As shown in Tab. 4, MVP consistently outperforms all fixed-path baselines and achieves new state-of-the-art results across all datasets. Two key observations of note are: 1) On POWER and GAS, the affine constraint leads to the best performance, whereas the spherical constraint is more effective on the higher-dimensional datasets (HEPMASS, MINIBOONE, and BSDS300). This highlights the adaptability of MVP to data geometry. 2) The gains over strong baselines (e.g., VP and trigonometric paths) are substantial, especially on BSDS300 where MVP improves NLL by more than 10 points, underscoring the advantage of explicitly minimizing path variance. These results validate our central claim, that adaptively selecting the path via MVP yields a data-dependent path that aligns better with the underlying structure, leading to improved density estimation.

Table 4: Test Negative Log-Likelihood (NLL) on five tabular datasets (lower is better). Our MVP path consistently outperforms all fixed-path baselines, achieving SOTA across the benchmarks.

| Path | Constraint | POWER | GAS | HEPMASS | MINIBOONE | BSDS300 |
| --- | --- | --- | --- | --- | --- | --- |
| Linear | Affine | $-0.19_{\pm 0.04}$ | $-5.00_{\pm 0.06}$ | $19.64_{\pm 0.14}$ | $18.85_{\pm 0.22}$ | $-98.37_{\pm 0.48}$ |
| Cosine | Spherical | $0.55_{\pm 0.03}$ | $-1.72_{\pm 0.07}$ | $21.62_{\pm 0.05}$ | $24.36_{\pm 0.17}$ | $-75.39_{\pm 1.17}$ |
| Föllmer | Spherical | $0.52_{\pm 0.06}$ | $-0.45_{\pm 0.07}$ | $21.57_{\pm 0.09}$ | $25.66_{\pm 0.34}$ | $-82.87_{\pm 0.20}$ |
| Trigonometric | Spherical | $-0.54_{\pm 0.15}$ | $-4.57_{\pm 0.16}$ | $19.50_{\pm 0.05}$ | $20.27_{\pm 0.10}$ | $-89.41_{\pm 0.04}$ |
| VP | Spherical | $-0.04_{\pm 0.08}$ | $-6.02_{\pm 0.13}$ | $18.06_{\pm 0.16}$ | $18.25_{\pm 0.08}$ | $-131.90_{\pm 0.39}$ |
| MVP (ours) | Spherical | $-0.15_{\pm 0.11}$ | $-6.92_{\pm 0.21}$ | **$18.01_{\pm 0.16}$** | **$17.81_{\pm 0.09}$** | **$-143.97_{\pm 0.22}$** |
| MVP (ours) | Affine | **$-0.81_{\pm 0.08}$** | **$-6.95_{\pm 0.10}$** | $18.79_{\pm 0.16}$ | $17.99_{\pm 0.10}$ | $-129.73_{\pm 0.20}$ |

## 5.5 ILLUSTRATION OF OPTIMIZED PATH SCHEDULES

We visualize the optimized schedules $\alpha_\phi(t)$ and $\beta_\phi(t)$ across various tasks to interpret the efficacy of the MVP principle, with detailed comparisons provided in Fig. 3 and Sec. B.6. While Theorem 4.2 establishes a theoretical link between path variance and estimation error , we emphasize that our implementation treats the minimization of $\mathcal{V}$ as a principled heuristic rather than a strict minimization of the ground-truth error. This analytical proxy objective serves to encourage smoother path transitions and indirectly control the Lipschitz constant $L$. As illustrated in the optimized schedules for pathological and high-dimensional distributions Figs. 5 and 6, the learned paths (red) significantly deviate from fixed baselines. Specifically, MVP tends to adopt a more gradual rate of change near the boundaries $t \in \{0, 1\}$, which serves as a heuristic to suppress instantaneous velocity spikes in the time score that often lead to numerical instability.

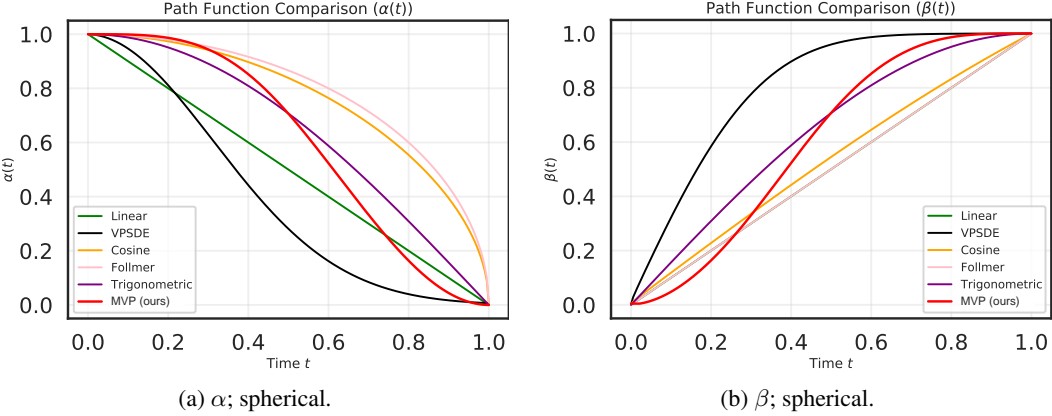

(a) $\alpha$; spherical.  (b) $\beta$; spherical.

Figure 3: Optimized KMM-parameterized path schedule for high-dimensional distributions.

This data-adaptive schedule is particularly effective for navigating the density-chasm encountered in high-dimensional settings. As shown in Figs. 3 and 6, while fixed schedules like Föllmer or cosine often traverse regions of vanishingly low density and subsequently diverge, the MVP path adapts its velocity to maintain more reliable score signals. The structural diversity of the learned schedules across structured manifolds Fig. 7 and real-world tabular datasets Fig. 8 further underscores the necessity of the flexible KMM parameterization. By allowing the schedule to adjust its rate of change to align with the specific manifold structure, MVP ensures that the probability flow is tailored to the data distribution. This alignment directly facilitates the substantial improvements in NLL and SOTA results reported in Tab. 4.

## 6 CONCLUSIONS AND FUTURE WORKS

This paper addresses the key paradox in score-based density ratio estimation (DRE) that estimator performance strongly depends on the chosen path schedule while being theoretically path-invariant. Our analysis reveals that this discrepancy arises from an overlooked path variance term in training objectives. Based on this observation, we established the **M**inimum **P**ath **V**ariance (MVP) principle, arguing that this term should be explicitly minimized rather than ignored. We then proposed MVP, a framework that derives a closed-form expression for path variance and directly optimizes it. By parameterizing the path with a Kumaraswamy Mixture (KMM) Model, MVP learns a data-adaptive low-variance path. This eliminates the need for heuristic path selection and ensures closer alignment with the ideal score-matching objective. Experiments demonstrate that MVP achieves SOTA performance in estimation error and stability across challenging DRE benchmarks.

**Future Works.** The MVP principle is general and could benefit other score-based generative models by learning optimal noise schedules to improve sample quality and stability. Beyond accuracy, future research could focus on computational efficiency, for example by leveraging optimal transport–based path formulations or adopting lighter parameterizations to further accelerate training. We believe that explicitly optimizing the geometry of interpolation paths will continue to afford significant advances in density ratio estimation and related score-based methods.

ACKNOWLEDGMENTS

This work was supported in part by grants from National Natural Science Foundation of China (52539005), the fundamental research program of Guangdong, China (2023A1515011281), the China Scholarship Council (202306150167), Guangdong Basic and Applied Basic Research Foundation (24202107190000687),Foshan Science and Technology Research Project(2220001018608).

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

## REPRODUCIBILITY STATEMENT

To ensure the reproducibility of our results, we have made every effort to provide a comprehensive account of our work. All experimental setups, including dataset pre-processing and hyperparameter settings, are detailed in the main paper (see Sec. 5) and the appendix (see Sec. B). We provide clear pseudocode for our core algorithms, including the optimization of the path parameters (Algorithm 2) and the final density ratio estimation procedure (Algorithm 1), with the core of our PyTorch implementation detailed in Algorithm 3. Furthermore, a complete implementation of our method, MVP, along with scripts to replicate all experiments, will be made publicly available upon publication.

## LLM DISCLAIMER

During the preparation of this manuscript, a large language model (LLM) was used as a tool to aid the research and writing process. The uses of the LLM were primarily for two purposes: (1) to assist in polishing the writing, including improving grammar, clarity, and phrasing of sentences; and (2) for retrieval and discovery, such as finding related work and summarizing existing literature to help situate our contributions within the broader academic context. The core ideas, theoretical derivations, experimental design, and interpretation of results were conducted by the authors.

## ETHICS STATEMENT

This paper presents foundational work aimed at advancing the field of machine learning, specifically in the area of density ratio estimation. The research does not involve human subjects, private data, or applications with direct ethical risks. While the direct societal impacts of this theoretical work are limited, we acknowledge that future extensions of our methods, particularly via our open-source codebase, could be applied in various domains. We encourage that any such application in sensitive contexts incorporate a domain-specific ethical review to assess potential impacts.

APPENDIX

APPENDIX CONTENTS

## A ASSUMPTIONS AND PROOFS

### A.1 ASSUMPTIONS

**Assumption A.1.** For every $\boldsymbol{x}$, the map $\tau \mapsto \log p_s \tau(\boldsymbol{x})$ is $L$-Lipschitz on $[0, 1]$ with a constant $L$ independent of $\boldsymbol{x}$, and $p_\tau(\boldsymbol{x}) > 0$ for the $\boldsymbol{x}$ under consideration. Consequently, $\tau \mapsto \log p_\tau(\boldsymbol{x})$ is absolutely continuous, its (weak) derivative exists for almost every $\tau \in [0, 1]$, and satisfies

$$|\partial_\tau \log p_\tau(\boldsymbol{x})| \leq L, \quad \text{a.e. } \tau \in [0, 1]. \tag{18}$$

**Assumption A.2.** Under Assumption A.1, for any $\boldsymbol{x}$ and any $0 \leq s \leq t \leq 1$, the time integral of the time score is bounded by

$$\log \frac{p_t(\boldsymbol{x})}{p_s(\boldsymbol{x})} = \int_s^t \partial_\tau \log p_\tau(\boldsymbol{x}) \mathrm{d}\tau \leq \int_s^t |\partial_\tau \log p_\tau(\boldsymbol{x})| \, \mathrm{d}\tau \leq L(t - s). \tag{19}$$

### A.2 PROOF OF LEMMA 4.1

**Lemma 4.1.** *Let $\{p_t\}_{t \in [0,1]}$ be a probability path connecting $p_0$ and $p_1$. Under Assumption A.1, the expected estimation error admits the following upper bound:*

$$\mathbb{E}_{p_1(\boldsymbol{x})}\left[\Delta(\boldsymbol{x})\right] \leq \left(\frac{e^{nL} - 1}{nL}\right)^{\frac{1}{n}} \left(\int_0^1 \left(\mathbb{E}_{p_t(\boldsymbol{x})}\left[\epsilon(\boldsymbol{x}, t)^2\right]\right)^m \mathrm{d}t\right)^{\frac{1}{m}}, \tag{8}$$

*where $\epsilon(\boldsymbol{x}, t) = s^{(t)}(\boldsymbol{x}, t) - s_{\boldsymbol{\theta}}^{(t)}(\boldsymbol{x}, t)$ is the error of the score model, the exponents satisfy $\frac{1}{n} + \frac{1}{m} = 1$, and $L$ is the Lipschitz constant in Assumption A.1.*

*Proof.* For simplicity, we denote $\epsilon(\boldsymbol{x}, t) = s^{(t)}(\boldsymbol{x}, t) - s_{\boldsymbol{\theta}}^{(t)}(\boldsymbol{x}, t)$. We can express the density ratio estimation error $\Delta(\boldsymbol{x})$ by applying the Cauchy-Schwarz inequality:

$$\begin{aligned}
\Delta(\boldsymbol{x}) &= \left|\log r(\boldsymbol{x}) - \log \hat{r}(\boldsymbol{x})\right|^2 \\
&= \left|\int_0^1 s^{(t)}(\boldsymbol{x}, t)\mathrm{d}t - \int_0^1 s_{\boldsymbol{\theta}}^{(t)}(\boldsymbol{x}, t)\mathrm{d}t\right|^2 \\
&= \left|\int_0^1 \left(s^{(t)}(\boldsymbol{x}, t) - s_{\boldsymbol{\theta}}^{(t)}(\boldsymbol{x}, t)\right)\mathrm{d}t\right|^2 = \left|\int_0^1 \epsilon(\boldsymbol{x}, t)\mathrm{d}t\right|^2 \\
&\leq \left(\int_0^1 1^2 \mathrm{d}t\right) \cdot \left(\int_0^1 \epsilon(\boldsymbol{x}, t)^2 \mathrm{d}t\right) \quad \text{(Cauchy-Schwarz inequality)} \\
&= \int_0^1 \epsilon(\boldsymbol{x}, t)^2 \mathrm{d}t.
\end{aligned} \tag{20}$$

Taking expectations over $p_1(\boldsymbol{x})$ and applying Tonelli's theorem (the integrand is nonnegative) gives:

$$\mathbb{E}_{p_1(\boldsymbol{x})}\left[\Delta(\boldsymbol{x})\right] \leq \mathbb{E}_{p_1(\boldsymbol{x})}\left[\int_0^1 \epsilon(\boldsymbol{x}, t)^2 \mathrm{d}t\right] = \int_0^1 \mathbb{E}_{p_1(\boldsymbol{x})}\left[\epsilon(\boldsymbol{x}, t)^2\right] \mathrm{d}t. \quad \left(\epsilon(\boldsymbol{x}, t)^2 \geq 0\right) \tag{21}$$

Next, we rewrite the inner expectation with respect to $p_1$ in terms of $p_t$. Using the change of measure:

$$\begin{aligned}
\mathbb{E}_{p_1(\boldsymbol{x})}\left[\epsilon(\boldsymbol{x}, t)^2\right] &= \int \epsilon(\boldsymbol{x}, t)^2 p_1(\boldsymbol{x})\mathrm{d}\boldsymbol{x} = \mathbb{E}_{p_t(\boldsymbol{x})}\left[\epsilon(\boldsymbol{x}, t)^2 \cdot \frac{p_1(\boldsymbol{x})}{p_t(\boldsymbol{x})}\right] \\
&= \mathbb{E}_{p_t(\boldsymbol{x})}\left[\epsilon(\boldsymbol{x}, t)^2 \cdot \exp\left(\log p_1(\boldsymbol{x}) - \log p_t(\boldsymbol{x})\right)\right] \\
&= \mathbb{E}_{p_t(\boldsymbol{x})}\left[\epsilon(\boldsymbol{x}, t)^2 \cdot \exp\left(\int_t^1 \partial_s \log p_s(\boldsymbol{x})\mathrm{d}s\right)\right] \\
&\leq \mathbb{E}_{p_t(\boldsymbol{x})}\left[\epsilon(\boldsymbol{x}, t)^2 \cdot \exp\left(\int_t^1 |\partial_s \log p_s(\boldsymbol{x})| \, \mathrm{d}s\right)\right] \quad (\star) \\
&\leq \mathbb{E}_{p_t(\boldsymbol{x})}\left[\epsilon(\boldsymbol{x}, t)^2 \cdot e^{L(1-t)}\right], \quad (\star\star)
\end{aligned} \tag{22}$$

where $(\star)$ follows since $\epsilon(\boldsymbol{x}, t)^2 \geq 0$ a.e., and $(\star\star)$ follows under Assumption A.1.

Substituting Eq. (22) into Eq. (21) yields the bound

$$
\begin{aligned}
\mathbb{E}_{p_1(\boldsymbol{x})}\left[\Delta(\boldsymbol{x})\right] &\leq \int_0^1 \mathbb{E}_{p_t(\boldsymbol{x})}\left[\epsilon(\boldsymbol{x}, t)^2 \cdot e^{L(1-t)}\right] \mathrm{d}t \\
&= \int_0^1 e^{L(1-t)} \mathbb{E}_{p_t(\boldsymbol{x})}\left[\epsilon(\boldsymbol{x}, t)^2\right] \mathrm{d}t \\
&\leq \left(\int_0^1 e^{nL(1-t)}\mathrm{d}t\right)^{\frac{1}{n}} \left(\int_0^1 \left(\mathbb{E}_{p_t(\boldsymbol{x})}\left[\epsilon(\boldsymbol{x}, t)^2\right]\right)^m \mathrm{d}t\right)^{\frac{1}{m}} \\
&= \left(\frac{e^{nL} - 1}{nL}\right)^{\frac{1}{n}} \left(\int_0^1 \left(\mathbb{E}_{p_t(\boldsymbol{x})}\left[\epsilon(\boldsymbol{x}, t)^2\right]\right)^m \mathrm{d}t\right)^{\frac{1}{m}},
\end{aligned}
\tag{23}
$$

where the second inequality follows from Hölder's inequality with $\frac{1}{n} + \frac{1}{m} = 1$. Different choices of $(n, m)$ yield different bounds:

- $(n = \infty, m = 1)$:
$$
\mathbb{E}_{p_1(\boldsymbol{x})}[\Delta(\boldsymbol{x})] \leq e^L \int_0^1 \mathbb{E}_{p_t(\boldsymbol{x})}[\epsilon(\boldsymbol{x}, t)^2]\mathrm{d}t,
\tag{24}
$$
which preserves the training loss in its original integral form, at the cost of a looser exponential factor $e^L$.

- $(n = 2, m = 2)$ (Cauchy–Schwarz):
$$
\mathbb{E}_{p_1(\boldsymbol{x})}[\Delta(\boldsymbol{x})] \leq \left(\frac{e^{2L} - 1}{2L}\right)^{\frac{1}{2}} \cdot \left(\int_0^1 \left(\mathbb{E}_{p_t(\boldsymbol{x})}[\epsilon(\boldsymbol{x}, t)^2]\right)^2 \mathrm{d}t\right)^{\frac{1}{2}},
\tag{25}
$$
which yields a tighter $L$-dependent factor but replaces the training loss with an $L^2$-type form.

- General $(n, m)$: one may also choose intermediate $(n, m)$ to trade off between the looseness of the exponential factor $\left(\int_0^1 e^{nL(1-t)}\mathrm{d}t\right)^{1/n}$ and the form of the training loss norm.

This finishes the proof.

$\square$

## A.3 PROOF OF THEOREM 4.2

**Theorem 4.2** (Minimum Path Variance Principle). *In the loss decomposition (Eq. (9)), the second-moment term is exactly the path variance* $\mathcal{V} \triangleq \int_0^1 \mathrm{Var}_{p_t(\boldsymbol{x})}(\partial_t \log p_t(\boldsymbol{x}))\mathrm{d}t$ *of a probability path* $\{p_t\}_{t \in [0,1]}$. *Consequently, the expected estimation error of a model* $s_{\boldsymbol{\theta}}^{(t)}$ *is bounded by*

$$
\mathbb{E}_{p_1(\boldsymbol{x})}[\Delta(\boldsymbol{x})] \leq e^L \left[\mathcal{L}_{\mathrm{STSM}}(\boldsymbol{\theta}) + \int_0^1 \mathrm{Var}_{p_t(\boldsymbol{x})}(\partial_t \log p_t(\boldsymbol{x}))\mathrm{d}t\right].
\tag{10}
$$

*This establishes the **M**inimum **V**ariance **P**ath (**MVP**) principle: Minimizing the overall error bound requires jointly minimizing the tractable model loss* $\mathcal{L}_{\mathrm{STSM}}$ *and the path variance* $\mathcal{V}$.

*Proof.* We begin from the upper bound established in Lemma 4.1 (for the case $n = \infty, m = 1$):

$$
\mathbb{E}_{p_1(\boldsymbol{x})}[\Delta(\boldsymbol{x})] \leq e^L \int_0^1 \mathbb{E}_{p_t(\boldsymbol{x})}[\epsilon(\boldsymbol{x}, t)^2]\mathrm{d}t,
\tag{26}
$$

where $\epsilon(\boldsymbol{x}, t) = s^{(t)}(\boldsymbol{x}, t) - s_{\boldsymbol{\theta}}^{(t)}(\boldsymbol{x}, t)$ is the error of the score model. The integral term is precisely the ideal, but intractable, time score matching loss $\mathcal{L}_{\mathrm{TSM}}(\boldsymbol{\theta})$ (see Eq. (56)). The key insight, as established in Choi et al. (2022), is that the ideal loss can be exactly decomposed into two terms:

$$
\int_0^1 \mathbb{E}_{p_t(\boldsymbol{x})}[\epsilon(\boldsymbol{x}, t)^2]\mathrm{d}t = \mathcal{L}_{\mathrm{TSM}}(\boldsymbol{\theta}) = \mathcal{L}_{\mathrm{STSM}}(\boldsymbol{\theta}) + \mathbb{E}_{p(t)}\mathbb{E}_{p_t(\boldsymbol{x})}\left|s^{(t)}(\boldsymbol{x}, t)\right|^2,
\tag{27}
$$

where $s^{(t)}(\boldsymbol{x}, t) \triangleq \partial_t \log p_t(\boldsymbol{x})$. For the second term, we can establish the variance identity:

$$
\begin{aligned}
\mathbb{E}_{p_t(\boldsymbol{x})} \left| s^{(t)}(\boldsymbol{x}, t) \right|^2 &= \mathrm{Var}_{p_t(\boldsymbol{x})} \left( s^{(t)}(\boldsymbol{x}, t) \right) + \left| \mathbb{E}_{p_t(\boldsymbol{x})} \left[ s^{(t)}(\boldsymbol{x}, t) \right] \right|^2 \\
&= \mathrm{Var}_{p_t(\boldsymbol{x})} \left( s^{(t)}(\boldsymbol{x}, t) \right) + \left| \int_{\mathbb{X}} p_t(\boldsymbol{x}) \partial_t \log p_t(\boldsymbol{x}) \mathrm{d}\boldsymbol{x} \right|^2 \\
&= \mathrm{Var}_{p_t(\boldsymbol{x})} \left( s^{(t)}(\boldsymbol{x}, t) \right) + \left| \int_{\mathbb{X}} \partial_t p_t(\boldsymbol{x}) \mathrm{d}\boldsymbol{x} \right|^2 \\
&= \mathrm{Var}_{p_t(\boldsymbol{x})} \left( s^{(t)}(\boldsymbol{x}, t) \right) + \left| \partial_t \int_{\mathbb{X}} p_t(\boldsymbol{x}) \mathrm{d}\boldsymbol{x} \right|^2 \\
&= \mathrm{Var}_{p_t(\boldsymbol{x})} \left( s^{(t)}(\boldsymbol{x}, t) \right),
\end{aligned}
\tag{28}
$$

since $\partial_t \int_{\mathbb{X}} p_t(\boldsymbol{x}) \mathrm{d}\boldsymbol{x} = \partial_t(1) = 0$. This term is the path variance, which is explicitly path-dependent as it is determined by the time score $s^{(t)}(\boldsymbol{x}, t)$.

Substituting Eqs. (27) and (28) into Eq. (26) completes the proof:

$$
\begin{aligned}
\mathbb{E}_{p_1(\boldsymbol{x})}[\Delta(\boldsymbol{x})] &\le e^L \int_0^1 \mathbb{E}_{p_t(\boldsymbol{x})}[\epsilon(\boldsymbol{x}, t)^2] \mathrm{d}t \\
&= e^L \left[ \mathcal{L}_{\mathrm{STSM}}(\boldsymbol{\theta}) + \mathbb{E}_{p(t)} \mathbb{E}_{p_t(\boldsymbol{x})} \left| s^{(t)}(\boldsymbol{x}, t) \right|^2 \right] \\
&= e^L \left[ \mathcal{L}_{\mathrm{STSM}}(\boldsymbol{\theta}) + \mathbb{E}_{p(t)} \left[ \mathrm{Var}_{p_t(\boldsymbol{x})} \left( s^{(t)}(\boldsymbol{x}, t) \right) \right] \right] \\
&= e^L \left[ \mathcal{L}_{\mathrm{STSM}}(\boldsymbol{\theta}) + \int_0^1 \mathrm{Var}_{p_t(\boldsymbol{x})} \left( s^{(t)}(\boldsymbol{x}, t) \right) \mathrm{d}t \right].
\end{aligned}
\tag{29}
$$

$\square$

### A.4 PROOF OF PROPOSITION 4.3

**Proposition 4.3.** *The path variance $\mathcal{V}$ for the DI and DDBI interpolants, which depends on the path schedules $(\alpha, \beta) = \{\alpha(t), \beta(t)\}_{t \in [0,1]}$, can be expressed in the following analytical forms:*

1. ***For the DI in Eq.* (2) *with a Gaussian prior $p_0(\boldsymbol{x}) = \mathcal{N}(\boldsymbol{x}; \boldsymbol{0}, \boldsymbol{I}_d)$:***

$$
\mathcal{V}_{DI}[\alpha, \beta] = \int_0^1 \left( \frac{2d\dot{\alpha}(t)^2}{\alpha(t)^2} + \frac{\dot{\beta}(t)^2}{\alpha(t)^2} \mathbb{E}_{p_1(\boldsymbol{x}_1)} \left[ \|\boldsymbol{x}_1\|^2 \right] \right) \mathrm{d}t.
\tag{11}
$$

2. ***For the DDBI in Eq.* (3) *with noise schedule $\sigma_t^2 = t(1-t)\gamma^2 + (\alpha(t)^2 + \beta(t)^2)\varepsilon$:***

$$
\mathcal{V}_{DDBI}[\alpha, \beta] = \int_0^1 \left( \frac{d}{2} \frac{(\dot{\sigma}_t^2)^2}{(\sigma_t^2)^2} + \frac{\mathbb{E}_{p_0(\boldsymbol{x}_0) p_1(\boldsymbol{x}_1)} \left\| \dot{\alpha}(t)\boldsymbol{x}_0 + \dot{\beta}(t)\boldsymbol{x}_1 \right\|^2}{\sigma_t^2} \right) \mathrm{d}t,
\tag{12}
$$

*Proof.* Let $s^{(t)}(\boldsymbol{x}, t) \triangleq \partial_t \log p_t(\boldsymbol{x})$ be the true time score. We provide the derivation for each case separately. The core strategy for both is to apply the law of total variance to decompose $\mathrm{Var}_{p_t(\boldsymbol{x})} (\partial_t \log p_t(\boldsymbol{x}))$ and then analyze each component.

**Case 1 (DI).** For the DI case, the interpolant is $\mathbf{x}_t = \alpha(t)\mathbf{x}_0 + \beta(t)\mathbf{x}_1$, where $\mathbf{x}_0 \sim p_0(\boldsymbol{x}) = \mathcal{N}(\boldsymbol{x}; \boldsymbol{0}, \boldsymbol{I}_d)$ and $\mathbf{x}_1 \sim p_1(\boldsymbol{x})$. The transition kernel for a given $\boldsymbol{x}_1$ is $p_t(\boldsymbol{x} \mid \boldsymbol{x}_1) = \mathcal{N}\left(\boldsymbol{x}; \beta(t)\boldsymbol{x}_1, \alpha(t)^2 \boldsymbol{I}_d\right)$.

By the law of total variance, we can decompose the variance term:

$$
\begin{aligned}
&\mathrm{Var}_{p_t(\boldsymbol{x})} (\partial_t \log p_t(\boldsymbol{x})) \\
&= \underbrace{\mathbb{E}_{p_1(\boldsymbol{x}_1)} \left[ \mathrm{Var}_{p_t(\boldsymbol{x}|\boldsymbol{x}_1)} (\partial_t \log p_t(\boldsymbol{x} \mid \boldsymbol{x}_1)) \right]}_{\text{Term (I)}} + \underbrace{\mathrm{Var}_{p_1(\boldsymbol{x}_1)} \left( \mathbb{E}_{p_t(\boldsymbol{x}|\boldsymbol{x}_1)} [\partial_t \log p_t(\boldsymbol{x} \mid \boldsymbol{x}_1)] \right)}_{\text{Term (II)}}.
\end{aligned}
\tag{30}
$$

We first compute the inner expectation required for Term (II). The conditional time score is:

$$
\begin{aligned}
\partial_t \log p_t(\boldsymbol{x} \mid \boldsymbol{x}_1) &= \partial_t \left[ -\frac{d}{2}\log(2\pi) - d\log\alpha(t) - \frac{\|\boldsymbol{x} - \beta(t)\boldsymbol{x}_1\|^2}{2\alpha(t)^2} \right] \\
&= -d\frac{\dot\alpha(t)}{\alpha(t)} + \frac{\dot\alpha(t)}{\alpha(t)^3}\|\boldsymbol{x} - \beta(t)\boldsymbol{x}_1\|^2 + \frac{\dot\beta(t)}{\alpha(t)^2}(\boldsymbol{x} - \beta(t)\boldsymbol{x}_1)^\top \boldsymbol{x}_1 \qquad (31) \\
&= -d\frac{\dot\alpha(t)}{\alpha(t)} + \frac{\dot\alpha(t)}{\alpha(t)^3}\|\boldsymbol{z}\|^2 + \frac{\dot\beta(t)}{\alpha(t)^2}\boldsymbol{z}^\top \boldsymbol{x}_1,
\end{aligned}
$$

where we have defined the centered sample $\boldsymbol{z} = \boldsymbol{x} - \beta(t)\boldsymbol{x}_1 \sim \mathcal{N}(\boldsymbol{z}; \boldsymbol{0}, \alpha(t)^2 \boldsymbol{I}_d)$.

Using the moments $\mathbb{E}[\boldsymbol{z}] = \boldsymbol{0}$ and $\mathbb{E}[\|\boldsymbol{z}\|^2] = d\alpha(t)^2$, the inner expectation simplifies to:

$$
\mathbb{E}_{p_t(\boldsymbol{x}|\boldsymbol{x}_1)}\left[\partial_t \log p_t(\boldsymbol{x} \mid \boldsymbol{x}_1)\right] = -d\frac{\dot\alpha(t)}{\alpha(t)} + \frac{\dot\alpha(t)}{\alpha(t)^3}\left(d\alpha(t)^2\right) + \frac{\dot\beta(t)}{\alpha(t)^2}\mathbb{E}[\boldsymbol{z}]^\top \boldsymbol{x}_1 = 0. \qquad (32)
$$

Since the conditional expectation is zero for any $\boldsymbol{x}_1$, its variance, Term (II), is also zero. *The total variance thus simplifies entirely to Term (I).* Furthermore, because the conditional mean of the time score is zero, its conditional variance equals its conditional second moment:

$$
\begin{aligned}
\mathrm{Var}_{p_t(\boldsymbol{x}|\boldsymbol{x}_1)}\left(\partial_t \log p_t(\boldsymbol{x} \mid \boldsymbol{x}_1)\right) &= \mathbb{E}_{p_t(\boldsymbol{x}|\boldsymbol{x}_1)}\left[\left(\partial_t \log p_t(\boldsymbol{x} \mid \boldsymbol{x}_1)\right)^2\right] \\
&= \mathbb{E}_{p_t(\boldsymbol{x}|\boldsymbol{x}_1)}\left[\left(\frac{\dot\alpha(t)}{\alpha(t)^3}\|\boldsymbol{z}\|^2 - d\frac{\dot\alpha(t)}{\alpha(t)} + \frac{\dot\beta(t)}{\alpha(t)^2}\boldsymbol{z}^\top \boldsymbol{x}_1\right)^2\right] \\
&= \left(\frac{\dot\alpha(t)}{\alpha(t)^3}\right)^2 \mathbb{E}[\|\boldsymbol{z}\|^4] + \left(d\frac{\dot\alpha(t)}{\alpha(t)}\right)^2 - 2d\frac{\dot\alpha(t)^2}{\alpha(t)^4}\mathbb{E}[\|\boldsymbol{z}\|^2] + \left(\frac{\dot\beta(t)}{\alpha(t)^2}\right)^2 \mathbb{E}[(\boldsymbol{z}^\top \boldsymbol{x}_1)^2] \\
&= \frac{\dot\alpha(t)^2}{\alpha(t)^6}d(d+2)\alpha(t)^4 + d^2\frac{\dot\alpha(t)^2}{\alpha(t)^2} - 2d\frac{\dot\alpha(t)^2}{\alpha(t)^4}(d\alpha(t)^2) + \frac{\dot\beta(t)^2}{\alpha(t)^4}(\alpha(t)^2\|\boldsymbol{x}_1\|^2) \\
&= \frac{\dot\alpha(t)^2}{\alpha(t)^2}\left(d(d+2) + d^2 - 2d^2\right) + \frac{\dot\beta(t)^2}{\alpha(t)^2}\|\boldsymbol{x}_1\|^2 \\
&= \frac{2d\dot\alpha(t)^2}{\alpha(t)^2} + \frac{\dot\beta(t)^2}{\alpha(t)^2}\|\boldsymbol{x}_1\|^2.
\end{aligned} \qquad (33)
$$

Here, we use the standard Gaussian moments: $\mathbb{E}[\|\boldsymbol{z}\|^4] = d(d+2)\alpha(t)^4$, $\mathbb{E}[(\boldsymbol{z}^\top \boldsymbol{x}_1)^2] = \alpha(t)^2\|\boldsymbol{x}_1\|^2$, and that all cross-terms involving odd moments of $\boldsymbol{z}$ are zero.

Finally, substituting this result into (I) and taking the expectation over $p_1(\boldsymbol{x}_1)$ yields the total variance:

$$
\begin{aligned}
\mathrm{Var}_{p_t(\boldsymbol{x})}\left(\partial_t \log p_t(\boldsymbol{x})\right) &= \mathbb{E}_{p_1(\boldsymbol{x}_1)}\left[\mathrm{Var}_{p_t(\boldsymbol{x}|\boldsymbol{x}_1)}\left(\partial_t \log p_t(\boldsymbol{x} \mid \boldsymbol{x}_1)\right)\right] \\
&= \mathbb{E}_{p_1(\boldsymbol{x}_1)}\left[\frac{2d\dot\alpha_t^2}{\alpha_t^2} + \frac{\dot\beta_t^2}{\alpha_t^2}\|\boldsymbol{x}_1\|^2\right] \\
&= \frac{2d\dot\alpha_t^2}{\alpha_t^2} + \frac{\dot\beta_t^2}{\alpha_t^2}\mathbb{E}_{p_1(\boldsymbol{x}_1)}\left[\|\boldsymbol{x}_1\|^2\right].
\end{aligned} \qquad (34)
$$

**Case 2 (DDBI).** For the DDBI case, the interpolant is $\mathbf{x}_t = \alpha(t)\mathbf{x}_0 + \beta(t)\mathbf{x}_1 + \sigma_t\mathbf{z}$, with $\mathbf{x}_0 \sim p_0$, $\mathbf{x}_1 \sim p_1$, and noise schedule $\sigma_t^2 = t(1-t)\gamma^2 + (\alpha(t)^2 + \beta(t)^2)\varepsilon$. The transition kernel conditioned on both endpoints is $p_t(\boldsymbol{x} \mid \boldsymbol{x}_0, \boldsymbol{x}_1) = \mathcal{N}\left(\boldsymbol{x}; \boldsymbol{\mu}_t, \sigma_t^2 \boldsymbol{I}_d\right)$, where $\boldsymbol{\mu}_t = \alpha(t)\boldsymbol{x}_0 + \beta(t)\boldsymbol{x}_1$.

The law of total variance is now applied by conditioning on the pair $(\boldsymbol{x}_0, \boldsymbol{x}_1)$:

$$
\begin{aligned}
\mathrm{Var}_{p_t(\boldsymbol{x})}\left(\partial_t \log p_t(\boldsymbol{x})\right) &= \underbrace{\mathbb{E}_{p_0(\boldsymbol{x}_0)p_1(\boldsymbol{x}_1)}\left[\mathrm{Var}_{p_t(\boldsymbol{x}|\boldsymbol{x}_0,\boldsymbol{x}_1)}\left(\partial_t \log p_t(\boldsymbol{x} \mid \boldsymbol{x}_0, \boldsymbol{x}_1)\right)\right]}_{\text{Term (I)}} \\
&\quad + \underbrace{\mathrm{Var}_{p_0(\boldsymbol{x}_0)p_1(\boldsymbol{x}_1)}\left(\mathbb{E}_{p_t(\boldsymbol{x}|\boldsymbol{x}_0,\boldsymbol{x}_1)}\left[\partial_t \log p_t(\boldsymbol{x} \mid \boldsymbol{x}_0, \boldsymbol{x}_1)\right]\right)}_{\text{Term (II)}}.
\end{aligned} \qquad (35)
$$

Let $\boldsymbol{\mu}_t = \alpha(t)\boldsymbol{x}_0 + \beta(t)\boldsymbol{x}_1$ and $\dot{\boldsymbol{\mu}}_t = \dot{\alpha}(t)\boldsymbol{x}_0 + \dot{\beta}(t)\boldsymbol{x}_1$. The conditional time score is:

$$
\begin{aligned}
\partial_t \log p_t(\boldsymbol{x} \mid \boldsymbol{x}_0, \boldsymbol{x}_1) &= \partial_t \left[ -\frac{d}{2}\log(2\pi\sigma_t^2) - \frac{\|\boldsymbol{x} - \boldsymbol{\mu}_t\|^2}{2\sigma_t^2} \right] \\
&= -\frac{d}{2}\frac{\dot{\sigma}_t^2}{\sigma_t^2} + \frac{\|\boldsymbol{x} - \boldsymbol{\mu}_t\|^2}{2(\sigma_t^2)^2}(\dot{\sigma}_t^2) + \frac{(\boldsymbol{x} - \boldsymbol{\mu}_t)^\top \dot{\boldsymbol{\mu}}_t}{\sigma_t^2}.
\end{aligned}
\tag{36}
$$

Letting $\boldsymbol{z} = \boldsymbol{x} - \boldsymbol{\mu}_t \sim \mathcal{N}(\boldsymbol{0}, \sigma_t^2 \boldsymbol{I}_d)$ with $\mathbb{E}[\|\boldsymbol{z}\|^2] = d\sigma_t^2$, we take the conditional expectation:

$$
\begin{aligned}
\mathbb{E}_{p_t(\boldsymbol{x}|\boldsymbol{x}_0,\boldsymbol{x}_1)}\left[\partial_t \log p_t(\boldsymbol{x} \mid \boldsymbol{x}_0, \boldsymbol{x}_1)\right] &= -\frac{d}{2}\frac{\dot{\sigma}_t^2}{\sigma_t^2} + \frac{\mathbb{E}[\|\boldsymbol{z}\|^2]}{2(\sigma_t^2)^2}(\dot{\sigma}_t^2) + \frac{\mathbb{E}[\boldsymbol{z}]^\top \dot{\boldsymbol{\mu}}_t}{\sigma_t^2} \\
&= -\frac{d}{2}\frac{\dot{\sigma}_t^2}{\sigma_t^2} + \frac{d\sigma_t^2}{2(\sigma_t^2)^2}(\dot{\sigma}_t^2) = 0.
\end{aligned}
\tag{37}
$$

As in the DI case, the conditional expectation is zero, and thus Term (II) is zero and the total variance reduces to the Term I. Similarly, the conditional variance equals the conditional second moment:

$$
\begin{aligned}
\mathrm{Var}_{p_t(\boldsymbol{x}|\boldsymbol{x}_0,\boldsymbol{x}_1)} \left(\partial_t \log p_t(\boldsymbol{x} \mid \boldsymbol{x}_0, \boldsymbol{x}_1)\right) &= \mathbb{E}_{p_t(\boldsymbol{x}|\boldsymbol{x}_0,\boldsymbol{x}_1)}\left[ (\partial_t \log p_t(\boldsymbol{x} \mid \boldsymbol{x}_0, \boldsymbol{x}_1))^2 \right] \\
&= \left(-\frac{d}{2}\frac{\dot{\sigma}_t^2}{\sigma_t^2}\right)^2 + \mathbb{E}\left[\left(\frac{\|\boldsymbol{z}\|^2}{2(\sigma_t^2)^2}(\dot{\sigma}_t^2)\right)^2\right] + \mathbb{E}\left[\left(\frac{\boldsymbol{z}^\top \dot{\boldsymbol{\mu}}_t}{\sigma_t^2}\right)^2\right] - \frac{d^2}{2}\frac{(\dot{\sigma}_t^2)^2}{(\sigma_t^2)^2} \quad (\star) \\
&= \frac{d^2}{4}\frac{(\dot{\sigma}_t^2)^2}{(\sigma_t^2)^2} - \frac{d^2}{2}\frac{(\dot{\sigma}_t^2)^2}{(\sigma_t^2)^2} + \frac{(\dot{\sigma}_t^2)^2}{4(\sigma_t^2)^4}\mathbb{E}[\|\boldsymbol{z}\|^4] + \frac{1}{(\sigma_t^2)^2}\mathbb{E}[(\boldsymbol{z}^\top \dot{\boldsymbol{\mu}}_t)^2] \\
&= -\frac{d^2}{4}\frac{(\dot{\sigma}_t^2)^2}{(\sigma_t^2)^2} + \frac{(\dot{\sigma}_t^2)^2 d(d+2)(\sigma_t^2)^2}{4(\sigma_t^2)^4} + \frac{\|\dot{\boldsymbol{\mu}}_t\|^2\sigma_t^2}{(\sigma_t^2)^2} \quad (\star\star) \\
&= \left(-\frac{d^2}{4} + \frac{d(d+2)}{4}\right)\frac{(\dot{\sigma}_t^2)^2}{(\sigma_t^2)^2} + \frac{\|\dot{\boldsymbol{\mu}}_t\|^2}{\sigma_t^2} \\
&= \frac{(-d^2 + d^2 + 2d)}{4}\frac{(\dot{\sigma}_t^2)^2}{(\sigma_t^2)^2} + \frac{\|\dot{\boldsymbol{\mu}}_t\|^2}{\sigma_t^2} \\
&= \frac{d}{2}\frac{(\dot{\sigma}_t^2)^2}{(\sigma_t^2)^2} + \frac{\|\dot{\alpha}(t)\boldsymbol{x}_0 + \dot{\beta}(t)\boldsymbol{x}_1\|^2}{\sigma_t^2},
\end{aligned}
\tag{38}
$$

where $(\star)$ holds because the cross-terms involving odd moments of the centered Gaussian $\boldsymbol{z}$ are zero and the rest term satisfies $\mathbb{E}[2AB] = 2A \cdot \mathbb{E}[B] = \frac{d^2}{2}\frac{(\dot{\sigma}_t^2)^2}{(\sigma_t^2)^2}$ with $A = \frac{d}{2}\frac{\dot{\sigma}_t^2}{\sigma_t^2}, B = \frac{\|\boldsymbol{z}\|^2(\dot{\sigma}_t^2)}{2(\sigma_t^2)^2}$. $(\star\star)$ holds with the moments $\mathbb{E}[\|\boldsymbol{z}\|^4] = d(d+2)(\sigma_t^2)^2$ and the quadratic form $\mathbb{E}[(\boldsymbol{z}^\top \dot{\boldsymbol{\mu}}_t)^2] = \sigma_t^2\|\dot{\boldsymbol{\mu}}_t\|^2$.

Taking the final expectation over $p_0(\boldsymbol{x}_0)$ and $p_1(\boldsymbol{x}_1)$ gives the total variance:

$$
\mathrm{Var}_{p_t(\boldsymbol{x})}\left(\partial_t \log p_t(\boldsymbol{x})\right) = \frac{d}{2}\frac{(\dot{\sigma}_t^2)^2}{(\sigma_t^2)^2} + \frac{\mathbb{E}_{p_0,p_1}\left\|\dot{\alpha}(t)\mathbf{x}_0 + \dot{\beta}(t)\mathbf{x}_1\right\|^2}{\sigma_t^2}.
\tag{39}
$$

This completes the proof for the DDBI case. $\qquad\square$

## A.5 CONTROL OF THE LIPSCHITZ CONSTANT VIA MINIMUM VARIANCE PATH PRINCIPLE

*Remark* A.3. The connection between path variance $\mathcal{V}[\alpha, \beta]$ and the Lipschitz constant $L$ in Assumption A.1 is primarily *empirical* and *heuristic* in nature, which is why we present it as a remark rather than a formal proposition. In practice, minimizing $\mathcal{V}$ tends to produce smoother paths, and empirically correlates with reduced values of $L$.

*Argument.* Our heuristic is motivated by the observation that both $\mathcal{V}$ and the time score magnitude depend on the path derivatives $\dot{\alpha}(t)$ and $\dot{\beta}(t)$. Specifically, the closed-form expressions for $\mathcal{V}$ in

Proposition 4.3 are explicit functionals of squared path velocities (e.g., $\dot{\alpha}(t)^2$, $\dot{\beta}(t)^2$, or $\|\dot{\alpha}(t)\boldsymbol{x}_0 + \dot{\beta}(t)\boldsymbol{x}_1\|^2$). Thus, minimizing $\mathcal{V}$ discourages paths with large instantaneous velocities.

Because our KMM parameterization yields smooth, bounded paths (see Eq. (14)), extreme velocity spikes are structurally unlikely. In such a regular function class, minimizing the integrated squared velocity $\mathcal{V}$ *approximately suppresses* the maximum path velocity, which in turn appears to help keep $L$ from becoming excessively large in practice (see Assumption A.1 for detailed derivations).

This behavior is consistently observed in our experiments: the optimized paths (see Figs. 5 to 8 in Sec. B.6) are visibly smooth. We emphasize that this is a *phenomenological observation*, not a theoretical guarantee, and further analysis of the precise relationship between path geometry and $L$ remains an open question. $\qquad\square$

### A.6 PROOF OF THE EQUIVALENCE OF TIME SCORE MATCHING OBJECTIVES

**Proposition A.4.** *Let $p_t$ be the marginal distribution induced by the path schedule $\{\alpha(t), \beta(t)\}_{t \in [0,1]}$. For any fixed schedule, all three objectives share the same optimum: $s_{\boldsymbol{\theta}^\star}^{(t)}(\boldsymbol{x}, t) = \partial_t \log p_t(\boldsymbol{x})$.*

- *Time Score Matching (TSM):*

$$\mathcal{L}_{\text{TSM}}(\boldsymbol{\theta}) = \mathbb{E}_{p(t)p_t(\boldsymbol{x}_t)} \left| \partial_t \log p_t(\boldsymbol{x}_t) - s_{\boldsymbol{\theta}}^{(t)}(\boldsymbol{x}_t, t) \right|^2. \tag{40}$$

- *Conditional Time Score Matching (CTSM):*

$$\mathcal{L}_{\text{CTSM}}(\boldsymbol{\theta}) = \mathbb{E}_{p(t)p_t(\boldsymbol{x}_t)p(\boldsymbol{y})} \left| \partial_t \log p_t(\boldsymbol{x}_t \mid \boldsymbol{y}) - s_{\boldsymbol{\theta}}^{(t)}(\boldsymbol{x}_t, t) \right|^2, \tag{41}$$

*where $\boldsymbol{y}$ is an instance of the conditioning variable $\mathbf{y}$ ($\mathbf{y} = \mathbf{x}_1$ for the DI, $\mathbf{y} = (\mathbf{x}_0, \mathbf{x}_1)$ for the DDBI).*

- *Sliced Time Score Matching (STSM):*

$$\mathcal{L}_{\text{STSM}}(\boldsymbol{\theta}) = \mathbb{E}_{p(t)p_t(\boldsymbol{x})} \left[ \partial_t s_{\boldsymbol{\theta}}^{(t)}(\boldsymbol{x}, t) + \frac{1}{2} s_{\boldsymbol{\theta}}^{(t)}(\boldsymbol{x}, t)^2 \right] + \mathcal{L}_{\text{boundary}}, \tag{42}$$

*where $\mathcal{L}_{\text{boundary}} = \mathbb{E}_{p_0(\boldsymbol{x}_0)p_1(\boldsymbol{x}_1)} \left[ s_{\boldsymbol{\theta}}^{(t)}(\boldsymbol{x}_0, 0) - s_{\boldsymbol{\theta}}^{(t)}(\boldsymbol{x}_1, 1) \right].$*

*Proof.* We prove that all three objectives are minimized when $s_{\boldsymbol{\theta}}^{(t)}(\boldsymbol{x}_t, t) = \partial_t \log p_t(\boldsymbol{x}_t)$ by showing their differences are constants independent of $\boldsymbol{\theta}$.

**Part 1: Relating CTSM and TSM.** We begin with the CTSM objective. For fixed $t$, the inner expectation in CTSM can be expanded using the identity $\mathbb{E}[\boldsymbol{z}^2] = \text{Var}(\boldsymbol{z}) + \mathbb{E}[\boldsymbol{z}]^2$:

$$\mathcal{L}_{\text{CTSM}}(\boldsymbol{\theta}) = \mathbb{E}_{p(t)} \mathbb{E}_{p_t(\boldsymbol{x}_t)} \left[ \mathbb{E}_{p(\boldsymbol{y})} \left| \partial_t \log p_t(\boldsymbol{x}_t | \boldsymbol{y}) - s_{\boldsymbol{\theta}}^{(t)}(\boldsymbol{x}_t, t) \right|^2 \right]. \tag{43}$$

For a given $t$ and sample $\boldsymbol{x}_t$, we analyze the inner expectation over $\boldsymbol{y}$:

$$\begin{aligned}
&\mathbb{E}_{p(\boldsymbol{y})} \left[ \left| \partial_t \log p_t(\boldsymbol{x}_t \mid \boldsymbol{y}) - s_{\boldsymbol{\theta}}^{(t)}(\boldsymbol{x}_t, t) \right|^2 \right] \\
&= \mathbb{E}_{p(\boldsymbol{y})} \left[ \left| \partial_t \log p_t(\boldsymbol{x}_t \mid \boldsymbol{y}) - \mu(\boldsymbol{x}_t, t) + \mu(\boldsymbol{x}_t, t) - s_{\boldsymbol{\theta}}^{(t)}(\boldsymbol{x}_t, t) \right|^2 \right] \quad (\star) \\
&= \mathbb{E}_{p(\boldsymbol{y})} \left[ |\partial_t \log p_t(\boldsymbol{x}_t \mid \boldsymbol{y}) - \mu(\boldsymbol{x}_t, t)\|^2 \right] + \left\| \mu(\boldsymbol{x}_t, t) - s_{\boldsymbol{\theta}}^{(t)}(\boldsymbol{x}_t, t) \right\|^2 \\
&= \text{Var}_{p(\boldsymbol{y})} \left( \partial_t \log p_t(\boldsymbol{x}_t \mid \boldsymbol{y}) \right) + \left| \mu(\boldsymbol{x}_t, t) - s_{\boldsymbol{\theta}}^{(t)}(\boldsymbol{x}_t, t) \right|^2,
\end{aligned} \tag{44}$$

where $(\star)$ holds under $\mu(\boldsymbol{x}_t, t) = \mathbb{E}_{p(\boldsymbol{y})}[\partial_t \log p_t(\boldsymbol{x}_t \mid \boldsymbol{y})]$. The key insight is that the conditional expectation $\mu(\boldsymbol{x}_t, t)$ yields the marginal score:

$$\mu(\boldsymbol{x}_t, t) = \mathbb{E}_{p(\boldsymbol{y})} \left[ \partial_t \log p_t(\boldsymbol{x}_t \mid \boldsymbol{y}) \right] = \partial_t \log p_t(\boldsymbol{x}_t). \tag{45}$$

Substituting this back yields the final relationship:

$$\mathcal{L}_{\text{CTSM}}(\boldsymbol{\theta}) = \underbrace{\mathbb{E}_{p(t)p_t(\boldsymbol{x}_t)} \left| \partial_t \log p_t(\boldsymbol{x}_t) - s_{\boldsymbol{\theta}}^{(t)}(\boldsymbol{x}_t, t) \right|^2}_{\mathcal{L}_{\text{TSM}}(\boldsymbol{\theta})} + \underbrace{\mathbb{E}_{p(t)p_t(\boldsymbol{x}_t)} \left[ \text{Var}_{p(\boldsymbol{y})} \left( \partial_t \log p_t(\boldsymbol{x}_t \mid \boldsymbol{y}) \right) \right]}_{V_{\text{cond}}}.$$

$$(46)$$

Since $V_{\text{cond}}$ is independent of $\boldsymbol{\theta}$, both objectives share the same minimizer.

**Part 2: Relating STSM and TSM.** The STSM objective is derived via integration by parts from the TSM objective. The standard derivation shows that (Choi et al., 2022):

$$\mathcal{L}_{\text{TSM}}(\boldsymbol{\theta}) = \mathcal{L}_{\text{STSM}}(\boldsymbol{\theta}) + \underbrace{\mathbb{E}_{p(t)} \mathbb{E}_{p_t(\boldsymbol{x}_t)} \left| \partial_t \log p_t(\boldsymbol{x}_t) \right|^2}_{V_{\text{marg}}}. \tag{47}$$

Since $V_{\text{marg}}$ is independent of $\boldsymbol{\theta}$ and is a constant for a given path schedule, the optimum is shared.

Finally, since $\mathcal{L}_{\text{CTSM}}(\boldsymbol{\theta}) = \mathcal{L}_{\text{TSM}}(\boldsymbol{\theta}) + \text{const}$ and $\mathcal{L}_{\text{STSM}}(\boldsymbol{\theta}) = \mathcal{L}_{\text{TSM}}(\boldsymbol{\theta}) + \text{const}$, all three objectives are minimized by the same model $s_{\boldsymbol{\theta}^*}^{(t)}(\boldsymbol{x}_t, t) = \partial_t \log p_t(\boldsymbol{x}_t)$. $\qquad\square$

# B EXPERIMENTAL SETTINGS AND MORE RESULTS

## B.1 FORMULATIONS AND VARIANCES OF COMMON PATH SCHEDULES

This section provides the precise mathematical formulations for the common, fixed probability path schedules $(\alpha(t), \beta(t))$ and their time derivatives $(\dot{\alpha}(t), \dot{\beta}(t))$ used in our comparative experiments, alongside their corresponding analytical variance expressions derived from Proposition 4.3. For brevity in the variance expressions, we define the data-dependent constants $C_0 = \mathbb{E}_{p_0(\boldsymbol{x}_0)}[\|\boldsymbol{x}_0\|^2]$, $C_1 = \mathbb{E}_{p_1(\boldsymbol{x}_1)}[\|\boldsymbol{x}_1\|^2]$, and $C_{01} = \mathbb{E}_{p_0, p_1}[\boldsymbol{x}_0^\top \boldsymbol{x}_1]$. The summary of path characteristics and associated variance properties is presented in Tab. 5.

Table 5: Summary of path characteristics and associated variance properties.

| Path Schedule | Constraint | $\mathcal{V}_{\text{DI}}[\alpha, \beta]$ | $\mathcal{V}_{\text{DDBI}}[\alpha, \beta]$ | Key Observation |
|---|---|---|---|---|
| Linear | $\alpha(t) + \beta(t) = 1$ | Divergent | Finite | DI is ill-posed; DDBI is stable. |
| VP | $\alpha(t)^2 + \beta(t)^2 = 1$ | Divergent | Finite | DI diverges as $t \to 1$. |
| Cosine | $\alpha(t)^2 + \beta(t)^2 = 1$ | Divergent | Finite | Smoother path, but DI still diverges. |
| Föllmer | $\alpha(t)^2 + \beta(t)^2 = 1$ | Divergent | Finite | Severe singularity in $\dot{\alpha}(t)$ at $t = 1$. |
| Trigonometric | $\alpha(t)^2 + \beta(t)^2 = 1$ | Divergent | Finite | Singularity from $\tan^2$ term at $t = 1$. |

**Linear Path.** The linear interpolant, a common baseline in stochastic interpolant literature (Albergo et al., 2023), defines a direct affine path satisfying $\alpha(t) + \beta(t) = 1$.

$$\begin{aligned} \alpha(t) &= 1 - t & \beta(t) &= t, \\ \dot{\alpha}(t) &= -1 & \dot{\beta}(t) &= 1. \end{aligned} \tag{48}$$

The corresponding DI variance integral, $\mathcal{V}_{\text{DI}}[\alpha, \beta] = \int_0^1 (2d + C_1)(1-t)^{-2} dt$, diverges due to the singularity at $t = 1$. For the DDBI case, the variance is given by:

$$\mathcal{V}_{\text{DDBI}}[\alpha, \beta] = \int_0^1 \left( \frac{d}{2} \frac{(\dot{\sigma}_t^2)^2}{(\sigma_t^2)^2} + \frac{C_0 + C_1 - 2C_{01}}{\sigma_t^2} \right) dt, \tag{49}$$

where the noise schedule is $\sigma_t^2 = t(1-t)\gamma^2 + ((1-t)^2 + t^2)\varepsilon$.

**VP Path.** This path is inspired by the variance-preserving (VP) SDE schedule (Ho et al., 2020; Song et al., 2021b), satisfying the spherical constraint $\alpha(t)^2 + \beta(t)^2 = 1$. We use the standard diffusion constants $\beta_0 = 0.1$ and $\beta_1 = 20$.

$$\alpha(t) = \exp\left(-0.25t^2(\beta_1 - \beta_0) - 0.5t\beta_0\right) \qquad \beta(t) = \sqrt{1 - \alpha(t)^2},$$
$$\dot{\alpha}(t) = -0.5\left((\beta_1 - \beta_0)t + \beta_0\right)\alpha(t) \qquad \dot{\beta}(t) = \frac{\alpha(t)^2}{\beta(t)}\left(0.5((\beta_1 - \beta_0)t + \beta_0)\right). \tag{50}$$

Due to the term $1/\alpha(t)^2$ which diverges as $t \to 1$, the DI variance is infinite. The DDBI variance is finite and given by:

$$\mathcal{V}_{\text{DDBI}}[\alpha, \beta] = \int_0^1 \left(\frac{d}{2}\frac{\gamma^4(1-2t)^2}{(\sigma_t^2)^2} + \frac{\dot{\alpha}(t)^2 C_0 + \dot{\beta}(t)^2 C_1 + 2\dot{\alpha}(t)\dot{\beta}(t)C_{01}}{\sigma_t^2}\right) dt, \tag{51}$$

where the spherical constraint simplifies the noise schedule to $\sigma_t^2 = t(1-t)\gamma^2 + \varepsilon$.

**Cosine Path.** The cosine schedule (Nichol & Dhariwal, 2021) also satisfies the spherical constraint. It is defined via $\bar{\alpha}(t) = \cos\left(\frac{t+s}{1+s}\frac{\pi}{2}\right) / \cos\left(\frac{s}{1+s}\frac{\pi}{2}\right)$, with $s = 0.008$.

$$\alpha(t) = \sqrt{\bar{\alpha}(t)} \qquad \beta(t) = \sqrt{1 - \bar{\alpha}(t)},$$
$$\dot{\alpha}(t) = \frac{\dot{\bar{\alpha}}(t)}{2\alpha(t)} \qquad \dot{\beta}(t) = -\frac{\dot{\bar{\alpha}}(t)}{2\beta(t)}. \tag{52}$$

The DI variance for this path is divergent. The DDBI variance takes the same functional form as the VP case, substituting the appropriate derivatives.

**Föllmer Path.** This path is derived from Föllmer's construction of a Brownian bridge (Föllmer, 2005; Chen et al., 2024) and satisfies the spherical constraint.

$$\alpha(t) = \sqrt{1 - t^2} \qquad \beta(t) = t,$$
$$\dot{\alpha}(t) = -\frac{t}{\alpha(t)} \qquad \dot{\beta}(t) = 1. \tag{53}$$

The DI variance, $\mathcal{V}_{\text{DI}}[\alpha, \beta] = \int_0^1 \left(\frac{2dt^2}{(1-t^2)^2} + \frac{C_1}{1-t^2}\right) dt$, diverges severely at $t = 1$. The DDBI variance is given by:

$$\mathcal{V}_{\text{DDBI}}[\alpha, \beta] = \int_0^1 \left(\frac{d}{2}\frac{\gamma^4(1-2t)^2}{(\sigma_t^2)^2} + \frac{\frac{t^2}{1-t^2}C_0 + C_1 - \frac{2t}{\sqrt{1-t^2}}C_{01}}{\sigma_t^2}\right) dt. \tag{54}$$

**Trigonometric Path.** This path follows a circular trajectory in the $(\alpha, \beta)$ space (Albergo et al., 2023), satisfying the spherical constraint $\alpha(t)^2 + \beta(t)^2 = 1$.

$$\alpha(t) = \cos\left(\frac{\pi}{2}t\right) \qquad \beta(t) = \sin\left(\frac{\pi}{2}t\right),$$
$$\dot{\alpha}(t) = -\frac{\pi}{2}\beta(t) \qquad \dot{\beta}(t) = \frac{\pi}{2}\alpha(t). \tag{55}$$

The DI variance, $\mathcal{V}_{\text{DI}}[\alpha, \beta] = (\pi/2)^2 \int_0^1 (2d\tan^2(\frac{\pi}{2}t) + C_1)dt$, diverges due to the tangent term at $t = 1$. The DDBI variance takes the same functional form as the VP case.

### B.2 EXPERIMENTAL SETTINGS FOR OBJECTIVES

Our experimental framework follows the theoretical analysis in Sec. 4.1. For each interpolant class (DI and DDBI), we have two primary objectives: 1) the minimization of the path variance $\mathcal{V}[\alpha_\phi, \beta_\phi]$ to learn an optimal path schedule, and 2) the minimization of a joint score matching loss to train the joint score model $s_\theta^{(\boldsymbol{x},t)}(\boldsymbol{x}, t) \triangleq [s_\theta^{(\boldsymbol{x})}(\boldsymbol{x}, t), s_\theta^{(t)}(\boldsymbol{x}, t)]$. Here, $s_\theta^{(\boldsymbol{x})}$ and $s_\theta^{(t)}$ are the data and time score models, aiming to approximate the data score $s^{(\boldsymbol{x})}(\boldsymbol{x}, t) \triangleq \partial_{\boldsymbol{x}} \log p_t(\boldsymbol{x})$ and time score $s^{(t)}(\boldsymbol{x}, t) \triangleq \partial_t \log p_t(\boldsymbol{x})$. The superscripts $(t)$ and $(\boldsymbol{x})$ are labels indicating "time" and "data", not derivatives. We use these notations consistently throughout the paper. This section provides the precise mathematical formulations for these objectives as implemented in our experiments.

### B.2.1 COMMONLY USED OBJECTIVES

In this section, we introduce three objectives for training the time score model.

**Time Score Matching (TSM).** To approximate the true marginal time score $s^{(t)}(\boldsymbol{x}, t)$, the time score model $s_{\boldsymbol{\theta}}^{(t)}(\boldsymbol{x}, t)$ is trained by minimizing the ideal, but intractable, time score-matching loss:

$$\mathcal{L}_{\text{TSM}}(\boldsymbol{\theta}) = \mathbb{E}_{p(t)p_t(\boldsymbol{x})} \left| s^{(t)}(\boldsymbol{x}, t) - s_{\boldsymbol{\theta}}^{(t)}(\boldsymbol{x}, t) \right|^2. \tag{56}$$

**Sliced Time Score Matching (STSM).** Since the target score $s^{(t)}(\boldsymbol{x}, t)$ is unknown, this objective cannot be optimized directly. A standard technique, as shown in Choi et al. (2022), is to apply integration by parts to derive a tractable objective. This equivalent training loss, which is termed as the sliced time score matching (STSM), is given by:

$$\mathcal{L}_{\text{STSM}}(\boldsymbol{\theta}) = 2\mathbb{E}_{p_0(\boldsymbol{x}_0)p_1(\boldsymbol{x}_1)} \left[ s_{\boldsymbol{\theta}}^{(t)}(\boldsymbol{x}_0, 0) - s_{\boldsymbol{\theta}}^{(t)}(\boldsymbol{x}_1, 1) \right] + \mathbb{E}_{p(t)p_t(\boldsymbol{x})} \left[ 2\partial_t s_{\boldsymbol{\theta}}^{(t)}(\boldsymbol{x}, t) + s_{\boldsymbol{\theta}}^{(t)}(\boldsymbol{x}, t)^2 \right], \tag{57}$$

where the first term denote the boundary conditions of $s_{\boldsymbol{\theta}}^{(t)}$, $\partial_t s_{\boldsymbol{\theta}}^{(t)}$ is the time derivative of $s_{\boldsymbol{\theta}}^{(t)}$, and $p(t)$ is a timestep distribution over $(0, 1)$ (see Sec. B.2.5 for details).

**Conditional Time Score Matching (CTSM).** In the limit of infinite data and model capacity, the time score model $s_{\boldsymbol{\theta}}^{(t)}$ exactly recovers the time score $s^{(t)}$ by minimizing the conditional time score matching (CTSM) objective (Yu et al., 2025):

$$\mathcal{L}_{\text{CTSM}}(\boldsymbol{\theta}) = \mathbb{E}_{p(t)p_t(\boldsymbol{x}_t)p(\boldsymbol{y})} \left| \partial_t \log p_t(\boldsymbol{x}_t \mid \boldsymbol{y}) - s_{\boldsymbol{\theta}}^{(t)}(\boldsymbol{x}_t, t) \right|^2, \tag{58}$$

where $p(t)$ is a timestep distribution (see Sec. B.2.5 for details), $\boldsymbol{x}_t$ is induced from DI (see Eq. (2)) or DDBI (see Eq. (3)), and $\boldsymbol{y}$ is an instance of the conditioning variable $\mathbf{y}$ ($\mathbf{y} \triangleq \mathbf{x}_1$ for DI, $\mathbf{y} \triangleq (\mathbf{x}_0, \mathbf{x}_1)$ for DDBI).

**We Use CTSM.** As shown in Proposition A.4, all these three objectives are equivalent, guaranteeing that the learned model satisfies $s_{\boldsymbol{\theta}^\star}^{(t)}(\boldsymbol{x}, t) = \partial_t \log p_t(\boldsymbol{x})$. Hence, we adopt CTSM for training since (i) the conditional score is available in closed form, (ii) no derivatives are required, and (iii) the score model is evaluated only once per batch.

### B.2.2 TOTAL OBJECTIVE FOR DRE WITH THE DI

**Path Variance Objective.** For the DI setting, we learn the path parameters $\phi$ of our KMM-based schedule by minimizing the analytical variance given in Proposition 4.3:

$$\min_{\phi} \quad \mathcal{V}_{\text{DI}} [\alpha_{\phi}, \beta_{\phi}] = \int_0^1 \left( \frac{2d\dot{\alpha}_{\phi}(t)^2}{\alpha_{\phi}(t)^2} + \frac{\dot{\beta}_{\phi}(t)^2}{\alpha_{\phi}(t)^2} \mathbb{E}_{p_1(\boldsymbol{x}_1)} \left[ \|\boldsymbol{x}_1\|^2 \right] \right) \mathrm{d}t. \tag{59}$$

**Joint Score Matching Objective.** We adopt the joint training strategy from Choi et al. (2022); Yu et al. (2025), where the joint score network $s_{\boldsymbol{\theta}}^{(\boldsymbol{x}, t)}(\boldsymbol{x}, t) \triangleq [s_{\boldsymbol{\theta}}^{(\boldsymbol{x})}(\boldsymbol{x}, t), s_{\boldsymbol{\theta}}^{(t)}(\boldsymbol{x}, t)]$ is trained to simultaneously predict both the conditional time score and the conditional data score. The network thus outputs a scalar time score $s_{\boldsymbol{\theta}}^{(t)} \in \mathbb{R}$ and a vector data score $s_{\boldsymbol{\theta}}^{(\boldsymbol{x})} \in \mathbb{R}^d$.

The time score target is $s_{\text{DI}}^{(t)}(\boldsymbol{x}_t, t; \boldsymbol{x}_1) \triangleq \partial_t \log p_t(\boldsymbol{x}_t \mid \boldsymbol{x}_1)$, given by:

$$s_{\text{DI}}^{(t)}(\boldsymbol{x}_t, t; \boldsymbol{x}_1) = \frac{\dot{\alpha}(t)}{\alpha(t)} \left( \|\boldsymbol{x}_0\|^2 - d \right) + \frac{\dot{\beta}(t)}{\alpha(t)} \boldsymbol{x}_0^\top \boldsymbol{x}_1. \tag{60}$$

The data score target is $s_{\text{DI}}^{(\boldsymbol{x})}(\boldsymbol{x}_t, t; \boldsymbol{x}_1) \triangleq \nabla_{\boldsymbol{x}_t} \log p_t(\boldsymbol{x}_t \mid \boldsymbol{x}_1)$. This is given by:

$$s_{\text{DI}}^{(\boldsymbol{x})}(\boldsymbol{x}_t, t; \boldsymbol{x}_1) = -\frac{\boldsymbol{x}_t - \beta(t)\boldsymbol{x}_1}{\alpha(t)^2} = -\frac{\alpha(t)\boldsymbol{x}_0}{\alpha(t)^2} = -\frac{\boldsymbol{x}_0}{\alpha(t)}. \tag{61}$$

Thus, the conditional joint score matching objective (CJSM) is to minimize the sum of the mean squared errors for both scores:

$$\mathcal{L}_{\text{CJSM}}(\boldsymbol{\theta}) = \mathbb{E}_{p(t)p_t(\boldsymbol{x}_t)p_1(\boldsymbol{x}_1)} \left| s_{\text{DI}}^{(\boldsymbol{x},t)}(\boldsymbol{x}_t, t; \boldsymbol{x}_1) - s_{\boldsymbol{\theta}}^{(\boldsymbol{x},t)}(\boldsymbol{x}_t, t) \right|^2, \tag{62}$$

where $s_{\text{DI}}^{(\boldsymbol{x},t)}(\boldsymbol{x}_t, t; \boldsymbol{x}_1) = [s_{\text{DI}}^{(\boldsymbol{x})}(\boldsymbol{x}_t; \boldsymbol{x}_1), s_{\text{DI}}^{(t)}(\boldsymbol{x}_t; \boldsymbol{x}_1)]$ is the joint score target and $s_{\boldsymbol{\theta}}^{(\boldsymbol{x},t)}(\boldsymbol{x}_t, t) = [s_{\boldsymbol{\theta}}^{(\boldsymbol{x})}(\boldsymbol{x}_t, t), s_{\boldsymbol{\theta}}^{(t)}(\boldsymbol{x}_t, t)]$ is the joint score model.

**Total Objective.** The total objective is the weighted sum of the path variance objective and the joint score matching objective:

$$\mathcal{L}_{\text{total}}(\boldsymbol{\theta}, \boldsymbol{\phi}) = \lambda_1 \mathcal{L}_{\text{CJSM}}(\boldsymbol{\theta}) + \lambda_2 \mathcal{V}_{\text{DI}}[\alpha_{\boldsymbol{\phi}}, \beta_{\boldsymbol{\phi}}], \quad \lambda_1 + \lambda_2 = 1, \tag{63}$$

where $\lambda_1$ and $\lambda_2$ are the weighting coefficients determined by the Soft Optimal Uncertainty Weighting (UW-SO) method (see Sec. B.2.4 for details).

### B.2.3 TOTAL OBJECTIVE FOR DRE WITH THE DDBI

**Path Variance Objective.** Similarly, for the DDBI setting, the KMM path parameters $\boldsymbol{\phi}$ are learned by minimizing the corresponding analytical variance from Proposition 4.3:

$$\min_{\boldsymbol{\phi}} \quad \mathcal{V}_{\text{DDBI}}[\alpha_{\boldsymbol{\phi}}, \beta_{\boldsymbol{\phi}}] = \int_0^1 \left( \frac{d}{2} \frac{(\dot{\sigma}_t^2)^2}{(\sigma_t^2)^2} + \frac{\mathbb{E}_{p_0(\boldsymbol{x}_0)p_1(\boldsymbol{x}_1)} \left\| \dot{\alpha}_{\boldsymbol{\phi}}(t)\boldsymbol{x}_0 + \dot{\beta}_{\boldsymbol{\phi}}(t)\boldsymbol{x}_1 \right\|^2}{\sigma_t^2} \right) dt, \tag{64}$$

where $\sigma_t^2 = t(1-t)\gamma^2 + (\alpha_{\boldsymbol{\phi}}(t)^2 + \beta_{\boldsymbol{\phi}}(t)^2)\varepsilon$, $\gamma \geq 0$ is the noise factor, and $\varepsilon$ is a small number.

**Joint Score Matching Objective.** Following the same joint training strategy, the network predicts both the conditional time and data scores. The target scores are conditioned on the endpoints $(\boldsymbol{x}_0, \boldsymbol{x}_1)$.

The time score target is $s_{\text{DDBI}}^{(t)}(\boldsymbol{x}_t, t; \boldsymbol{x}_0, \boldsymbol{x}_1) \triangleq \partial_t \log p_t(\boldsymbol{x}_t \mid \boldsymbol{x}_0, \boldsymbol{x}_1)$, given by:

$$s_{\text{DDBI}}^{(t)}(\boldsymbol{x}_t, t; \boldsymbol{x}_0, \boldsymbol{x}_1) = \frac{\dot{\sigma}_t^2}{2\sigma_t^2} \left( \frac{\|\boldsymbol{z}\|^2}{\sigma_t^2} - d \right) + \frac{(\dot{\alpha}(t)\boldsymbol{x}_0 + \dot{\beta}(t)\boldsymbol{x}_1)^\top \boldsymbol{z}}{\sigma_t}. \tag{65}$$

The data score target is $s_{\text{DDBI}}^{(\boldsymbol{x})}(\boldsymbol{x}_t, t; \boldsymbol{x}_0, \boldsymbol{x}_1) \triangleq \nabla_{\boldsymbol{x}_t} \log p_t(\boldsymbol{x}_t \mid \boldsymbol{x}_0, \boldsymbol{x}_1)$, which simplifies to:

$$s_{\text{DDBI}}^{(\boldsymbol{x})}(\boldsymbol{x}_t, t; \boldsymbol{x}_0, \boldsymbol{x}_1) = -\frac{\boldsymbol{x}_t - (\alpha(t)\boldsymbol{x}_0 + \beta(t)\boldsymbol{x}_1)}{\sigma_t^2} = -\frac{\sigma_t \boldsymbol{z}}{\sigma_t^2} = -\frac{\boldsymbol{z}}{\sigma_t}. \tag{66}$$

In these expressions, $\sigma_t^2 = t(1-t)\gamma^2 + (\alpha(t)^2 + \beta(t)^2)\varepsilon$ and $\boldsymbol{z} = \frac{\boldsymbol{x}_t - (\alpha(t)\boldsymbol{x}_0 + \beta(t)\boldsymbol{x}_1)}{\sigma_t}$.

Similarly, the CJSM objective is to minimize the sum of the mean squared errors for both scores:

$$\mathcal{L}_{\text{CJSM}}(\boldsymbol{\theta}) = \mathbb{E}_{p(t)p_t(\boldsymbol{x}_t)p_0(\boldsymbol{x}_0)p_1(\boldsymbol{x}_1)} \left| s_{\text{DDBI}}^{(\boldsymbol{x},t)}(\boldsymbol{x}_t, t; \boldsymbol{x}_0, \boldsymbol{x}_1) - s_{\boldsymbol{\theta}}^{(\boldsymbol{x},t)}(\boldsymbol{x}_t, t) \right|^2, \tag{67}$$

where $s_{\text{DDBI}}^{(\boldsymbol{x},t)}(\boldsymbol{x}_t, t; \boldsymbol{x}_0, \boldsymbol{x}_1) = [s_{\text{DDBI}}^{(\boldsymbol{x})}(\boldsymbol{x}_t, t; \boldsymbol{x}_0, \boldsymbol{x}_1), s_{\text{DDBI}}^{(t)}(\boldsymbol{x}_t, t; \boldsymbol{x}_0, \boldsymbol{x}_1)]$ is the joint score target and $s_{\boldsymbol{\theta}}^{(\boldsymbol{x},t)}(\boldsymbol{x}_t, t) = [s_{\boldsymbol{\theta}}^{(\boldsymbol{x})}(\boldsymbol{x}_t, t), s_{\boldsymbol{\theta}}^{(t)}(\boldsymbol{x}_t, t)]$ is the joint score model.

**Total Objective.** Similarly, the total objective is given by:

$$\mathcal{L}_{\text{total}}(\boldsymbol{\theta}, \boldsymbol{\phi}) = \lambda_1 \mathcal{L}_{\text{CJSM}}(\boldsymbol{\theta}) + \lambda_2 \mathcal{V}_{\text{DDBI}}[\alpha_{\boldsymbol{\phi}}, \beta_{\boldsymbol{\phi}}], \quad \lambda_1 + \lambda_2 = 1. \tag{68}$$

### B.2.4 ADAPTIVE OBJECTIVE WEIGHTING TRICK

We balance heterogeneous training objectives, including the path and the joint score matching objectives, using the soft optimal uncertainty weighting (UW-SO) method of Kirchdorfer et al. (2024), an analytical refinement of the uncertainty-based weighting framework of Kendall et al. (2018).

UW-SO computes closed-form weights at every optimization step, eliminating the need for learned noise parameters while preserving the benefits of uncertainty-aware balancing.

Given $J$ loss terms $\{\mathcal{L}_j\}_{j=1}^J$, we compute their normalized weights as

$$\lambda_j = \frac{\exp\left(\frac{1}{T(\mathcal{L}_j + \epsilon)}\right)}{\sum_{j=1}^J \exp\left(\frac{1}{T(\mathcal{L}_j + \epsilon)}\right)}, \tag{69}$$

where $\epsilon > 0$ ensures numerical stability and $T > 0$ is a softmax temperature (we set $T = 2$ by default). The final training objective is a stop-gradient weighted sum:

$$\mathcal{L}_{\text{total}} = \sum_{j=1}^J \mathsf{sg}(\lambda_j)\mathcal{L}_j, \tag{70}$$

where $\mathsf{sg}$ denotes the stop-gradient operation, preventing the weights from affecting the backward pass. This yields a parameter-free, stable, and easily reproducible loss combination strategy, which we adopt as the default in all experiments.

### B.2.5 Variance-based Importance Sampling for Timesteps

Rather than sampling $t \sim \mathcal{U}(0, 1)$, we adopt a variance-based importance sampler inspired by Song et al. (2021a); Yu et al. (2025), which improves training stability by focusing computational effort on time steps where the ground-truth score signal is more reliable (i.e., has lower variance).

We utilize this sampler to fundamentally simplify the training objective. We start with the objective containing the optimal inverse-variance weighting function, $\lambda(t) \propto 1/\left(\text{Var}_{p_t(\boldsymbol{x})}\left(s^{(t)}(\boldsymbol{x}, t)\right) + \varepsilon\right)$, and employ importance sampling by drawing time steps $t \sim p(t)$. The resulting objective is proportional to:

$$\mathcal{L}_{\text{TSM}} \propto \mathbb{E}_{t \sim p(t)}\left[\frac{\lambda(t)}{p(t)} \cdot \left|s^{(t)}(\boldsymbol{x}, t) - s_{\boldsymbol{\theta}}^{(t)}(\boldsymbol{x}, t)\right|^2\right]. \tag{71}$$

By setting the sampling distribution $p(t)$ to be directly proportional to this optimal weighting function:

$$p(t) \propto \lambda(t) \propto \frac{1}{\text{Var}_{p_t(\boldsymbol{x})}\left(s^{(t)}(\boldsymbol{x}, t)\right) + \varepsilon}, \tag{72}$$

the importance sampling factor $\frac{\lambda(t)}{p(t)}$ becomes a constant, $\frac{1}{C}$. This results in an effectively uniform weighting of the squared error term:

$$\mathcal{L}_{\text{TSM}} \propto \frac{1}{C} \cdot \mathbb{E}_{t \sim p(t)}\left[\left|s^{(t)}(\boldsymbol{x}, t) - s_{\boldsymbol{\theta}}^{(t)}(\boldsymbol{x}, t)\right|^2\right] \propto \mathbb{E}_{t \sim p(t)}\left[\left|s^{(t)}(\boldsymbol{x}, t) - s_{\boldsymbol{\theta}}^{(t)}(\boldsymbol{x}, t)\right|^2\right]. \tag{73}$$

This mathematically guarantees the benefits of optimal weighting while maintaining an objective free of explicit, non-unity weighting functions.

Analytical expressions of the variance are available for DI and DDBI (Proposition 4.3). In practice, we pre-compute variances on a time grid and sample $t$ from the resulting categorical distribution, updating it periodically as the path schedule $\left(\alpha_{\boldsymbol{\phi}}(t), \beta_{\boldsymbol{\phi}}(t)\right)$ evolves.

### B.3 Experimental Settings for the Kumaraswamy Distribution

**A Stable Kumaraswamy.** The Kumaraswamy distribution suffers from severe numerical issues near the $[0, 1]$ boundaries, where direct evaluation of $t^a$ or $(1 - t^a)^b$ leads to catastrophic cancellation and unstable gradients (Wasserman & Mateos, 2025). To address this, we compute powers as,

$$x^y = \exp\left(y \cdot \text{log1p}(x - 1)\right), \tag{74}$$

which preserves precision by avoiding subtraction of nearly equal numbers. This stable power function is applied to the CDF and PDF (Eq. (13)), ensuring reliable gradients during training.

**Initialization Strategy for Mixture Components.** To prevent component collapse and encourage specialization, we initialize the $K$ Kumaraswamy components with modes spread across $[0, 1]$. For each $\tau_k = \frac{k+0.5}{K+1}$, the shape parameters $(a_k, b_k)$ are set asymmetrically:

$$(a_k, b_k) = \begin{cases} (1.5 + 0.5k, 3.0 + 2.0(K - k)) & \text{if } \tau_k < 0.5, \\ (3.0 + 2.0k, 1.5 + 0.5(K - k)) & \text{if } \tau_k \geq 0.5. \end{cases} \tag{75}$$

This design produces right-skewed components ($a_k < b_k$) for $\tau_k < 0.5$ and left-skewed components ($a_k > b_k$) for $\tau_k \geq 0.5$, ensuring that their modes

$$\text{mode}(k) = \left( \frac{a_k - 1}{a_k b_k - 1} \right)^{1/a_k}, \qquad a_k > 1, b_k > 1, \tag{76}$$

are distributed across $[0, 1]$. The weights are initialized uniformly, $w_k = 1/K$. Such diversity-oriented initialization accelerates convergence and avoids degenerate solutions (Dai & Wipf, 2020).

**PyTorch implementation of MVP path with KMM parameterization.** We also provide the PyTorch implementation of the proposed MVP in this paper, as shown in Algorithm 3.

**Algorithm 3** PyTorch implementation of MVP path with KMM parameterization.

```python
class MVPPath(nn.Module):
    """
    An implementation for computing the MVP path schedules
    alpha(t) and beta(t) using a Kumaraswamy Mixture Model (KMM).
    """
    def __init__(self, n_comp=4, constraint_type="affine", eps=1e-5):
        super().__init__()
        self.constraint_type, self.eps = constraint_type, eps

        # Learnable parameters for the KMM, see Sec. B.3 for details
        self.logits = nn.Parameter(torch.zeros(1, n_comp))
        self.log_a = nn.Parameter(torch.randn(1, n_comp))
        self.log_b = nn.Parameter(torch.randn(1, n_comp))

    def _get_kmm_params(self):
        "Applies transformations to get valid KMM parameters."
        weights = F.softmax(self.logits, dim=-1)
        a = F.softplus(self.log_a)  # Ensures a > 0
        b = F.softplus(self.log_b)  # Ensures b > 0
        return weights, a, b

    def _stable_pow(self, x, y):
        "Numerically stable computation of x^y for x close to 1."
        return torch.exp(y * torch.log1p(x - 1))

    def _kmm_cdf(self, t, weights, a, b):
        "Computes the KMM's Cumulative Distribution Function."
        t_a = self._stable_pow(t, a)
        comp_cdfs = 1 - self._stable_pow(1 - t_a, b)
        return torch.sum(weights * comp_cdfs, dim=-1, keepdim=True)

    def _kmm_pdf(self, t, weights, a, b):
        "Computes the KMM's Probability Density Function."
        t_a = self._stable_pow(t, a)
        t_a_m1 = self._stable_pow(t, a - 1)
        one_m_t_a_b_m1 = self._stable_pow(1 - t_a, b - 1)

        comp_pdfs = a * b * t_a_m1 * one_m_t_a_b_m1
        return torch.sum(weights * comp_pdfs, dim=-1, keepdim=True)

    def forward(self, t):
        "Computes alpha, beta, and their derivatives at time t."
        t = t.clamp(self.eps, 1 - self.eps)
        weights, a, b = self._get_kmm_params()

        # alpha is the inverted CDF; dot_alpha is the negative PDF
        # see Eq. (14) for details
        alpha = 1 - self._kmm_cdf(t, weights, a, b)
        dot_alpha = -self._kmm_pdf(t, weights, a, b)

        # Beta and its derivative depend on the constraint
        if self.constraint_type == "spherical":
            # In this case, alpha(t)^2 + beta(t)^2 = 1
            beta = torch.sqrt(1 - alpha.pow(2) + self.eps)
            dot_beta = -alpha * dot_alpha / beta
        else:  # "affine"
            # In this case, alpha(t) + beta(t) = 1
            beta = 1 - alpha
            dot_beta = -dot_alpha
        return alpha, beta, dot_alpha, dot_beta
```

### B.4 EXPERIMENTAL SETTINGS AND RESULTS FOR $f$-DIVERGENCE ESTIMATION

**Mutual Information (MI) Estimation.** MI quantifies the statistical dependence between two random variables $\mathbf{x} \sim p(\boldsymbol{x})$ and $\mathbf{y} \sim q(\boldsymbol{y})$, and is defined as $\text{MI}(\mathbf{x}, \mathbf{y}) = \mathbb{E}_{p(\boldsymbol{x},\boldsymbol{y})}\left[\log \frac{p(\boldsymbol{x},\boldsymbol{y})}{p(\boldsymbol{x})q(\boldsymbol{y})}\right]$. MI estimation naturally reduces to DRE since $\text{MI}(\mathbf{x}, \mathbf{y})$ involves the expectation of a log density ratio.

**Details on Geometrically Pathological Distributions.** To further probe the robustness of our method, we evaluate it on a suite of five MI estimation tasks involving geometrically pathological distributions. These tasks, inspired by the benchmark suite from Czyż et al. (2023), are specifically designed to challenge the underlying assumptions of many standard estimators.

1. **Edge-Singular Gaussian.** This task assesses model stability in edge-case correlation scenarios. We consider two jointly Gaussian random variables, $(\mathbf{x}, \mathbf{y}) \sim \mathcal{N}(\mathbf{0}, [[1, \rho], [\rho, 1]])$. The mutual information, given by $\text{MI}(\mathbf{x}; \mathbf{y}) = -0.5 \log(1 - \rho^2)$, becomes numerically challenging at the extremes of the correlation coefficient $\rho$. As $\rho \to 1^-$, the distributions become nearly degenerate and collinear, leading to numerical divergence in many estimators. Conversely, as $\rho \to 0^+$, the dependency signal becomes vanishingly weak, testing the model's sensitivity and its ability to avoid vanishing gradients.

2. **Half-Cube Map.** This distribution challenges models with heavy-tailed data. We begin with two correlated Gaussian variables and apply the homeomorphism half-cube$(\mathbf{z}) = |\mathbf{z}|^{3/2}\text{sign}(\mathbf{z})$ to each. The resulting variables, $\mathbf{x}' = \text{half-cube}(\mathbf{x})$ and $\mathbf{y}' = \text{half-cube}(\mathbf{y})$, have significantly heavier tails than the original Gaussians. While this transformation preserves the mutual information, i.e., $\text{MI}(\mathbf{x}'; \mathbf{y}') = \text{MI}(\mathbf{x}; \mathbf{y})$, the distorted, heavy-tailed geometry can be difficult for non-parametric methods that rely on local density estimation.

3. **Asinh Mapping.** This task tests the model's ability to handle distributions with highly concentrated densities. We start with two independent standard Gaussian random variables, $\mathbf{x}$ and $\mathbf{y}$, and apply the inverse hyperbolic sine transformation, $\text{asinh}(\mathbf{z}) = \log(\mathbf{z} + \sqrt{\mathbf{z}^2 + 1})$, to each. The resulting variables, $\mathbf{x}' = \text{asinh}(\mathbf{x})$ and $\mathbf{y}' = \text{asinh}(\mathbf{y})$, form our two distributions. This mapping compresses the tails of the original Gaussian distributions into a central region with extremely high curvature, creating sharp density peaks that can cause gradient instability and pose significant challenges for kernel-based methods.

4. **Additive Noise.** This scenario is designed to evaluate performance on distributions with fragmented support and sharp discontinuities. Let $\mathbf{x}$ be a random variable uniformly distributed on $[0, 1]$, i.e., $\mathbf{x} \sim \mathcal{U}(\mathbf{0}, \mathbf{1})$, and let $\mathbf{n}$ be an independent noise variable, $\mathbf{n} \sim \mathcal{U}(-\epsilon, \epsilon)$. We define the second variable as $\mathbf{y} = \mathbf{x} + \mathbf{n}$. The resulting joint distribution is piecewise-constant and possesses sharp, non-differentiable boundaries. This structure violates the smoothness assumptions underlying many score-based estimators, providing a stringent test of model robustness. The true MI is given by $\text{MI}(\mathbf{x}; \mathbf{y}) = \log(2\epsilon) + 0.5$ for $\epsilon \leq 0.5$.

5. **Gamma-Exponential.** This task features a complex, non-linear dependency structure. We define a hierarchical relationship where the first variable $\mathbf{x}$ is drawn from a Gamma distribution, $\mathbf{x} \sim \text{Gamma}(\rho, 1)$. The second variable $\mathbf{y}$ is then drawn from an Exponential distribution whose rate parameter is given by the value of $\mathbf{x}$, i.e., $\mathbf{y} \mid \mathbf{x} = \boldsymbol{x} \sim \text{Exponential}(\boldsymbol{x})$. The resulting joint distribution is highly asymmetric and non-Gaussian. The true mutual information is $I(\mathbf{x}; \mathbf{y}) = \psi(\rho + 1) - \log(\rho)$, where $\psi$ is the digamma function.

The results in Tab. 6 can be summarized as follows: (1) On the Edge-Singular Gaussian task, MVP (both affine and spherical) attains the lowest or near-lowest MSE, especially under extreme correlations ($\rho \to 0, \pm 1$). On the Half-Cube Map task with heavy tails, the spherical constraint yields stable and accurate estimates, outperforming all baselines. (2) On the Asinh Mapping task with concentrated densities and sharp curvature, MVP again achieves the lowest error, while baselines such as VP struggle with density peaks. (3) The largest gains appear on the Additive Noise and Gamma–Exponential tasks. With sharp discontinuities and complex non-linear dependencies, fixed-path methods degrade severely, whereas MVP, particularly with the spherical constraint, achieves dramatically lower MSE, often by an order of magnitude.

Overall, no fixed path is uniformly effective across all five benchmarks, with performance varying by data geometry. In contrast, MVP consistently delivers state-of-the-art accuracy by optimizing path variance, producing data-adaptive trajectories tailored to each problem's structure.

Table 6: Full MSE results for MI estimation on five geometrically pathological datasets. The top row of each sub-table indicates the varying correlation coefficient $\rho$. Our MVP path demonstrates consistently superior or competitive performance across the wide range of challenging data geometries, highlighting the benefits of a learnable path.

(a) MSE results for the Edge-Singular Gaussian dataset.

| Path | Constraint | -0.9 | -0.8 | -0.7 | -0.6 | -0.5 | -0.4 | -0.3 | -0.2 | -0.1 | 0.0 | 0.1 | 0.2 | 0.3 | 0.4 | 0.5 | 0.6 | 0.7 | 0.8 | 0.9 |
|---|---|---|---|---|---|---|---|---|---|---|---|---|---|---|---|---|---|---|---|---|
| Linear | affine | 0.0025 | 0.0010 | 0.0005 | **0.0001** | 0.0002 | 0.0005 | 0.0002 | 0.0001 | 0.0000 | 0.0002 | 0.0002 | 0.0002 | 0.0002 | 0.0001 | 0.0002 | **0.0001** | 0.0005 | 0.0006 | 0.0014 |
| Cosine | spherical | 0.0014 | 0.0007 | 0.0007 | 0.0003 | 0.0006 | 0.0005 | 0.0002 | 0.0003 | 0.0005 | **0.0001** | 0.0003 | 0.0002 | 0.0003 | 0.0004 | 0.0003 | 0.0005 | 0.0008 | 0.0006 | 0.0028 |
| Föllmer | spherical | 0.0007 | 0.0012 | 0.0004 | 0.0004 | 0.0004 | 0.0004 | 0.0003 | 0.0002 | 0.0002 | **0.0001** | **0.0001** | **0.0001** | **0.0001** | 0.0002 | 0.0004 | 0.0005 | 0.0005 | 0.0006 | 0.0009 |
| Trigonometric | spherical | 0.0021 | 0.0004 | 0.0002 | 0.0002 | 0.0004 | 0.0004 | 0.0004 | 0.0001 | 0.0002 | 0.0003 | 0.0004 | **0.0001** | **0.0001** | 0.0001 | 0.0002 | 0.0003 | 0.0003 | 0.0010 | 0.0034 |
| VP | spherical | 0.0013 | 0.0003 | 0.0014 | 0.0009 | 0.0004 | 0.0007 | 0.0014 | 0.0020 | 0.0021 | 0.0025 | 0.0022 | 0.0020 | 0.0013 | 0.0006 | 0.0001 | 0.0005 | 0.0003 | **0.0004** | 0.0023 |
| **MVP (ours)** | affine | 0.0010 | **0.0002** | 0.0005 | 0.0005 | **0.0001** | **0.0003** | **0.0001** | 0.0001 | **0.0000** | 0.0002 | **0.0001** | **0.0001** | **0.0001** | **0.0000** | 0.0001 | **0.0001** | **0.0002** | 0.0007 | **0.0001** |
| **MVP (ours)** | spherical | **0.0005** | 0.0006 | **0.0001** | 0.0003 | 0.0004 | 0.0004 | 0.0002 | **0.0000** | 0.0001 | 0.0002 | 0.0001 | 0.0001 | 0.0004 | 0.0001 | **0.0000** | 0.0004 | 0.0005 | 0.0006 | 0.0007 |

(b) MSE results for the Half-Cube Map dataset.

| Path | Constraint | -0.9 | -0.8 | -0.7 | -0.6 | -0.5 | -0.4 | -0.3 | -0.2 | -0.1 | 0.0 | 0.1 | 0.2 | 0.3 | 0.4 | 0.5 | 0.6 | 0.7 | 0.8 | 0.9 |
|---|---|---|---|---|---|---|---|---|---|---|---|---|---|---|---|---|---|---|---|---|
| Linear | affine | 0.0007 | 0.0006 | 0.0005 | 0.0004 | 0.0007 | 0.0007 | 0.0006 | **0.0005** | 0.0009 | 0.0005 | 0.0005 | 0.0010 | 0.0016 | 0.0006 | 0.0009 | 0.0010 | 0.0013 | 0.0005 | 0.0019 |
| Cosine | spherical | 0.0015 | 0.0044 | 0.0024 | 0.0021 | 0.0005 | 0.0008 | 0.0041 | 0.0015 | 0.0038 | **0.0003** | 0.0058 | 0.0004 | 0.0029 | 0.0025 | 0.0004 | 0.0004 | 0.0006 | 0.0005 | **0.0008** |
| Föllmer | spherical | 0.0046 | 0.0039 | 0.0022 | 0.0011 | 0.0031 | 0.0050 | 0.0043 | 0.0045 | 0.0037 | 0.0042 | 0.0036 | 0.0055 | 0.0032 | 0.0035 | 0.0003 | **0.0003** | 0.0004 | **0.0003** | 0.0010 |
| Trigonometric | spherical | 0.0020 | 0.0005 | 0.0004 | 0.0004 | 0.0009 | 0.0006 | 0.0006 | 0.0006 | 0.0006 | 0.0012 | 0.0005 | **0.0003** | 0.0006 | 0.0002 | 0.0003 | 0.0006 | 0.0007 | 0.0005 | 0.0034 |
| VP | spherical | 0.0006 | 0.0009 | 0.0038 | 0.0035 | 0.0037 | 0.0071 | 0.0083 | 0.0106 | 0.0095 | 0.0086 | 0.0006 | 0.0085 | 0.0060 | 0.0067 | 0.0033 | 0.0049 | 0.0006 | 0.0021 | 0.0011 |
| **MVP (ours)** | affine | **0.0005** | **0.0004** | 0.0006 | **0.0003** | **0.0004** | 0.0006 | 0.0007 | 0.0007 | 0.0008 | 0.0008 | 0.0008 | 0.0026 | 0.0014 | 0.0025 | **0.0002** | 0.0007 | 0.0004 | 0.0004 | 0.0013 |
| **MVP (ours)** | spherical | 0.0005 | 0.0005 | **0.0002** | 0.0006 | 0.0008 | **0.0002** | **0.0003** | **0.0005** | **0.0002** | 0.0012 | **0.0002** | 0.0004 | **0.0006** | 0.0006 | **0.0002** | **0.0003** | **0.0003** | **0.0003** | 0.0014 |

(c) MSE results for the Asinh Mapping dataset.

| Path | Constraint | -0.9 | -0.8 | -0.7 | -0.6 | -0.5 | -0.4 | -0.3 | -0.2 | -0.1 | 0.0 | 0.1 | 0.2 | 0.3 | 0.4 | 0.5 | 0.6 | 0.7 | 0.8 | 0.9 |
|---|---|---|---|---|---|---|---|---|---|---|---|---|---|---|---|---|---|---|---|---|
| Linear | affine | 0.0004 | 0.0010 | 0.0005 | 0.0005 | 0.0006 | 0.0237 | 0.0007 | 0.0001 | 0.0012 | **0.0004** | 0.0003 | 0.0000 | 0.0003 | 0.0002 | 0.0006 | 0.0008 | 0.0006 | 0.0012 | 0.0166 |
| Cosine | spherical | 0.0058 | 0.0016 | 0.0006 | 0.0009 | 0.0015 | 0.0015 | 0.0017 | 0.0015 | 0.0034 | 0.0007 | 0.0002 | 0.0001 | **0.0001** | 0.0002 | 0.0004 | 0.0005 | 0.0004 | 0.0009 | 0.0014 |
| Föllmer | spherical | 0.0024 | 0.0009 | 0.0017 | 0.0019 | 0.0015 | 0.0018 | 0.0014 | 0.0040 | 0.0027 | 0.0019 | 0.0001 | 0.0001 | 0.0001 | 0.0002 | 0.0007 | 0.0009 | **0.0001** | 0.0007 | 0.0022 |
| Trigonometric | spherical | 0.0015 | 0.0009 | 0.0005 | 0.0004 | 0.0003 | 0.0003 | 0.0005 | 0.0003 | 0.0004 | 0.0005 | 0.0007 | 0.0001 | 0.0001 | **0.0004** | 0.0005 | 0.0015 | 0.0012 | 0.0007 |  |
| VP | spherical | 0.0005 | 0.0008 | 0.0020 | 0.0047 | 0.0048 | 0.0099 | 0.0101 | 0.0108 | 0.0118 | 0.0012 | 0.0023 | 0.0012 | 0.0027 | 0.0014 | 0.0014 | 0.0005 | 0.0005 | 0.0020 | 0.0020 |
| **MVP (ours)** | affine | 0.0003 | **0.0003** | 0.0004 | **0.0002** | 0.0001 | **0.0003** | **0.0005** | **0.0000** | 0.0002 | **0.0004** | **0.0001** | 0.0003 | 0.0005 | 0.0005 | 0.0005 | 0.0005 | **0.0001** | 0.0003 | **0.0004** |
| **MVP (ours)** | spherical | **0.0002** | **0.0003** | **0.0003** | 0.0007 | **0.0001** | 0.0007 | 0.0010 | **0.0000** | **0.0001** | 0.0000 | 0.0006 | **0.0000** | 0.0001 | 0.0007 | 0.0005 | **0.0003** | 0.0004 | **0.0002** | 0.0006 |

(d) MSE results for the Additive Noise dataset.

| Path | Constraint | 0.1 | 0.2 | 0.3 | 0.4 | 0.5 | 0.6 | 0.7 | 0.8 | 0.9 |
|---|---|---|---|---|---|---|---|---|---|---|
| Linear | affine | 0.3053 | 0.0541 | 0.0233 | 0.0465 | 0.0367 | 0.0714 | 0.0339 | 0.0307 | 0.0268 |
| Cosine | spherical | 0.4803 | 0.1086 | 0.0788 | 0.0969 | 0.0719 | 0.0867 | 0.1015 | 0.1204 | 0.1304 |
| Föllmer | spherical | 0.4731 | 0.1048 | 0.0510 | 0.0916 | 0.0864 | 0.1024 | 0.1226 | 0.0944 | 0.1115 |
| Trigonometric | spherical | 0.3615 | 0.0565 | 0.0508 | 0.0654 | 0.0596 | 0.0517 | 0.0502 | 0.0448 | 0.0603 |
| VP | spherical | 0.3455 | 0.0290 | 0.0303 | 0.0346 | 0.0210 | 0.0313 | 0.0426 | 0.0390 | 0.0123 |
| **MVP (ours)** | affine | 0.2360 | 0.0199 | 0.0081 | 0.0167 | 0.0161 | 0.0216 | 0.0304 | 0.0155 | **0.0010** |
| **MVP (ours)** | spherical | **0.1016** | **0.0134** | **0.0078** | **0.0009** | **0.0138** | **0.0117** | **0.0196** | **0.0026** | 0.0029 |

(e) MSE results for the Gamma-Exponential dataset.

| Path | Constraint | 1.0 | 1.1 | 1.2 | 1.3 | 1.4 | 1.5 | 1.6 | 1.7 | 1.8 |
|---|---|---|---|---|---|---|---|---|---|---|
| Linear | affine | 2.8166 | 0.7064 | 0.1686 | 0.1368 | 0.0513 | 0.0420 | 0.0051 | 0.0147 | 0.0066 |
| Cosine | spherical | 7.2466 | 1.0785 | 0.4875 | 0.2582 | 0.2802 | 0.3063 | 0.2936 | 0.1864 | 0.3438 |
| Föllmer | spherical | 0.4006 | 0.4275 | 0.6304 | 0.3741 | 0.2822 | 0.4662 | 0.2473 | 0.3838 | 0.3105 |
| Trigonometric | spherical | 10.4420 | 1.3841 | 0.2705 | 0.3854 | 0.0184 | 0.0681 | 0.1109 | 0.0103 | 0.0668 |
| VP | spherical | 2.7080 | 1.3872 | 0.1116 | 0.1224 | 0.1320 | 0.1817 | 0.4407 | 0.0919 | 0.2507 |
| **MVP (ours)** | affine | 0.4785 | 1.5466 | 0.1137 | 0.1274 | 0.0915 | **0.0063** | **0.0032** | 0.0294 | 0.0069 |
| **MVP (ours)** | spherical | **0.2721** | **0.2304** | **0.0917** | **0.0434** | **0.0075** | 0.0187 | 0.0091 | **0.0030** | **0.0004** |

**High-dimensional & High-discrepancy Distributions.** To rigorously test model performance under extreme conditions, we designed an experiment focused on MI estimation between two high-dimensional Gaussian distributions with a large discrepancy. This setup is intended to specifically invoke the "density-chasm" problem, a known failure mode for DRE-based methods where the path between the two distributions traverses a region of vanishingly low density (Rhodes et al., 2020). The goal is to evaluate whether our path optimization strategy can effectively navigate this chasm where fixed paths may fail. We define the two distributions as $p_0(\boldsymbol{x}) = \mathcal{N}(\boldsymbol{0}, \boldsymbol{I}_d)$ and $p_1(\boldsymbol{x}) = \mathcal{N}(\boldsymbol{0}, \Sigma)$. The covariance matrix $\Sigma$ is constructed to be block-diagonal, where each $2 \times 2$ block along the diagonal is given by $\Lambda = \begin{pmatrix} 1 & \rho \\ \rho & 1 \end{pmatrix}$, with $\rho = 0.5$. This structure creates strong pairwise correlations within each block while ensuring that the blocks themselves are independent. This results in a highly ill-conditioned and low-rank covariance matrix, a known challenge for score-based DRE methods (Choi et al., 2022). The experiments are conducted across several dimensions, $d \in \{40, 80, 120, 160\}$. As the dimension increases, so does the true MI between the distributions, and thus the discrepancy,

with corresponding MI values of approximately $\{10, 20, 30, 40\}$ nats. This allows us to systematically evaluate model robustness as the density-chasm problem becomes progressively more severe.

## B.5 EXPERIMENTAL SETTINGS AND RESULTS FOR DENSITY ESTIMATION

**Structured and Multi-modal Distributions.** We use a suite of six 2D synthetic datasets to evaluate the model's ability to learn complex and structured distributions. Five of these are standard benchmarks used in Chen et al. (2025c), while the tree dataset follows the setup from Karras et al. (2024). These datasets are specifically chosen to probe different failure modes of density models.

- **Disconnected Topologies.** The circles and rings datasets test the model's ability to capture distributions with multiple, disconnected components and assign zero density to the regions between them.

- **Intricate Structures.** The 2spirals and pinwheel datasets feature highly structured, non-linear manifolds that require the model to learn complex, curving paths.

- **Discontinuous and Branching Densities.** The checkerboard dataset presents a particularly difficult challenge with its discontinuous, grid-like density. The tree dataset is designed to assess a model's capacity to generate sharp, branching topological structures, a known challenge for many generative models.

Together, these datasets form a comprehensive testbed for evaluating the flexibility and robustness of a density estimation method.

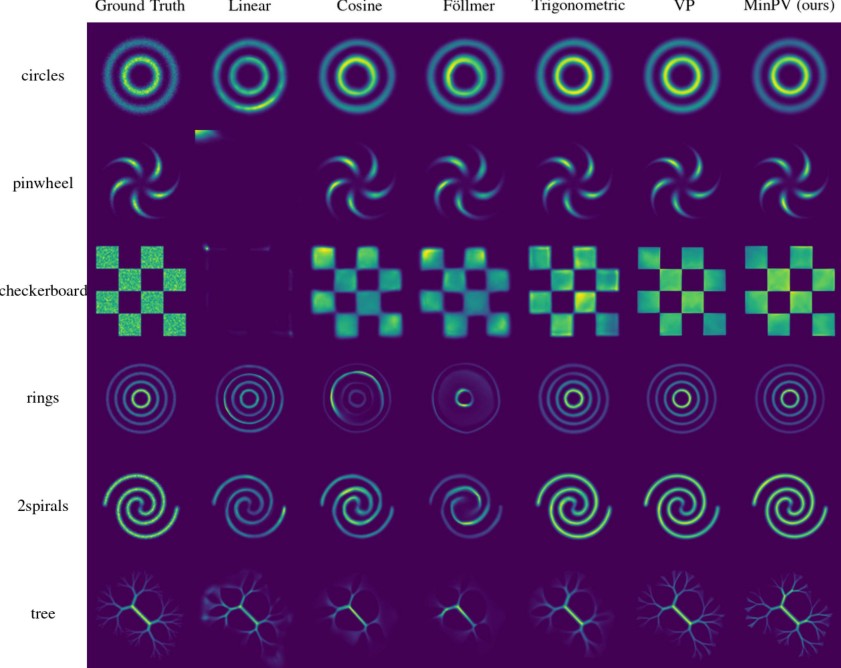

Figure 4: Density estimation on six structured and multi-modal datasets. MVP successfully learns a data-adaptive path tailored to the specific manifold of each dataset.

**Real-world Tabular Distributions.** We evaluate on five tabular datasets widely used in density estimation (Grathwohl et al., 2018), covering domains from particle physics to image patches. These datasets pose challenging, non-Gaussian structures with unknown generative processes and complex correlations, making them suitable for testing model expressiveness. We follow the preprocessing and splits of Grathwohl et al. (2018) for fair comparison.

### B.6 ILLUSTRATION OF OPTIMIZED PATH SCHEDULES

We visualize the path schedules learned under the MVP principle across different tasks and datasets. Here, "prior" and "posterior" denote the initialized and optimized path schedules.

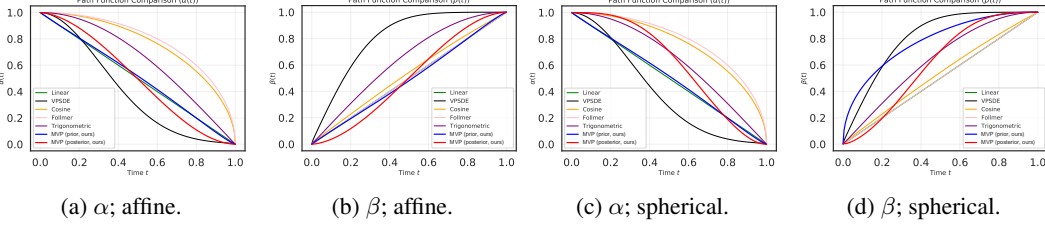

(a) $\alpha$; affine.  (b) $\beta$; affine.  (c) $\alpha$; spherical.  (d) $\beta$; spherical.

Figure 5: Optimized KMM-parameterized path schedule for geometrically pathological distributions.

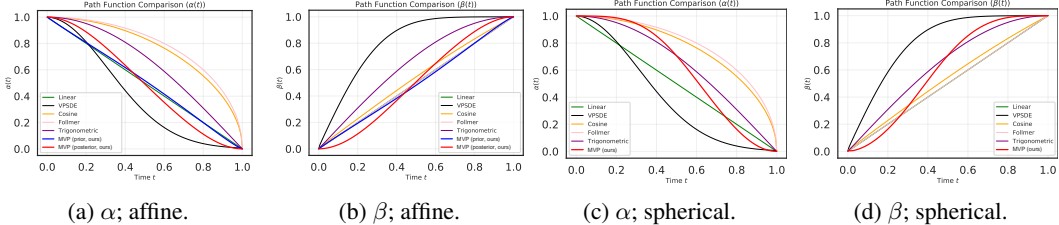

(a) $\alpha$; affine.  (b) $\beta$; affine.  (c) $\alpha$; spherical.  (d) $\beta$; spherical.

Figure 6: Optimized KMM-parameterized path schedule for high-dimensional distributions.

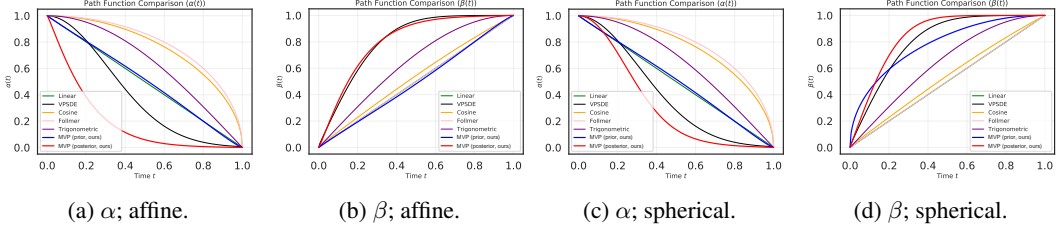

(a) $\alpha$; affine.  (b) $\beta$; affine.  (c) $\alpha$; spherical.  (d) $\beta$; spherical.

Figure 7: Optimized KMM-parameterized path schedule for structured and multi-modal datasets.

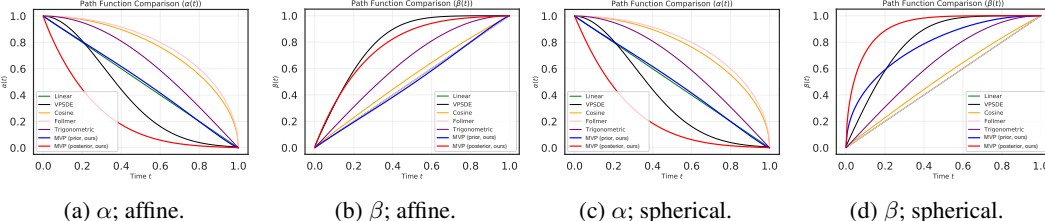

(a) $\alpha$; affine.  (b) $\beta$; affine.  (c) $\alpha$; spherical.  (d) $\beta$; spherical.

Figure 8: Optimized KMM-parameterized path schedule for real-world tabular datasets.

