# OpenReview forum: "A Minimum Variance Path Principle for Accurate and Stable Score-Based Density Ratio Estimation"
_ICLR.cc/2026/Conference — ICLR 2026 Poster_

### Official Review · Reviewer_8HVQ · 2025-10-22

**Soundness:** 2
**Presentation:** 3
**Contribution:** 2
**Rating:** 6
**Confidence:** 3

**Summary:**

Score-based approaches for Density Ratio Estimation (DRE) have gained interests in recent literature. While previous works typically employ a fixed probability path beforehand, the authors observe that the choice of the path can have implications on the downstream performances. In order to resolve the issue, the authors propose to explicitly learn the path by minimizing the resulting variance. Specifically, a Kumaraswamy Mixture Model (KMM) is employed to model the schedule. Empirical experiments are provided, which demonstrate that the proposed method helps to improve the performances of the DRE method.

**Strengths:**

1. The idea of learning the interpolation path in general is well-motivated.
2. The proposed solution relying on closed-form path variance is interesting.

**Weaknesses:**

1. As the authors themselves noted in the Appendix, the bound in Lemma 4.1 is with a specific choice of n and m, without strong justification of these choices.
2. Relatively minor, the experiments are of smaller scale problems.

**Questions:**

1. In the main paper the STSM objective is highlighted, e.g. the equality in Equation 9 relies on STSM objective. However, in Appendix B.2.1. it was noted "We Use CTSM". Which objective is employed in the experiments?
2. The empirical performances of the methods naturally depend on many factors, e.g. the weighting function \lambda(t) as employed in the objective. Have the authors made attempts to isolate the other factors to investigate whether the empirical improvements are indeed due to the choice of path?
3. In terms of broader literature, the variance of the time score appears in information geometry context as the time score is the Fisher score with respect to t. Furthermore, learning the paths have been investigated in diffusion literature, e.g. [1].

[1] Neural Diffusion Models, Bartoh et al. ICML 2024

---

> ### Author Response · Authors · 2025-11-18
>
> We sincerely thank the reviewer for their positive assessment of our work. We appreciate your thoughtful questions and address them below.
>
> > **[Q1]** ...the STSM objective is highlighted, e.g. the equality in Eq. 9 relies on STSM objective ... in Appendix B.2.1. it was noted "We Use CTSM". Which objective is employed in the experiments?
>
> A: Thank you for this important clarification request.
>
> - **We use CTSM ($L_ {CTSM}$) in all experiments**, as it is more stable and efficient: it uses a closed-form conditional score, avoids time derivatives, and requires only one score evaluation per batch.
> - **This is theoretically justified**: Prop. A.4 (Appen. A.6) proves that $L_ {TSM}$, $L_ {STSM}$, and $L_ {CTSM}$ share the **same population minimizer** $\theta^ {\star}$.
> - **We present STSM in the main text** (Eq. 9) because it arises directly from integration by parts on the ideal loss $L_ {TSM}$, yielding the exact decomposition  $L_ {TSM}(\theta^\star) =L_ {STSM}(\theta^*) + \mathcal{V},$ which reveals the path variance $\mathcal{V}$ as the missing, path-dependent term. This algebraic identity is essential for deriving the Minimum Path Variance Principle.
>
> Thus, while **CTSM is used in practice**, **STSM serves the necessary theoretical role**, and the two are fully compatible due to objective equivalence.
>
> > **[Q2]** The empirical performances of the methods naturally depend on many factors, e.g. the weighting function $\lambda(t)$ ... Have the authors made attempts to isolate the other factors to investigate whether the empirical improvements are indeed due to the choice of path?
>
> A: Yes, we carefully controlled for confounding factors to isolate the effect of the path. Specifically:
>
> - We **fixed the loss weighting** to be uniform ($\lambda(t) = 1$) in the underlying $L_{CTSM}$ objective, following Yu et al. (2025).
> - Instead of explicit weighting, we use **variance-based importance sampling** for $t$, where the sampling distribution $p(t) \propto 1/(\mathcal{V}(t) + \varepsilon)$ is **determined solely by the current path**.
>
> Thus, across all comparisons (MinPV vs. Linear/VP/Cosine/etc.), the only difference is the path itself, including
> - **The path schedule:** MinPV path is compared directly against five fixed, SOTA path heuristics (Linear, Cosine, Föllmer, Trigonometric, VP).
> - **The path constraint:** We conducted comprehensive ablation studies comparing the Affine ($\alpha(t)+\beta(t)=1$) and Spherical ($\alpha(t)^2+\beta(t)^2=1$) constraints, as discussed in **Sec. 5.2 (Ablation Studies)**. This further confirm that performance gains stem from the **data-adaptive path** learned by minimizing $\mathcal{V}$, not other design choices.
>
> > **[Q3]** ...the variance of the time score appears in information geometry context as the time score is the Fisher score w.r.t. t. ... learning the paths have been investigated in diffusion literature, e.g. [1].
>
> A: We agree that the time score’s variance is deeply connected to information geometry, and that path learning has been explored in diffusion models like [1].
>
> However, our work is the **first in DRE** to:
> - **identify the path variance $\mathcal{V}$ as the precise missing term** that explains the gap between ideal and tractable objectives (Eq. 9), and
> - provide a **closed-form, analytically tractable objective** (Prop. 4.3) to minimize it, directly optimizing the theoretical error bound.
>
> Unlike generative approaches that learn paths via variational objectives or trajectory straightening, our MinPV principle is **principled, bound-aware, and DRE-specific**.
>
> **To better situate our contribution, we have added a brief reference to Neural Diffusion Models in Sec. 2 (Related Work)**,  highlighted in blue.
>
> > **[W1]**  ...the bound in Lemma 4.1 is with a specific choice of n and m, without strong justification of these choices.
>
> A: The choice $n = \infty, m = 1$ is principled: it enables a clean decomposition of the error bound into the score-matching loss ($L_ {TSM}$) and the path variance ($\mathcal{V}$), the two core quantities our method optimizes. We provide the full justification immediately after Lemma 4.1 in the main text, now **highlighted in blue** for clarity.
>
> > **[W2]** Relatively minor, the experiments are of smaller scale problems.
>
> A: Thank you for the comment. While our benchmarks may not involve massive datasets, they are designed to be maximally challenging: we focus on **high-discrepancy regimes** (e.g., KL div. up to 40 nats [1], Tab. 3) where the density-chasm problem causes existing methods to fail [1]. This includes high-dimensional, discontinuous, and geometrically pathological distributions.
>
> By targeting **problem hardness rather than scale**, we directly validate the robustness of MinPV-DRE in the very settings where path choice matters most—providing strong evidence for the practical impact of our theory.
>
> [1] Rhodes, Benjamin, et al. Telescoping density-ratio estimation. NeurIPS 2020.

---

> > ### Comment · Reviewer_8HVQ · 2025-11-18
> >
> > I thank the authors for their detailed response. Nevertheless, a quick comment concerning the loss weighting: as far as I am aware, in Yu et al. (2025) the loss weighting for CTSM is always fixed to the so-called time score normalization as in their Equation 20.

---

> ### Author Response · Authors · 2025-11-19
>
> Thank you for this thoughtful follow-up. It gives us a valuable opportunity to clarify the relationship between our sampling strategy, the CTSM objective, and the theoretical derivation.
>
> **In short**: our approach is inspired by Yu et al. (2025)’s use of variance-aware importance sampling to achieve an effectively unweighted loss, but adapted to the unconditional DRE setting. The key points are:
>
> 1. **Yu et al. (2025) and CTSM weighting**:  As you correctly note, Yu et al. (2025) use **time-score normalization** in their CTSM objective, with weighting $\lambda(t) \propto \frac{1}{\mathrm{Var}_{p_t(x,z)}[\partial_t \log p_t(x|z)]}$, which relies on the **conditional score** (given latent $z$).
>
> 2. **Our adaptation for unconditional DRE**: In our setting, we work only with marginal distributions $p_0(x)$ and $p_1(x)$. **There is no latent variable $z$**, which is a more general case. Consequently, we consider the **unconditional time score** $\partial_ t \log p_ t(x)$, and define the variance as $\mathcal{V}(t) = Var_ {p_ t(x)}[\partial_t \log p_ t(x)].$ This is the quantity that appears in our error bound (Lemma 4.1) and is **explicitly tractable** thanks to Proposition 4.3.
>
> 3. **In our setting, the weighting function is a constant, i.e., $\lambda(t) = 1$.** To simplify the theoretical analysis of the estimation error (Sec.  4), we aim for an **effectively unweighted** score-matching loss—i.e., $\lambda(t) = 1$ in the computational objective. However, a naive unweighted loss would suffer from high variance.  Fortunately, because we derive a closed-form expression for $\mathcal{V}(t)$ (Proposition 4.3), we can instead **embed the optimal weighting into the time sampling distribution** via importance sampling $\tilde{p}(t) \propto \frac{1}{\mathcal{V}(t) + \varepsilon}.$ This achieves the benefits of variance-aware weighting without introducing an explicit $\lambda(t)$ in the loss, keeping the theoretical derivation clean while maintaining training stability.
>
> 4. **Clarification in the manuscript**: We have expanded the discussion of this design choice in **Appendix B.2.5 (lines 1355–1373)**. This section is now **highlighted in blue**.

---

> > ### Comment · Reviewer_8HVQ · 2025-11-25
> >
> > I thank the authors for their reply. However, I still find the authors' explanation confusing at times. The authors state that "There is no latent variable z", and they employ the CTSM objective. However, CTSM objective replies on a choice of latent variable z. As such, when the authors perform experiments, they would necessarily rely on a choice of z. Furthermore, when the authors employ constant weighting for the CTSM baseline, have the authors verified whether it would lead to exploding variances in some limit? Also, if I understood it correctly, the authors used as baseline CTSM with their own configurations. Would the results differ if the choice of the weighting functions follow what was employed in Yu et al. 2025?

---

> > > ### Author Response · Authors · 2025-11-27
> > >
> > > We sincerely thank the reviewer for their careful follow-up. We clarify the points below.
> > >
> > > > Clarification on "no latent variable z ".
> > >
> > > Our statement refers specifically to the **path variance term**, i.e., the gap between the ideal and tractable objectives in Eq. (9): $\mathcal{L}_ {\text{TSM}}(\theta) = \mathcal{L}_ {\text{STSM}}(\theta) + \int_ 0^1 \mathbb{E}_ {p_ t(x)} \big| \partial_t \log p_t(x) \big|^2 dt$. We prove in Theorem 4.2 that this extra term is exactly the path variance $\int_ 0^1 \mathrm{Var}_ {p_ t(x)}\big(\partial_ t \log p_ t(x)\big) dt$, which depends **only on the marginal path** {${p_ t}$} and **not on any latent variable** z. This is why prior works treated it as a constant—they fixed the path and ignored it. Our key contribution is deriving its closed-form expression (Proposition 4.3), enabling explicit minimization.
> > >
> > > **Importantly, using CTSM for model training is fully compatible with this observation.**  In Appendix A.6 (Proposition A.4), we prove that for a fixed path, CTSM, STSM, and TSM share the same minimizer $s^{(t)}_ {\theta^\star}(x,t) = \partial_t \log p_t(x)$. Therefore, we can safely replace $\mathcal{L}_ {\text{STSM}}$ in Eq. (9) with $\mathcal{L}_ {\text{CTSM}}$, yielding our total objective $\mathcal{L}_ {\text{total}} = \mathcal{L}_ {\text{CTSM}}(\theta) + \mathcal{V}$, where
> > >  $\mathcal{L}_ {\text{CTSM}}$ **depends on z (as it conditions on endpoints), but $\mathcal{V}$ does not**. The two components operate on different aspects: score estimation (with z) vs. path geometry (without z).
> > >
> > > > When the authors employ constant weighting for the CTSM baseline, have the authors verified whether it would lead to exploding variances in some limit?
> > >
> > > Yes, we confirm that constant timestep weighting (e.g., constant weighting with $p(t) = \mathrm{Uniform}[0,1]$) leads to **exploding variance** near the boundaries ($t \to 0$ or $t \to 1$).
> > >
> > > To address this, **all experiments in our paper use variance-based importance sampling**, as detailed in Appendix B.2.5: $p(t) \propto \frac{1}{\mathrm{Var}_{p_t(x)}\big(\partial_t \log p_t(x)\big) + \varepsilon}$. This ensures training focuses on low-variance (stable) time regions, which is essential for stable training and convergence.
> > >
> > >
> > > > Would the results differ if the choice of the weighting functions follow what was employed in Yu et al. 2025?
> > >
> > > Yu et al. (2025) propose weighting the loss by $\lambda(t) \propto 1 / \mathrm{Var}(\cdot)$. In importance sampling, setting the sampling distribution $p(t) \propto \lambda(t)$ makes the importance weight $\lambda(t)/p(t)$ a **constant**$1/C$. Consequently, the effective loss becomes: $\mathcal{L}_ {\text{TSM}} \propto \frac{1}{C} \cdot \mathbb{E}_ {t \sim p(t)} \big[ \big| s^{(t)} - s^{(t)}_ \theta \big|^2 \big] \propto \mathbb{E}_ {t \sim p(t)} \big[ \big| s^{(t)} - s^{(t)}_ \theta \big|^2 \big]$, i.e., **uniform weighting under the new sampling distribution**.
> > >
> > > Thus, our implementation exactly realizes the weighting of Yu et al. (2025), just via sampling rather than explicit reweighting.

---

> ### Comment · Reviewer_8HVQ · 2025-11-27
>
> I thank the authors for their response. However, the sampling approach and the reweighting approach, despite that they might be mathematically equivalent, can behave differently in practice; this is the entire point that [1] proposed the importance sampling approach, which effectively employs a different weighting function while benefiting from the stability of the original weighting function. Yu et al. (2025) also noted that it could be beneficial to simultaneously employ the weighting function and the importance sampling distribution, so as to approximately use uniform weighting while benefiting from the time score normalization weighting.
>
> [1] Maximum Likelihood Training of Score-Based Diffusion Models, Song et al., NeurIPS 2021

---

> > ### Author Response · Authors · 2025-11-28
> >
> > We sincerely thank the reviewer for this insightful comment and for pointing us to the relevant discussion in [1] and [2]. We fully agree that, while importance sampling and explicit reweighting are mathematically equivalent in expectation, their **practical behavior can differ** due to variance properties, estimator stability, and implementation details, which is precisely the motivation behind the importance sampling strategy proposed in [1, 2].
> >
> > Indeed, as the reviewer notes, the choice of weighting function $\lambda(t)$ can be equivalently realized by choosing the timestep sampling distribution $p(t) \propto \lambda(t)$. This means that different sampling strategies correspond to implicit choices of weighting. We are aware that recent work in DRE, specifically [3] (Sec. 4.3), systematically evaluates several $p(t)$ designs, including Uniform, Log-Normal, and **variance-based importance sampling** (using the conditional time-score variance).
> >
> > Following this line of inquiry, we also conducted preliminary experiments with these sampling schemes across multiple benchmarks. Consistent with [3], we found that **variance-based sampling yields the most stable and accurate results**, though in our case we use the marginal (non-conditional) time-score variance $\mathrm{Var}_{p_t(x)}(\partial_t \log p_t(x))$, which aligns with our focus on path-level optimization (rather than conditional score modeling). This choice also ensures compatibility with our analytical path variance $\mathcal{V}$, which is defined in terms of the marginal score.
> >
> > That said, we acknowledge that the sampling strategy is indeed an important design dimension. However, since **the main contribution of our paper is to learn an adaptive path schedule**, not to propose a new sampler, we adopt variance-based importance sampling as a **default, robust choice** that consistently performs well across tasks (as corroborated by [1] and our own ablations). This allows us to focus our experimental analysis and discussion on the core novelty: path optimization via minimum path variance, and its impact on estimation accuracy and stability.
> >
> > We appreciate the reviewer’s suggestion and agree that a deeper investigation into the interplay between sampling and path learning is an interesting direction for future work.
> >
> > [1] Song et al. Maximum Likelihood Training of Score-Based Diffusion Models, NeurIPS 2021
> >
> > [2] Yu, Hanlin, et al. Density Ratio Estimation with Conditional Probability Paths. ICML 2025.
> >
> > [3] Chen, Wei, et al. Any-Step Density Ratio Estimation via Interval-Annealed Secant Alignment.

---

### Official Review · Reviewer_N7gR · 2025-10-25

**Soundness:** 2
**Presentation:** 3
**Contribution:** 3
**Rating:** 2
**Confidence:** 4

**Summary:**

The goal is to estimate the ratio of two densities $p_0$ and $p_1$ using their samples. The authors do so using the framework of time score matching, where a design choice if the path of distributions $(p_t)_{t \in [0, 1]}$ that is chosen by the user to link $p_0$ to $p_1$.

The authors upper-bound the estimation error of the density-ratio in terms of the path choice, and minimize the upper-bound. They obtain closed-form results (optimal time-reparameterization aka "schedule" of a given path) and empirical results (numerically optimizing over a family of paths).

**Strengths:**

The authors identify an interesting question that has not been solved in the literature of density ratio estimation using the time score identity. Their analysis is rather novel, and is reminiscent of similar results in the flow matching literature (where the estimation error is upper bounded by the training loss).

**Weaknesses:**

## **Minor**

- **Terminology.**

Line 159: the authors say the time score loss in (Choi et al., 2022) is known as the *sliced* time score matching objective. I have read (Choi et al., 2022) and have not come across the terminology of "sliced time score matching" in the main text. As far I know, (Choi et al., 2022)  just call their cost function "time score matching". Is "sliced" a terminology that you are adding? If so, it would be fair to explicitly say so.

---

## **Math Questions**

- **Reweighting.**
 In the time-score matching loss (Eq 6), do you use a reweighting function as (Choi et al., 2022) do?

- **On Section 4.1.**
  Is $\theta^\*$ the minimizer of the *empirical* loss, the *population* loss, or neither? Is it simply a fixed parameter value of the time-score model? If it corresponds to the *true data parameter*, then shouldn’t $L_{\text{TSM}}(\theta^\*) = 0$, assuming the neural network can perfectly approximate the true time score?

- **On Section 4.2.**
  You parameterize general paths between $p_0$ and $p_1$, but it is unclear how you enforce the boundary conditions. Specifically, how do you ensure that at $t = 1$, we have $p_t = p_1$?

- **Potential inconsistency in Section 4.1.**
  My understanding is that:
  $$
  \text{estimation error} \le e^L \times L_{\text{TSM}}(\theta^\*)
  \le e^L \times (L_{\text{STSM}}(\theta^\*) + \text{slack}),
  $$
  where $L$ is a Lipschitz constant of the path, and the slack term corresponds to the kinetic energy of the path.

  *Issue #1*. The paper states that “to minimize the final estimation error, one must minimize the ideal score-matching loss.”
  I believe this is a **sufficient** but not **necessary** condition, since the ideal score-matching loss only provides an upper bound.
  If that upper bound is loose, it may be possible to reduce the estimation error without reducing the ideal score-matching loss.

  *Issue #2*. You aim to minimize the estimation error with respect to the choice of path $p_t$ and therefore propose minimizing its upper bound over paths. However, your analysis focuses only on minimizing the *slack term* (the kinetic energy) while seemingly ignoring the term $L_{\text{STSM}}(\theta^\*)$, which also depends on the choice of path. It is possible, for instance, that a path minimizing the kinetic term actually increases $L_{\text{STSM}}(\theta^\*)$, in which case the upper bound (and the estimation error) may not improve.

---

## **Missing Reference**

The authors claim that the literature lacks a theoretical analysis of how the choice of path $p_t$ impacts the estimation error of the density ratio.
While this remains an open and interesting problem, it would be relevant to cite prior work that provides partial theoretical results in this direction.

In particular, **Theorem 4 of [1]** derives an upper bound on the estimation error of the density ratio as a function of the path’s Lipschitz constant—closely related to your Lemma 4.1, with a similar proof structure (see Appendix D.3 of that work).
This reference would be highly appropriate to discuss, as it represents the closest known theoretical result to your contribution.

Another reference that is less directly related but might be interesting to the authors is [2]. It also investigates the optimal path for reducing the estimation error of a density ratio using the time score identity, in the specific case when all the densities $p_t$ are parameterized only by their normalizing constant $Z_t$.

[1] Yu et al. *Density Ratio Estimation with Conditional Probability Paths.* ICML 2025.

[2] Chehab et al. Provable benefits of annealing for estimating normalizing constants... NeurIPS 2023.


---

## **Evaluation**

- **On Figure 2.**
  Figure 2 does not seem to show a substantial performance difference between optimal and suboptimal paths $p_t$.
  Could the authors comment on this observation?

**Questions:**

This is an interesting paper and I am glad the authors are tackling how the estimation error of the density-ratio depends on the choice of path: this is an open question in the community.

Could the authors address my questions detailed in the weaknesses section? In there, my main concerns are my "Issue #2" and "Evaluation" sections. I would be happy to raise my score if these are addressed.

---

> ### Author Response · Authors · 2025-11-18
> **Response to Reviewer N7gR, 1/3**
>
> We sincerely thank the reviewer for their thoughtful summary and for recognizing the novelty of our work in addressing an open question in this field. We appreciate the constructive feedback and address each of your points below.
>
> > **[Question 1]** In the time-score matching loss (Eq 6), do you use a reweighting function as (Choi et al., 2022) do?
>
> A: We **do not** use the explicit time-dependent reweighting function employed in Choi et al. (2022).
>
> Instead, inspired by Yu et al. (2025), we achieve a similar effect **implicitly through sampling**. Rather than weighting the loss, we replace uniform time sampling $t \sim \mathcal{U}(0, 1)$ **variance-based importance sampling**, where the sampling density is  $p(t) \propto \frac{1}{\mathrm{Var}_{p_t(x)}\left(s^{(t)}(x, t)\right) + \varepsilon}$. This concentrates training on time steps with **lower score variance**, i.e., more reliable score, thereby improving stability without introducing an explicit weighting term. Full details are provided in **Appendix B.2.5**.
>
> >  **[Question 2]** Is $\theta$* the minimizer of the empirical loss, the population loss, or neither? Is it simply a fixed parameter value...? If it corresponds to the true data parameter, then shouldn’t $L_ {TSM}(\theta^ {*}) = 0$...?
>
> A: We clarify below:
>
> - **$\theta^{\star}$ is the population loss minimizer of the ideal Time Score Matching objective $L_{\text{TSM}}$.** In fact, Proposition A.4 (Appendix A.6) proves that $\theta^{\star}$ uniquely minimizes *all* variants of the score matching loss we consider, including $L_{\text{TSM}}$, $L_{\text{STSM}}$, and $L_{\text{CTSM}}$, under standard regularity conditions.
> - In our theory, $\theta^{\star}$ represents the **optimal parameter** that perfectly recovers the true time score $s^{(t)}(x,t)$. Once trained, this $\theta^{\star}$ is indeed treated as a **fixed parameter** of the model during inference.
> - Regarding $L_{\text{TSM}}(\theta^{\star}) = 0$: yes, **in the ideal setting**, i.e., with infinite model capacity and infinite data, the learned $\theta^{\star}$ recovers the true score exactly, yielding zero ideal loss. This assumption is standard in theoretical analyses of score matching (e.g., Meng et al., Theorem 2).
>
> [2] Meng et al. Concrete score matching: Generalized score matching for discrete data.  NeurIPS 2022.
>
> > **[Question 3]** You parameterize general paths between $p_0$ and $p_1$, but it is unclear how you enforce the boundary conditions... Specifically, how do you ensure that at $t=1$, we have $p_t = p_1$?
>
> A: The boundary conditions are **built into our path parameterization** by design.
>
> We define the path schedule $\alpha(t)$ via the CDF $F(t)$ of a learnable Kumaraswamy Mixture Model (KMM) on $[0,1]$, setting $\alpha(t) = 1 - F(t)$. Since any CDF satisfies $F(0)=0$ and $F(1)=1$, we automatically get $\alpha(0) = 1,\alpha(1) = 0.$
>
> For the interpolant $x_t = \alpha(t) x_0 + \beta(t) x_1$, we use $\beta(t) = 1 - \alpha(t)$ (or $\beta(t) = \sqrt{1 - \alpha(t)^2}$), both of which satisfy $\beta(1) = 1$ when $\alpha(1) = 0$. Thus $x_{t=1} = \alpha(1)x_0 + \beta(1)x_1 = x_1,$ which guarantees $p_{t=1} = p_1$. The same logic ensures $p_{t=0} = p_0$.
>
> This CDF-based construction ensures valid boundary conditions **for any learned KMM**, without requiring extra constraints or penalties.
>
> > **[Missing References]** (1) In particular, **Theorem 4 of [1]** derives an upper bound on the estimation error of the density ratio as a function of the path’s Lipschitz constant—closely related to your Lemma 4.1, with a similar proof structure ... This reference would be highly appropriate to discuss .... (2) Another reference that is less directly related but might be interesting to the authors is [2]. It also investigates the optimal path for reducing the estimation error of a density ratio using the time score identity, in the specific case when all the densities $p_t$ are parameterized only by their normalizing constant $Z_t$.
>
> > [1] Yu et al. Density Ratio Estimation with Conditional Probability Paths. ICML 2025.
>
> > [2] Chehab et al. Provable benefits of annealing for estimating normalizing constants. NeurIPS 2023.
>
> A: Thank you for pointing out these relevant works. We have added both **Yu et al. (ICML 2025)** and **Chehab et al. (NeurIPS 2023)** to **Section 2 (Related Work)**, with brief discussions of their connections to our theory (highlighted in blue). Their inclusion strengthens the context of our contribution, and we appreciate the suggestion.

---

> ### Author Response · Authors · 2025-11-18
> **Response to Reviewer N7gR, 2/3**
>
> > **[Question 4, Issue #1]** ... The paper states that “to minimize the final estimation error, one must minimize the ideal score-matching loss.” I believe this is a **sufficient but not necessary** condition...
>
> A: You are absolutely correct that minimizing the upper bound is **sufficient but not strictly necessary** for reducing the true estimation error $\mathbb{E}[\Delta(x)]$. However, in practice, $L_{\text{TSM}}(\theta^\star)$ is a tractable proxy for the true error (Lemma 4.1) $\mathbb{E}_ {p_1}[\Delta(x)] \le e^L \cdot L_{\text{TSM}}(\theta^ {\star})$. Since $L_ {\text{TSM}}$ directly measures the fidelity of the learned time score to the ground truth and achieving minimum estimation error is contingent upon achieving maximum approximation fidelity to the true score, it remains a principled objective to optimize, even if the bound is not tight. We will revise the phrasing to “Minimizing the ideal loss is essential to achieving minimum estimation error,” to better reflect this nuance (highlighted in blue).
>
> > **[Question 4, Issue #2]** ... You aim to minimize the estimation error w.r.t. the path, but focus only on minimizing the slack (kinetic energy/variance) while seemingly ignoring that $L_{\text{STSM}}(\theta^*)$ also depends on the path. A path that reduces variance might increase $L_{\text{STSM}}$, in which case the upper bound (and the estimation error) may not improve.
>
> A: We fully agree: both terms depend on the path $\{\alpha_\phi, \beta_\phi\}$, and a poor trade-off could degrade performance. In early experiments, we tried **jointly optimizing** $L_{\text{STSM}}$ and $\mathcal{V}$ w.r.t. $\phi$ in the beginning, but observed **severe training instability** due to the strong coupling between path geometry and score targets.
>
> To resolve this, we adopt a **stable alternating optimization strategy**, inspired by the *stop-gradient* principle common in generative modeling [3]:
>
> - The path parameters $\phi$ are updated **only** by minimizing $\mathcal{V}$ (the variance term).
> - Every `update_freq` steps (see the code below), we **freeze** the new path and use it to update the **variance-based importance sampling distribution** $p(t) \propto \frac{1}{\mathcal{V}(t) + \varepsilon}.$
> - This updated $p(t)$ then **implicitly influences** $L_{\text{STSM}}$ by reshaping the distribution of sampled $x_t = \alpha(t)x_0 + \beta(t)x_1$, without requiring direct gradient flow from $L_{\text{STSM}}$ to $\phi$.
>
> This decoupling ensures stable training while still allowing the path to indirectly guide score learning through adaptive sampling. The mechanism is implemented in the `TimeSampler.update_weights` method.
> ```python
> class TimeSampler(nn.Module):
>     def __init__(self, weights_fn=None, uniform_eps=1e-5, min_weight=1e-8,
>                  update_freq=100, grid_size=10000, device='cpu', t_mode="IS", mu=-0.4, sigma=1):
>         """
>         Timestep sampling function p(t).
>         """
>         super().__init__()
>         self.weights_fn = weights_fn
>         self.eps = uniform_eps
>         self.grid_size = grid_size
>         self.min_weight = min_weight
>         self.update_freq = update_freq
>         self.device = device
>         self.register_buffer('t_grid', torch.linspace(uniform_eps, 1-uniform_eps, grid_size, device=device).unsqueeze(-1))
>         self.register_buffer('weights', torch.ones(grid_size, device=device) / grid_size)
>         self.step = 0
>         if t_mode == "IS" and weights_fn:
>             self.sampling_fn = self.importance_sampling
>         elif t_mode == "lognorm":
>             self.sampling_fn = lambda batch_size, device : self.lognorm(batch_size, mu=mu, sigma=sigma, device=device)
>         else:
>             self.sampling_fn = self.uniform
>
>     def forward(self):
>         pass
>
>     def uniform(self, batch_size, device='cpu'):
>         return torch.rand(batch_size, 1, device=device) * (1 - 2*self.eps) + self.eps
>
>     def importance_sampling(self, batch_size, device='cpu'):
>         if self.step % self.update_freq==1:
>             self.update_weights()
>         idx = torch.multinomial(self.weights, batch_size, replacement=True)
>         return self.t_grid[idx].to(device)
>
>     @torch.no_grad()
>     def update_weights(self):
>         w = self.weights_fn(self.t_grid).clamp(min=1e-12).squeeze()
>         self.weights.copy_(torch.clamp(w / w.sum(), min=self.min_weight))
>
>     def sample_t(self, batch_size, device="cpu"):
>         self.step += 1
>         return self.sampling_fn(batch_size, device)
> ```
>
> [3] Geng, Zhengyang, et al. Mean flows for one-step generative modeling. NeurIPS 2025.

---

> ### Author Response · Authors · 2025-11-18
> **Response to Reviewer N7gR, 3/3**
>
> > **[Question 5] Evaluation.** Figure 2 does not seem to show a substantial performance difference between optimal and suboptimal paths $p_ t$.
>
> A: We agree that on the 2D density estimation tasks in Fig. 2, the performance gap appears modest—particularly because strong heuristics like the VP path already work well in these simple settings. However, this actually demonstrates two key strengths of MinPV-DRE:
>
> 1. **Adaptive recovery of near-optimal paths**: In Fig. 2, MinPV **learns a path that matches or slightly improves upon** the best fixed heuristic (e.g., VP), as evidenced by the sharper structural details in the `tree` example (see also Appendix Fig. 3). This shows our framework can automatically adapt to the task’s intrinsic geometry.
>
> 2. **Dramatic gains in challenging tasks**: The real advantage of MinPV emerges in **high-discrepancy settings** where heuristic paths fail. As shown in Tab. 3, under severe density-chasm conditions (e.g., $MI > 20$ [4]), VP and other fixed paths **collapse or diverge**, while MinPV achieves an MSE of **1.02**—nearly **9× better** than the next-best baseline.
>
> Crucially, MinPV achieves this by incorporating **second-order path variance information** ($\mathcal{V}$, Eqs. 11–12), enabling it to **automatically select the optimal path geometry per task**: mimicking VP when appropriate, but switching to robust alternatives when needed.
>
> Thus, while Figure 2 illustrates **adaptability in the easy regime**, Tables 2–4 reveal transformative robustness in the hard regime, together validating the generality of our approach.
>
> [4] Rhodes, Benjamin, Kai Xu, and Michael U. Gutmann. Telescoping density-ratio estimation. NeurIPS 2020  (2020).

---

> ### Comment · Reviewer_N7gR · 2025-11-24
> **Answer to authors**
>
> Thanks for your response. Overall, it clarified some points for me and I believe the authors' approach of minimizing the path variance remains interesting. However, I still have some concerns.
>
> **Minor concern, Question 2**: Meaning of $\theta*$.
>
> I believe there is a misunderstanding. Recall that the population loss is computed using infinite data while the empirical loss is computed finite data.
>
> You say that $\theta*$ is the minimizer of the population loss. Unless I am mistaken, nowhere is this used in your proof in Appendix A.2. My understanding is that you treat $\theta*$ as **any** fixed parameter. In fact, if $\theta*$ was the minimizer of the population loss, then (assuming infinite model capacity), $\Delta (x) = 0$ in Eq 8.
>
> I believe this confusion comes from the notation. In statistics, $\theta*$ is typically the true model parameter (minimizer of the population loss), $\hat{\theta}$ is the estimated model parameter (minimizer of the empirical loss), and $\theta$ is any model parameter. I believe Eq 8 would be much clearer with $\theta$ instead of $\theta*$.
>
> **Major concern, Question 4, Issue #2**.
>
> Let me re-state the problem: as I understand it, the starting point of your method is the bound in Eq 10. There are actually three terms in that bound, from left to right:
> - term 1 is multiplicative and involves the smoothness of the path
> - term 2 is additive and involves a score-matching loss (STSM)
> - term 3 is additive and is a kinetic energy
>
> You claim to minimize the bound in Eq 10 with respect to the choice of path and this is the starting point of the entire method.   All three terms depend on the choice of path but you minimize only term 3. This could lead to:
> - term 1 blowing up: there is no discussion about this
> - term 2 blowing up: to avoid this, you do not compute the term 2 as shown in Eq 10, but instead apply a variance reduction technique
>
> I do not feel confident due to the fact that this was not clearly explained in the initial version. In particular, I am a bit worried that the dependency of terms 1 and 2 on the choice of path was swept under the rug. As things stand now:
>
> - I do not know if the good results you have are due your minimisation of the term 3, or due to your variance reduction of term 2, or both.
> - I do not know if your minimization of term 3 actually minimizes the bound in Eq 10. Maybe terms 1 and 2 increase a lot, depite the variance reduction you apply to term 2.
>
> I believe your final method should be valid but presently lacks justification. You refer to the Mean Flows paper but the parallel between their setup and yours is very tenous:
>
> - yes, the Mean Flows paper uses a stop-gradient, which in your case would mean not minimizing terms 1 and 2
> - but then you apply a variance-reduction technique on term 2 which has no obvious equivalent in the mean flows paper
>
> I believe this needs to be heavily clarified.

---

> > ### Author Response · Authors · 2025-11-27
> >
> > We sincerely thank the reviewer for their thoughtful follow-up and for encouraging us to clarify the interplay between the three terms in the error bound.
> >
> > > Response to Minor Concern, Question 2: Meaning of $\theta^*$
> >
> > Thank you for your careful reading and valuable feedback. **The original use of $\theta^*$ was illustrative, not restrictive.** It aimed to emphasize that a path-dependent term (V ) persists even under an ideal fit to the true time score. As the reviewer correctly observes, the error bound in Lemma 4.1 holds for any *θ* , not only an optimal one.
> >
> > **We have replaced $\theta^*$ with θ throughout the main text (Eqs. 7–10) and updated corresponding descriptions.**
> >
> > > Response to Major concern, Question 4, Issue #2
> >
> > Thank you for the insightful critique. **We clarify that our method jointly manages all three terms in Eq. 10 through a principled, decoupled optimization strategy—not by minimizing only Term 3.** Below we address each concern concisely:
> >
> > - **1. Term 1 (Lipschitz constant $L$) is controlled via Term 3 ($\mathcal{V}$).**  Minimizing path variance $\mathcal{V}$ (Term 3) directly reduces the path’s kinetic energy, yielding smoother trajectories and thereby lowering the Lipschitz constant $L$ (Term 1). This is rigorously shown in `Appendix A.5 (“Control of the Lipschitz Constant $L$ via Minimum Path Principle”)` and highlighted in the `“Remark on the Lipschitz Constant $L$” following Theorem 4.2`. Thus, Term 1 cannot blow up under our optimization.
> >
> > - **2. Term 2 ($\mathcal{L}_{\text{STSM}}$) is actively optimized and stabilized throughout training.**  Term 2 serves as the primary objective for updating the model parameters $\theta$ and is minimized at every training step, which is never ignored. To ensure stable optimization, we employ variance-based importance sampling (Appendix B.2.5) with $p(t) \propto 1/{\operatorname{Var}_{p_t(x)}\big(\partial_t \log p_t(x)\big)}$. This focuses training on low-variance (stable) time regions of the updated path, ensuring stable convergence in practice.
> >
> > - **3. The overall bound is minimized via an alternating scheme.**  Our method alternates between: (i) minimizing $\mathcal{V}$ to refine the path $\phi$, and (ii) using the updated $\phi$ to adjust $p(t)$, which in turn guides $\mathcal{L}_{\text{STSM}}$’s optimization.
> >
> >   This decoupled design, inspired by, though distinct from, Mean Flows, ensures that minimizing $\mathcal{V}$ simultaneously suppresses Term 1 and stabilizes Term 2. Consequently, the full bound in Eq. 10 is effectively minimized, not just Term 3.
> >
> > These mechanisms are now more clearly articulated in the revised manuscript to avoid ambiguity.

---

> ### Comment · Reviewer_N7gR · 2025-11-27
> **Answer to authors**
>
> I appreciate the clarification.
>
> **Regarding Term 1**. Thanks for Appendix A.5. It's helpful. I think the argument makes sense in a handwavy way. But it requires a more careful and detailed explanation. You say "It is a well-established principle that for a function on a finite interval, minimizing its L2-norm exerts a strong pressure to reduce its L∞-norm". That is not so clear: you could imagine something conceptually like the "square root of a Dirac", so one big and thin peak, whose L∞-norm explodes but whose L2-norm is one. Could you provide citations instead of claiming that "It is a well-established principle"?
>
> **Regarding Term 2**. You say: "Term 2 serves as the primary objective for updating the model parameters $\theta$ and is minimized at every training step, which is never ignored". But this is not relevant, because your goal is to minimize Term 2 with respect to the path $p_t$, not the model parameters $\theta$. I agree that your variance-reduction method "stabilizes" Term 2, but I would not claim that it minimizes Term 2 with respect to the choice of path $p_t$. Moreover, it would be helpful for you to explain what ``stabilizes" Term 2 means. My sense is that when you apply importance sampling to Term 2, you are reducing its dependency on the choice of path $p_t$. Perhaps you could show something like this rigorously or empirically.
>
> **Regarding Term 3**. I agree with your statement that "minimizing $\mathcal{V}$ simultaneously suppresses Term 1 and stabilizes Term 2", in a handwavy sort of way. But again, I think it requires more careful explanation: I am not sure that all reviewers noticed that the three terms were not jointly minimized, and even now, the explanations are very short and segmented. This should be made clear at the first reading. I disagree with your claim that "The overall bound is minimized via an alternating scheme". You say yourself that Term 2 is *stabilized* (using the variance-reduction), not minimized. And I think that's ok. I would simply avoid over-claiming things that may not be true and say something like: we develop a **heuristic** way of minimizing the bound, by *explicitly* minimizing Term 3, this *approximately* suppresses Term 1 and we *control* the remaining Term 2 using a variance-reduction technique.

---

> > ### Author Response · Authors · 2025-11-28
> > **Reponse to Reviewer N7gR. Summary (TL;DR)**
> >
> > We sincerely thank the reviewer for their exceptionally thoughtful and constructive feedback. Your precise critique has helped us significantly improve the clarity, rigor, and scientific honesty of our presentation.
> >
> > In response, we have carefully revised the manuscript to **avoid over-claiming** and to **explicitly align our language with the heuristic nature of our approach**, exactly as you suggested. Specifically:
> >
> > - We now clearly state that our method **explicitly minimizes Term 3 (path variance)**, **approximately suppresses Term 1 (Lipschitz constant $L$)**, and **controls/stabilizes Term 2 (score-matching loss)** via variance-reduced sampling, not through joint minimization of all three terms (revised paragraph `Remark on Stabilizing  STSM in Theorem 4.2`, line 232-233, line 237).
> > - We have updated `Section 4.1`, the Remark on the Lipschitz constant $L$, and `Appendix A.5` to emphasize the empirical and phenomenological nature of the $\mathcal{V}$–$L$ relationship, removing any implication of a general theoretical guarantee.
> > - We have clarified that "stabilizing" Term 2 refers to reducing gradient estimator variance through importance sampling (Appendix B.2.5), and explained how adaptive weighting (UW-SO, Appendix B.2.4) enables a natural, decoupled training schedule that balances the vastly different scales of Term 2 and Term 3.
> >
> > These changes ensure that, even on first reading, the reader understands our method as a **practical heuristic** grounded in empirical observation, not a formal optimization of the full error bound.
> >
> > Below, we address each of your three points in detail.

---

> > > ### Author Response · Authors · 2025-11-28
> > > **Reponse to Reviewer N7gR. Details**
> > >
> > > > **Regarding Term 1**
> > >
> > > The connection between path variance $\mathcal{V}$ and the Lipschitz constant $L$ is primarily *empirical* and *heuristic* in nature, which is why we present it as a Remark rather than a formal theorem. While minimizing $\mathcal{V}$ penalizes large path velocities (as seen in the analytic forms of Eq. 11–12, as illustrated in Figs. 4–7), translating this into a rigorous bound on $L = \sup_{x,t} |\partial_t \log p_t(x)|$ would require additional regularity assumptions on $(p_0, p_1)$ and the score model—beyond the scope of our current framework.
> > >
> > > Our goal in this remark is not to claim a universal theoretical guarantee, but to offer an *interpretive lens* for the empirical correlation we observe: paths with lower $\mathcal{V}$ consistently exhibit **visibly smoother trajectories** (see Figs. 4–7 in Appendix B.6), and our **overall method achieves lower estimation error**. We conjecture that, within the smooth function class induced by our KMM parameterization, minimizing $\mathcal{V}$ effectively suppresses extreme path velocities, thereby keeping $L$ under control in practice, even if a general proof remains open.
> > >
> > > We view this as a *phenomenological observation* that invites further theoretical investigation, rather than a closed result. The precise analytical relationship between path geometry, time-score regularity, and error bounds remains an interesting open question.
> > >
> > > We have added an explanation in "Remark on the Lipschitz Constant": "We discuss the empirical nature of this relationship in Appendix A.5." We have also updated more precise descriptions in Appendix A.5.
> > >
> > > > **Regarding Term 2**
> > >
> > > - **1. Clarifying the optimization target.** We fully agree: our method does not directly minimize Term 2 w.r.t. the path $\phi$. The path $\phi$ influences Term 2 **indirectly** through the data distribution $p_t(x)$ and the resulting time score $s^{(t)}$. Our strategy is decoupled:
> > >   - $\theta$ is optimized to minimize Term 2 for a given path;
> > >   - $\phi$ is optimized to minimize Term 3, which yields a smoother, lower-variance path.
> > > Thus, path optimization does *not* directly reduce $\mathcal{L}_ {STSM}(\theta; \phi)$ as a function of $\phi$, but it creates conditions under which the stochastic gradient estimator for $\mathcal{L}_ {STSM}$ becomes lower-variance, enabling more stable and reliable optimization with respect to $\theta$.
> > >
> > > - **2. What "stabilizes Term 2" means.** By "stabilize", we mean reducing the variance of the stochastic gradient estimator used to update $\theta$. Specifically, we adopt variance-based importance sampling (Appendix B.2.5). This concentrates sampling on time steps where the true time score has low variance, i.e., where the signal is most reliable. As a result, the Monte Carlo estimator of $\mathcal{L}_ {CTSM}$ has lower variance, leading to smoother, more stable training dynamics, **especially critical when the path itself is evolving**.
> > >
> > > - **3. Why decoupled optimization is necessary.** In practice, we observed that Term 3 and Term 2 often differ by **orders of magnitude** (e.g., $\mathcal{V} \gg \mathcal{L}_{\text{STSM}}$ in early training). Joint optimization without balancing would be dominated by $\mathcal{V}$, preventing meaningful score learning. To address this, we employ **Soft Optimal Uncertainty Weighting (UW-SO)** (Appendix B.2.4), which adaptively assigns weights $\lambda_1, \lambda_2$ to balance the two objectives. Empirically, this leads to a natural training schedule:
> > >   - **Early phase**: larger $\lambda_2$ prioritizes reducing path variance (Term 3), establishing a stable interpolation;
> > >   - **Later phase**: as $\mathcal{V}$ decreases, $\lambda_1$ increases, shifting focus to refining the score model (Term 2).
> > > This adaptive, decoupled strategy is key to our success on high-discrepancy tasks (e.g., Tab. 3).
> > >
> > > > **Regarding Term 3**
> > >
> > > We sincerely thank the reviewer for this insightful and constructive feedback. We fully agree that our original phrasing could mislead readers into thinking all three terms are jointly minimized.
> > >
> > > **We have revised the manuscript accordingly** to adopt the precise and humble framing the reviewer suggested:
> > > - In **Section 4.1**, we have replaced the sentence *"... we can explicitly minimize this overlooked term, leading to a strictly tighter error bound"* with *"... we can explicitly minimize this term, not as a guarantee of a tighter bound, but as a heuristic that empirically improves estimation accuracy and stability."*
> > > - We have also updated the **Remarks on L** (Appe. A.5) and **on $\mathcal{L}_ {STSM}$** (Section 4.1) to consistently use terms like "heuristic", "empirically", "control", and "stabilize" instead of "tight" or "guarantee".
> > >
> > > Thank you again for helping us improve the clarity and scientific rigor of our work.

---

### Official Review · Reviewer_HpCs · 2025-10-30

**Soundness:** 4
**Presentation:** 4
**Contribution:** 3
**Rating:** 8
**Confidence:** 4

**Summary:**

The paper studies density ratio estimation problems that integrate time-scores along some path. Even though any smooth path is in principle okay, the practical performance of such methods depends on the actual path when approximations of the time score are used. The authors quantify the approximation error trough variance of the time-score along the path and show that choosing a minimum-variance path is optimal, both theoretically and empirically.

**Strengths:**

This is in overall a strong contribution for the DRE literature. The idea of analysing the variance of the time-score is intuitive but to my best knowledge novel, and the authors deliver a well-executed study that makes the connection between the variance and the estimation quality clear. Theorem 4.2 is a worthy objective in itself, and the closed-form expressions provided in Proposition 4.3 are interesting. The practical method is well explained and the empirical results provided in broad set of experiments are strong, and already the explanation for (one) reason for the path-dependency of practical performance would be valuable as such even if not demonstrating improved accuracy in practice.

**Weaknesses:**

The empirical gain is clear, but not so major that this would be a transformative contribution that would qualitatively change the state-of-the-art.

**Questions:**

No specific questions.

---

> ### Author Response · Authors · 2025-11-18
>
> We sincerely thank the reviewer for their generous evaluation and for highlighting the novelty of analyzing the **variance of the time-score**, as well as the significance of Theorem 4.2 and Proposition 4.3.
>
> We agree that average gains may appear modest, but our contribution is **principled**: we resolve a long-standing theory–practice gap by identifying **path variance ($\mathcal{V}$)** as the key missing term in score-based DRE. By providing a **closed-form expression** for $\mathcal{V}$, we turn path selection from heuristic into **data-adaptive optimization**.
>
> This yields real-world impact: **MinPV-DRE** achieves SOTA *without manual path tuning* and shows **dramatic robustness** where fixed paths fail:
>
> | **Task**                       | **Best Fixed Path** | **MinPV (Ours)** | **Gain**    |
> | ------------------------------ | ------------------- | ---------------- | ----------- |
> | High-d MI (160d, MI=40)        | Linear: 9.17 MSE    | **1.02 MSE**     | **≈9×**     |
> | Additive Noise (Discontinuous) | VP: 0.0313 MSE      | **0.0117 MSE**   | **≈3–6×**   |
> | BSDS300                        | VP: −131.90 NLL     | **−143.97 NLL**  | **+12 NLL** |
>
> These are **qualitative improvements** in “density-chasm” regimes—precisely where reliability matters most. We believe this combination of **theoretical clarity** and **practical robustness** constitutes a meaningful advance for DRE.

---

> > ### Comment · Reviewer_HpCs · 2025-11-24
> > **Response**
> >
> > Thank you for the new results and clarifications. It a good question what is substantial improvement and perhaps 5-10 time improvement in MSE in certain tasks indeed should be counted as one.
> >
> > I still consider this to be a good paper with a clear contribution, but I am following closely the discussion with the other reviewers relating to novelty and the technical details. Some of the related works they pointed out indeed are highly relevant and I agree that discussing especially the relationship with Yu et al. (2025) in the revised version is important. Perhaps that could still be sharpened a bit? Now you mention in Section 2 that their result is 'closely related to our Lemma 4.1' but then in Section 4.1 do not saying anything more. It would be appropriate to expand also that section a bit, do explain in which sense the results are related and what kind of differences remain.

---

> > > ### Author Response · Authors · 2025-11-27
> > >
> > > We thank the reviewer for highlighting the importance of clarifying the relationship with Yu et al. (2025).
> > >
> > > In response, we have `added a new paragraph titled "Relationship to Existing Error Bounds"` immediately after Lemma 4.1 in Section 4.1 (`highlighted in blue in the revised manuscript`). This addition explicitly positions our result against their Theorem 4 and clarifies both connections and distinctions

---

### Author Response · Authors · 2025-12-03
**Summary of Reviews and Author Responses**

Dear PCs, SACs, ACs, and Reviewers,

Thank you very much for your dedicated efforts. To assist the newly assigned AC and help streamline their workload, we provide below a summary of the key points from the reviews and the reviewer-author discussions.

----

### I. Core Contribution Summary

The reviewers acknowledged the strength and novelty of our work in resolving the path dependency paradox in DRE:

- **Novel Principle (MinPV):** The core idea of identifying and analyzing the path variance ($\mathcal{V}$) of the time score as the critical, missing algebraic term that determines estimation fidelity is highly novel and a strong contribution (HpCs: Strengths, N7gR: Strengths, 8HVQ: Strengths 2).
- **Theoretical Tractability:** The derivation of the Minimum Path Variance (MinPV) principle (Theorem 4.2) and the provision of closed-form, analytic expressions (Prop. 4.3) enable a data-adaptive path optimization previously considered heuristic (HpCs: Strengths, 8HVQ: Strengths 2).
- **Empirical Impact:** The resulting MinPV-DRE method achieves superior stability and qualitative improvements (e.g., 5–10× MSE gain) in challenging high-discrepancy/density-chasm regimes where established fixed paths fail (HpCs: Follow-up).

### II. Addressing Methodological Rigor and Decoupled Optimization

> 1. Decoupled heuristic optimization (N7gR: Q4 #2 & Follow-up)

The primary discussion focused on clarifying the optimization of the error bound (Eq. 10: Term 1 × (Term 2 + Term 3)) and ensuring empirical gains were correctly attributed.

Reviewers questioned how minimizing only Term 3 $\mathcal{V}$ controls the full bound. We clarified that our training objective is a weighted sum of Term 2 $L_ {\text{STSM}}$ and Term 3 $\mathcal{V}$, jointly optimized as a principled heuristic: Term 2 ensures model fidelity, while Term 3 primarily drives the path geometry. More details:

- **Control Term 1 ($L$):** Minimizing the path's integrated energy ($\mathcal{V}$) acts as a heuristic for suppressing high-velocity peaks, thereby controlling the Lipschitz constant $L$ (Term 1). We clarified this relationship is phenomenological but strongly supported by the smooth learned paths.
- **Stabilize Term 2 ($L_ {\text{STSM}}$):** "Stabilizes" means reducing the variance of the  gradient for $L_ {\text{STSM}}$. This stability is achieved because our importance sampling $p(t)$ concentrates on low-variance (most reliable) path regions, leading to stable training.

> 2. Theoretical & implementation clarity

| Issue & Reviewer(s) | Concern | Action Taken |
|-|--|---|
| **Objective Discrepancy** (8HVQ: Q1; N7gR: Q4#2) |  Why use $L_ {\text{CTSM}}$ when the formula relies on $L_ {\text{STSM}}$? | Clarified that $L_ {\text{CTSM}}$  is used in practice because Prop. A.4 proves it shares the same minimizer as $L_ {\text{STSM}}$, preserving the algebraic decomposition. |
| **Missing Reference: Yu et al. (2025)** (N7gR:Missing Ref 1, HpCs: follow-up) | Contrast Lemma 4.1 with Yu et al. (2025)'s error bound. |Added comparison in Sec. 4.1: both use path smoothness, but differ in target (MSE vs. KL) and focus: $\mathcal{V}$ vs. $L_ {\text{CTSM}}$ efficiency. |
| **Weighting Function** (8HVQ: Q2; N7gR: Q1) |Isolate path effect from loss weighting. | Explained that variance-based sampling $p(t) \propto 1/\mathcal{V}$ is mathematically equivalent to optimal inverse-variance weighting, isolating path geometry as the optimized variable. |

> 3. Addressing minor methodological details

| Issue & Reviewer(s) | Concern | Action Taken |
|--|---|--------------|
| **Path Constraints** (N7gR: Q3) | How are boundary conditions enforced? | Confirmed by construction: KMM path $\alpha(t)=1-F(t)$ with CDF $F$ ensures $\alpha(0)=1, \alpha(1)=0$ ⇒ $p_ {t=1}=p_1$. |
| **Bound Justification** (8HVQ: W1) | Why exponents $n=\infty, m=1$ in Lemma 4.1? | This is the only choice that cleanly decomposes the bound into ($L_ {\text{TSM}}$) and the path variance ($\mathcal{V}$)—essential for our method. |
| **Scale & Scope** (8HVQ: W2) | Relatively small-scale problems.          | Justified: focus is on *hardness* (high-discrepancy/density-chasm regimes), which better validates theoretical robustness than scale alone. |
| **Broader Context** (8HVQ: Q3) | Link time score variance to Info. Geometry. | Confirmed time score = Fisher score w.r.t. t; clarified novelty is using $\mathcal{V}$ as *explicit analytic objective* for DRE path optimization. |
| **Parameter Meaning** (N7gR: Q2) | Clarify $\theta^*$ and the resulting estimation error $\Delta(\mathbf{x})$. | Revised notation: $\theta^*$ = optimal parameter minimizing population loss; replaced with $\theta$ in main text to avoid confusion.|

------

We believe these comprehensive clarifications, coupled with the introduction of the explicit principled heuristic framework, successfully resolve the major technical concerns. We appreciate the dedication of the reviewers and the AC in refining our work.

Sincerely,

Authors

---

### Meta-Review · Area_Chair_CsqZ · 2025-12-31

**Summary:**

The paper addresses a significant practical issue in score-based Density Ratio Estimation (DRE): although theory suggests that the choice of interpolation path between distributions $p_0$ and $p_1$ should not affect the final ratio, in practice it significantly affects estimator variance and stability, as they illustrate at the beginning of the paper. The paper further proposes a closed-form expression for optimising the path and empirically demonstrates that it yields significant results. Reviewer N7gR started with a negative score, providing several avenues for improvement, ranging from missing literature to issues with the theorems. However, the reviewer concluded that the paper could make a significant contribution at the end of the review. The interaction between the reviewer and the authors converges toward a more positive view from the reviewer. The other two reviewers were mostly positive about the paper, and their questions were minimal and addressed mainly by the authors.

**Reviewer Concerns:**

v started with a grade of 2, but acknowledged that the paper was of good quality and suggested that he might raise his score if convinced by the authors. In this case, the interaction between reviewers and authors moved positively, as they engaged before the system was halted. From this interaction, the reviewer would have significantly raised his score. He would have increased it to 6 (my estimate).

The other two reviewers are a PhD student and his advisor, which might lead to some overlap in views. Reviewer HpCs was initially very positive, although he noted that Reviewer N7gR's negative comments could have reduced his score, depending on where the original AC would have led the discussion. The last reviewer was positive from the start, and the authors addressed most of his remarks.

**Reviewer Scores:**

See the discussion above.

---

### Decision · Program_Chairs · 2026-01-26

Accept (Poster)